# Late middle Miocene caviomorph rodents from Tarapoto, Peruvian Amazonia

**Myriam Boivin**[1]*, **Laurent Marivaux**[2], **Walter Aguirre-Diaz**[3], **Aldo Benites-Palomino**[3,4], **Guillaume Billet**[5], **François Pujos**[6], **Rodolfo Salas-Gismondi**[3,7], **Narla S. Stutz**[2,8], **Julia V. Tejada-Lara**[2,3], **Rafael M. Varas-Malca**[3], **Anne H. Walton**[9], **Pierre-Olivier Antoine**[2]

1 Laboratorio de Paleontología de Vertebrados, Instituto de Ecorregiones Andinas (INECOA), Universidad Nacional de Jujuy, CONICET, San Salvador de Jujuy, Jujuy, Argentina, 2 Laboratoire de Paléontologie, Institut des Sciences de l'Evolution de Montpellier (ISEM, UMR 5554, CNRS/UM/IRD/EPHE), Université de Montpellier, Montpellier, France, 3 Departamento de Paleontología de Vertebrados, Museo de Historia Natural—Universidad Nacional Mayor San Marcos (UNMSM, DPV-MUSM), Lima, Peru, 4 Paläontologisches Institut und Museum, Universität Zürich, Zürich, Switzerland, 5 Département Origine et Evolution, Muséum national d'Histoire naturelle (MNHN, CR2P-UMR 7207 CNRS/MNHN/Sorbonne Université), Paris, France, 6 Laboratorio de Paleontología, Instituto Argentino de Nivología, Glaciología y Ciencias Ambientales (IANIGLA), CCT–CONICET–Mendoza, Mendoza, Argentina, 7 BioGeoCiencias Lab, Facultad de Ciencias y Filosofía/CIDIS, Laboratorios de Investigacion y Desarrollo (LID), Centro de Investigación para el Desarrollo Integral y Sostenible (CIDIS), Universidad Peruana Cayetano Heredia, Lima, Peru, 8 Programa de Pós-Graduação em Geociências, Universidade Federal do Rio Grande do Sul, Porto Alegre, Rio Grande do Sul, Brazil, 9 Springfield Technical Community College, Springfield, Massachusset, United States of America

* mboivin@idgym.unju.edu.ar

**Data Availability Statement:** All relevant data are within the manuscript and its Supporting Information files (S1 Table and S2 File). The fossil material is permanently stored in the Vertebrate

## Abstract

Miocene deposits of South America have yielded several species-rich assemblages of caviomorph rodents. They are mostly situated at high and mid- latitudes of the continent, except for the exceptional Honda Group of La Venta, Colombia, the faunal composition of which allowed to describe the late middle Miocene Laventan South American Land Mammal Age (SALMA). In this paper, we describe a new caviomorph assemblage from TAR-31 locality, recently discovered near Tarapoto in Peruvian Amazonia (San Martín Department). Based on mammalian biostratigraphy, this single-phased locality is unambiguously considered to fall within the Laventan SALMA. TAR-31 yielded rodent species found in La Venta, such as the octodontoid *Ricardomys longidens* Walton, 1990 (*nom. nud.*), the chinchilloids *Microscleromys paradoxalis* Walton, 1990 (*nom. nud.*) and *M. cribriphilus* Walton, 1990 (*nom. nud.*), or closely-related taxa. Given these strong taxonomic affinities, we further seize the opportunity to review the rodent dental material from La Venta described in the Ph.D. volume of Walton in 1990 but referred to as *nomina nuda*. Here we validate the recognition of these former taxa and provide their formal description. TAR-31 documents nine distinct rodent species documenting the four extant superfamilies of Caviomorpha, including a new erethizontoid: *Nuyuyomys chinqaska* gen. et sp. nov. These fossils document the most diverse caviomorph fauna for the middle Miocene interval of Peruvian Amazonia to date. This rodent discovery from Peru extends the geographical ranges of *Ricardomys longidens*, *Microscleromys paradoxalis*, and *M. cribriphilus*, 1,100 km to the south. Only one postcranial element of rodent was unearthed in TAR-31 (astragalus). This tiny tarsal bone most likely documents one of the two species of *Microscleromys* and its morphology indicates terrestrial generalist adaptations for this minute chinchilloid.

Paleontological collection of the "Museo de Historia Natural, Universidad Nacional Mayor de San Marcos" (MUSM), Lima, Peru.

**Funding:** LM and POA received funding from The Leakey Foundation and the LabEx CEBA (ANR-10-LABX-0025-01). POA received funding from the National Geographic Society and from the French 'Agence Nationale de la Recherche' (ANR) in the framework of the GAARAnti program (ANR-17-CE31-0009). LM received funding from the CoopIntEER CNRS-CONICET (n° 252540). POA and FP received funding from the ECOS-SUD/FONCyT (n° A-14U01) international collaboration programs. MB received funding from the 'Laboratoire de Planétologie et de Géodynamique de Nantes'. The funders had no role in study design, data collection and analysis, decision to publish, or preparation of the manuscript.

**Competing interests:** The authors have declared that no competing interests exist.

# Introduction

Caviomorphs (Caviomorpha Wood, 1955) are hystricognathous rodents originating from South America, such as guinea pig, New World porcupines, chinchillas, and spiny rats. Nowadays, they show a great taxonomical diversity and a large array of lifestyles (arboreal, fossorial, semi-aquatic, and terrestrial) [1]. Among caviomorphs, two main clades, each divided into two superfamilies, are recognised [2–4]: Erethicavioi Boivin, 2019 (comprising Cavioidea Fischer, 1817 + Erethizontoidea Bonaparte, 1845 + stem-groups) and Octochinchilloi Boivin, 2019 (comprising Chinchilloidea Bennett, 1833 + Octodontoidea Waterhouse, 1839 + stem-groups). Their evolutionary history extends back to the late middle Eocene, with the earliest representatives of the group so far recorded in Peruvian Amazonia [5–8].

The Miocene record of caviomorphs is particularly well documented. Numerous fossil-bearing localities yielded abundant specimens and species-rich assemblages of rodents [9]. Most of them are situated at high and middle latitudes, notably in Argentina (e.g., Pinturas, Sarmiento, Santa Cruz, Collón Cura, Cerro Azul and Ituzaingo formations) [9–15] and Chile (e.g., Laguna del Laja and Pampa Castillo) [16, 17]. At low latitudes of South America, two remarkable and very species-rich caviomorph faunas are known, La Venta in Colombia (Honda Group; late middle Miocene) [18] and Acre in Brazil (Solimões Fm. [Formation]; late Miocene; see [19]). Since the early discoveries of fossil-bearing localities in the Honda Group of the upper Magdalena valley in 1923, considerable field efforts led by petroleum companies and several scientific teams were performed in this region of Colombia, and notably in the La Venta area, until 1992 [20, 21]. As a result, 140 localities over 52 different stratigraphic levels, yielding thousands of fossils of many vertebrate groups were discovered in the Honda Group dating from the late middle Miocene [22]. In particular, the recognition in 1945 of the first fossil remains of primates in the Magdalena valley [23, 24] has been one of the most significant discoveries that motivated numerous subsequent field expeditions [20]. The mammal fauna from La Venta encompasses at least 77 species and 80 genera, representing 29 families [25, 26]. Concerning rodents, around 20 taxa documenting the four main caviomorph superfamilies have so far been reported (excluding the indeterminate genera and species) [18, 27–30]. In addition to La Venta, the Peruvian faunas from MD-67 (Madre de Dios) [31], the Fitzcarrald local fauna (near Atalaya, Ucayali) [32, 33], and CTA-45 (near Contamana, Loreto) [34], also middle Miocene in age, document representatives of the four caviomorph superfamilies (4, 8, and 2 taxa, respectively). Finally, a fourth late middle Miocene caviomorph-bearing locality also occurs in low latitudes, from the Socorro Formation in Venezuela, but so far it has only yielded a fragmented distal femur of a giant caviomorph [35].

Over the last decade, our yearly paleontological surveys in the Tarapoto area (San Martín Department, Peruvian Amazonia), have allowed the discovery of a new fossil-bearing locality, TAR-31, in the vicinity of the Juan Guerra village. This locality is considered to be late middle Miocene in age (Laventan SALMA) based on mammalian biostratigraphy (see [36]; see Material and Methods section of this paper). TAR-31 yielded a diverse assemblage of aquatic and terrestrial vertebrates, including numerous caviomorph specimens that are the subject of this paper. The primary purpose of the present work is to describe the new caviomorph materials found in TAR-31. This locality shares taxa or closely related taxa with La Venta fauna described by one of us (AHW) in the framework of her PhD [37]: *Microsteiromys*, *Ricardomys*, and *Microscleromys*. Because *Microsteiromys*, *Ricardomys*, *Microscleromys*, and the species included within these genera were not formally described (*nomina nuda*) [18, 37, 38], here we also revise the material from La Venta attributed to these taxa, in order to clarify their systematics.

## Material and methods

### TAR-31, Tarapoto, Peru

The rodent fossil material from Peruvian Amazonia described in the present work originates from the TAR-31 locality, situated in the vicinity of the small town Juan Guerra, along the Mayo River (Tarapoto area, San Martín Department, Peru; Fig 1A and 1B). TAR-31 consists of a 10–15 cm-thick yellow microconglomerate interbedded within a grey cross-stratified and sandstone-dominated fluvial unit (Fig 2A [36]). The latter is intercalated between thick violin-grey variegated paleosols pointing to the existence of a meandering river with sustainable floodplain [36]. These levels were originally mapped as belonging to the lower member of Ipururo Formation (middle Miocene in age in the Huallaga basin) [39–42], an assignation in agreement with the nature of their facies and depositional environment sequence.

The fossil content of the TAR-31 encompasses plants, amber clasts, crabs, chondrichthyans, osteichthyans, anurans, chelonians, crocodylomorphs, birds, and mammals (including metatherians, xenarthrans, liptoterns, notungulates, sirenians, chiropterans, primates, and caviomorph rodents). The TAR-31 mammal assemblage includes a didelphid marsupial (currently under study by one of us, NSS), the interatheriine notoungulate *Miocochilius* sp., the didolodontid *Megadolodus* sp., the platyrrhine primate *Neosaimiri* aff. *fieldsi* [36], and the caviomorph fauna described here (see 'Age of TAR-31' section of this present work). This assemblage recalls some of the Laventan SALMA localities: Quebrada Honda in Bolivia (13.1–12.2 Ma) [43–46], the Fitzcarrald local fauna in SE Peru [33], and especially the lower part of the Villavieja Formation in the La Venta area in Colombia (13.8–11.6 Ma; [47] and see above). Accordingly, the TAR-31 locality most likely documents the late middle Miocene Laventan SALMA [36].

The material from TAR-31 was collected by excavating and wet screening (2 and 1 mm meshes) about 550 kg of sediment during our yearly field expeditions (August 2015, 2016, and

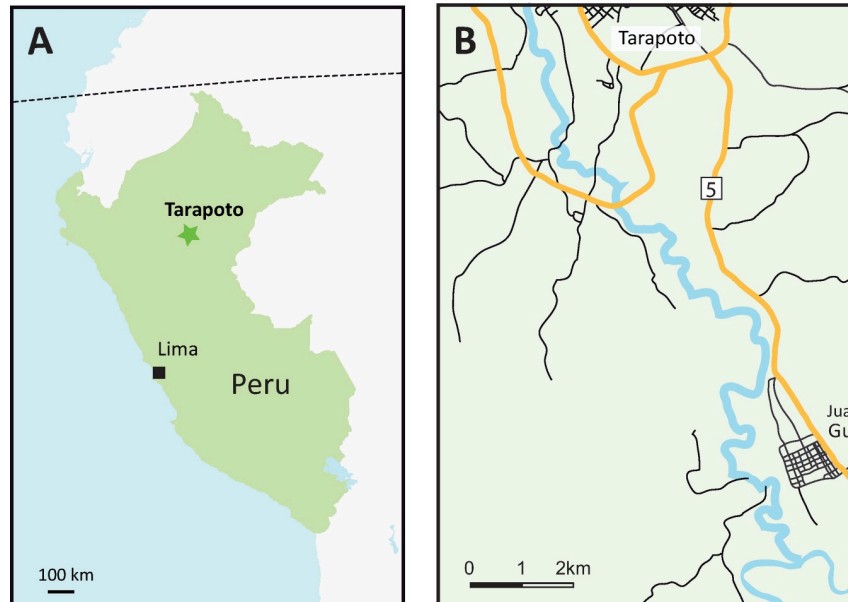

**Fig 1. Geographic location of TAR-31 (San Martín Department, Peru; late middle Miocene, Laventan SALMA).** (A) Location map of the Tarapoto area (star symbol) in Peru. (B) Location map of the TAR-31 (star symbol) in the Tarapoto area. SALMA, South American Land Mammal Age.

2018). The caviomorph material corresponds to about 400 isolated teeth (complete or fragmentary), one fragment of mandible (MUSM 4643), one fragment of maxilla (MUSM 4375), and one astragalus (MUSM 4658). No permits were required for the described study, which complied with all relevant regulations. The field work on TAR-31 was carried out in the framework of the ongoing cooperation agreement between the 'Museo de Historia Natural de la Universidad Nacional Mayor San Marcos' (Lima, Peru) and the 'Institut des Sciences de l'Evolution de Montpellier-Université de Montpellier' (France). The TAR-31 fossil specimens are permanently stored in the palaeontological collection of the 'Museo de Historia Natural, Universidad Nacional Mayor de San Marcos', Lima, Peru (MUSM).

## La Venta area, Colombia

The fossil rodent material from La Venta re-analysed here (originally studied in the framework of the Ph.D. of Anne H. Walton) [18, 37] originates from the badland exposures located in the vicinities of the Villavieja and La Victoria villages, in the upper valley of the Magdalena River (Huila Department, Colombia). Numerous fossil-bearing localities are known in the La Venta area (and the Magdalena upper valley in general) [26]. These are assigned to two successive formations of the Honda Group: La Victoria and Villavieja formations [48]. The studied material from this area was found in: (i) one locality from La Victoria Formation (IGM-DU loc. 075, between the Chunchullo Beds and the Tatcoa Sandstone Beds), and (ii) distinct localities from the Villavieja Formation, including several localities in the Fish Bed (CVP 5, 8, 9, 10, 10A, and 13B), two localities in the Monkey Bed (IGM-DU loc. 006–1 and IGM-DU loc. 022 or UCMP loc. V4536), and one in the El Cardón Red Beds (IGM-DU loc. 032 and its screen-washed locality 032–5). The Unit between the Chunchullo and Tatacoa Sandstone Beds is 149.3 m-thick, and is composed of very fine sandstones and mudstones alternating with metric medium-grained sandstone lenses [48]. The Villavieja Formation is divided into the Baraya Member, including the Fish and Monkey beds, and the Cerro Colorado Member, including the El Cardón Red Beds [48]. The Baraya Member is very fossiliferous and is composed mainly of gray mudstones and sandstones with minor layers of red mudstones, while the Cerro Colorado Member is less fossiliferous and consists of thick horizons of red mudstones with a very small amount of volcanic litharenites and chert litharenites [48]. The La Venta fauna is late middle Miocene in age (13.8–12.05 Ma) based on $^{40}$Ar/$^{39}$Ar datings realised on volcanic units from different localities of the Honda Group (on biotites, hornblendes, and plagioclases; 13.8–12.2 Ma) and magnetostratigraphic correlations of the Honda Group (13.61 [base of C5ABn chron] –12.05 Ma [top of C5An.1n]) [26, 47, 49]. The La Venta area yielded a very rich vertebrate fauna including chondrichthyans, osteichthyans, amphibians, squamates, chelonians, crocodylomorphs, birds and mammals (metatherians, xenarthrans, astrapotheres, litopterns, notoungulates, cetaceans, chiropterans, sirenians, primates and caviomorph rodents) [21, 22].

The caviomorph remains from La Venta included in this work were formerly studied by one of us [18, 37]. In a doctoral thesis manuscript, Walton [37] described several new genera and species such as *Microsteiromys jacobsi*, *Ricardomys longidens*, *Microscleromys paradoxalis*, and *Microscleromys cribriphilus*. As the latter taxa were not formally described, they should be considered as not available following the articles 8, 9, 11, 13 of the International Code of Zoological Nomenclature (i.e., *nomen nudum*; p. 132–143, 246 [38]). As formally recognised for *nomina nuda* ("A *nomen nudum* is not an available name, and therefore the same name may be made available later for the same or a different concept; in such a case it would take authorship and date [Arts. 21, 50] from that act of establishment, not from any earlier publication as a nomen nudum" [38], we chose to keep the taxonomic names proposed by Walton [37]. For reasons related to the covid pandemic and confinement, the original material from La Venta

could not be directly observed. The description and comparison of this material in the present work is based on published photos and drawings [18, 37]. Most of these specimens were collected by surface collecting and wet screening (2 and 0.6 or 0.25 mm meshes) in the years 1985 through 1989. This material is stored in the 'Museo Geológico José Royo y Gómez, Servio Geológico Colombiano' (IGM; formerly, 'Instituto Nacional de Investigaciones en Geociencias, Mineria y Química') and the 'Departamento de Geociencias, Universidad Nacional de Colombia' (UNC), Bogotá, Colombia, as well as in the Field Museum of Natural History, Chicago (FMNH), USA. Some specimens do not have collection number (unnumbered specimens). We used their field number to refer to them (S1 Table).

## Imaging

For the largest specimens (mandibular and maxillary fragments) and high-crowned teeth found at TAR-31, we used X-ray microtomography (μCT-scan) for obtaining three-dimensional digital models (3D surface renderings). The dental/jaw specimens and the astragalus were scanned with a resolution of 6 μm and 5 μm, respectively, using a μCT-scanning station EasyTom 150/Rx Solutions (Montpellier RIO Imaging [MRI], ISEM, Montpellier, France). They have been virtually delimited by manual segmentation under AVIZO 7.1 software (Visualization Sciences Group). The specimens were prepared within a 'Label Field' module of AVIZO, using the segmentation threshold selection tool. The other teeth figured in this paper were photographed with a Scanning Electron Microscope (SEM) HITACHI S 4000 (Institut des Neurosciences de Montpellier [INM], France). Some of the pictures of the astragalus are photographs of the original specimen, which are the result of the fusion of multifocus images obtained with an optical stereomicroscope (Leica M 205C) connected to a camera (Leica DFC 420C; ISEM).

## Nomenclature and comparisons

The main terminology for the rodent dentition follows Boivin & Marivaux [50], and the literature cited therein. Some extant caviomorphs and related extinct forms show a highly derived/specialised dental pattern with respect to that of the oldest representatives of this group (for which the aforementioned nomenclature was based on). This is the case of two taxa described below: (i) Caviidae indet. gen. et sp., for which nomenclatures proposed by several authors [51–55] were followed (Fig 2A); and (ii) the adelphomyine *Ricardomys longidens*, for which we partly followed the nomenclature proposed by Patterson & Pascual [56] (Fig 2B and 2C). Lower case letters are used for lower dentition (dp, for decidious premolar; p, for premolar; m, for molars) and upper case letters for upper dentition (dP, for decidious premolar; P, for premolar; M, for molars). The astragalar nomenclature is based on Ginot et al. [57], Rose & Chinnery [58], and Wible & Hughes [59]. The caviomorph taxa cited in the text and used for dental comparisons are listed in S2 Table. Three isolated teeth from MD-67 locality in Peru (MUSM 1974, 1975, and 4298; early middle Miocene) were originally attributed to cf. *Microsteiromys* sp [31]. They are reassigned here in light of the new material from Tarapoto. For the MUSM 4658 astragalus, we used the same comparative sampling as Boivin et al. [60]. In addition, we consulted supplementary astragalar material attributed to the erethizontid *Steiromys duplicatus* (MACN A 10055–78, 10081 and 10082; see [61]), the caviids *Dolicavia minuscula* (MMP 10055–78) and *Galea leucoblephara* (INBIAL-CV 00290), the neoepiblemid *Neoepiblema* sp. (probably *N. acreensis*; UFAC 5249 and 61840), the octodontids *Abalosia castellanosi* (MMP 1439-M) and *Pithanotomys columnaris* (MACN-Pv 7429–7431), the echimyid *Eumysops chapalmalensis* (MACN-Pv 17868), and the ctenomyid *Actenomys priscus* (MMP 367-S and 395-S). When the fossils described in this paper are compared with several taxa, they are

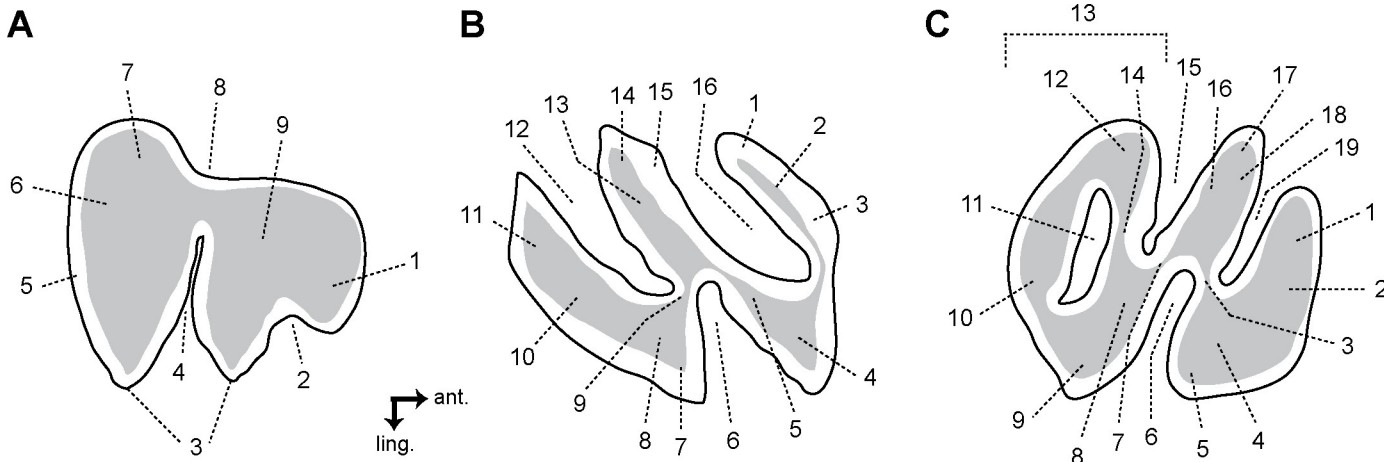

**Fig 2.** Dental nomenclature for cavioid (A) and adelphomyine (B, C) rodent teeth in occlusal view. (A) p4; 1, anterior projection of the anterior lobe; 2, interprismatic furrow; 3, apexes; 4, hypoflexid; 5, enamel; 6, dentine; 7, posterior lobe; 8, longitudinal furrow opposite to the hypoflexid (= primary internal flexid); 9, anterior lobe. (B) Lower molar; 1, metaconid (+ its posterior arm?); 2, first lamella; 3, metalophulid I; 4, protoconid; 5, ectolophid; 6, hypoflexid; 7, anterior outgrowth of the hypoconid; 8, hypoconid; 9, anterior arm of the hypoconid; 10, posterolophid; 11, third lamella; 12, metaflexid; 13, second lamella; 14, entoconid; 15, hypolophid; 16, confluence between the anteroflexid and the mesoflexid. (C) Upper molar; 1, first lamella; 2, anteroloph; 3, posterior arm of the protocone (= lingual protoloph); 4, protocone; 5, posterior outgrowth of the protocone; 6 hypoflexus; 7, mure; 8, anterior arm of the hypocone; 9, hypocone; 10, posteroloph; 11, confluence of the distal mesoflexus with the metaflexus; 12, mesostyle; 13, third lamella; 14, third transverse crest (= mesoloph and/or mesolophule); 15, mesial mesoflexus; 16, second lamella; 17, paracone; 18, labial protoloph; 19, paraflexus; ant., anterior; ling., lingual. (B, C) Note that the position of the fused structures is speculative. The dental terminology is modified after several authors [50–56]. Not to scale.

primarily listed according to their chronostratigraphic order (from the oldest to the most recent) and then alphabetically.

## Nomenclatural acts

The electronic edition of this article conforms to the requirements of the amended International Code of Zoological Nomenclature, and hence the new names contained herein are available under that Code from the electronic edition of this article. This published work and the nomenclatural acts it contains have been registered in ZooBank, the online registration system for the ICZN. The ZooBank LSIDs (Life Science Identifiers) can be resolved and the associated information viewed through any standard web browser by appending the LSID to the prefix "http://zoobank.org/". The LSID for this publication is: urn:lsid:zoobank.org:pub:ECE79D67-9203-4929-9CCF-7C8D51C8326E. The electronic edition of this work was published in a journal with an ISSN, and has been archived and is available from the following digital repositories: LOCKSS.

## Dental associations

Working on isolated teeth requires the consideration of distinct levels of morphological variation: (i) variation depending on the dental locus; (ii) variation owing to the occlusal wear; (iii) intra-specific or intra-taxonomic variation or polymorphism; and (iv) inter-specific or inter-taxonomic variation. In this study, several taxa from TAR-31 are described. The association of the material attributed to these taxa is based on several complementary criteria: (i) a similar occlusal size; (ii) a similar crown height (for a given occlusal size and stage of wear); (iii) a similar proportion of the maximum mesiodistal length and the maximum linguolabial width; (iv) a similar occlusal outline; (v) a similar occlusal pattern (bunolophodonty/lophodonty; taeniodonty/non-taeniodonty; presence/absence, development, position of cusp[-id]s/styl[e/id]s;

presence/absence, continuity/discontinuity, length, height and obliquity of loph[-id]s; and presence/absence and types of fusion of dental structures); and (vi) a compatible occlusal pattern between upper and lower teeth. For teeth at advanced stages of wear, we took into account the potential impact of occlusal wear on characters aforementioned. Based on a wide array of closely- or distantly-related rodent taxa, we preferred to use a size criterion for the recognition of m1s vs m2s and M1s vs M2s (i.e., m1/M1s to be smaller than the m2/M2s in a given species). The recognition of other loci is based on the occlusal shape of teeth (for more details, see descriptions in the Systematic Palaeontology section) and on the presence and/or absence of mesial/distal vertical contact facets due to contiguous teeth. The occlusal shape of M3s is further characterised by a smaller and more labial hypocone with respect to the protocone than on M1–2s. However, the recognition of molar loci is quite ambiguous for several taxa, especially for the first two upper molars. In that case, we determined these loci with uncertainty (e.g., lower molar, m1?, m2?, m3?, upper molar, M1 or M2, M1 or M2?, M3?).

## Measurements and statistics

For each dental specimen of TAR-31, we measured both the maximum mesiodistal length (len) and linguolabial width (wid), as well as the maximum lingual crown height (Hg) and labial crown height (Hb; S1 Table). These measurements were made by M. Boivin with a measuroscope Nikon 10 and a Keyence Digital Microscope VHX-2000F. For *Microsteiromys jacobsi* and *Ricardomys longidens* from La Venta, we used the measurements available in Walton [37] (S1 Table). However, we noted some inconsistencies regarding the m3 of the type of *Ricardomys*, IGM 183847, between the measurements reported in Table 2, p. 33 [37] and the scaled photo of the specimen in Fig 9A, p. 31 [37]. Similarly, the measurements of the type of *M. jacobsi* (FMNH PM 54672 in Table 6, p. 58 [37]) do not match with its scaled representation (in Fig 16A, p. 56 [37]). For these two specimens, we chose to use the measurements made from their figuration in the Figs 9A and 16A [37]. For the Laventan material attributed to *Microscleromys*, the measurements made from the figures of the specimens in Walton (Figs 11–13 [37]) and those from Walton (Fig 24.2 [18]) are different. We extracted the measurements from the first study (Ph.D.) because the specimens were photographed (instead of drawn) with a larger scale and more material is represented. The hypsodonty index of a tooth (HI) [62] equals its crown height divided by its anteroposterior length (H/ML); teeth with a HI < 1 are considered as brachydont, those with a HI = 1 are considered as mesodont, and those with a HI > 1 are considered as hypsodont. The latter are either protohypsodont when they still have roots, or euhypsodont if they lack roots [51]. Two hypsodonty indexes were calculated in function to the side of the crown considered: from Hb (HIb) and from Hg (HIg). The HI values mentioned in the descriptions were measured on specimens at the early stages of wear. The MUSM 4658 astragalus from TAR-31 was measured following the protocol of Ginot et al. [57] (Table 1).

**Table 1. Measurements (mm) of the astragalus from TAR-31, MUSM 4658.**

| ABW | AmTL | ATL | ATW | EL | EW | HH | HW | LBH |
|---|---|---|---|---|---|---|---|---|
| 2.265 | ~1.220 | 3.088 | 2.649 | 1.447 | 1.231 | 0.954 | 1.316 | 1.069 |
| LTL | MBH | MTL | NL | SL | SW | TW | mTAH | |
| 1.860 | 1.682 | 2.015 | 1.382 | 1.211 | 1.035 | 1.832 | 0.725 | |

ABW, Astragalus Body Width; AmTL, Astragalar-medial Tarsal facet Length; ATL, Astragalus Total Length; ATW, Astragalus Total Width; EL, Ectal facet Length; EW, Ectal facet Width; HH, Head Height; HW, Head Width; LBH, Lateral Body Height; LTL, Lateral Trochlear Length; MBH, Medial Body Height; MTL, Medial Trochlear Length; NL, Neck Length; SL, Sustentacular Facet Length; SW, Sustentacular Facet Width; TW, Trochlear Width; mTAH, medial Trochlear Arc Height.

In order to help identifying the number of species and discriminating which species of *Microscleromys* are present in TAR-31 and La Venta, we performed three types of statistical analyses on the maximum mesiodistal length and linguolabial width by locus (i.e., dp4, p4, m1–2(?), m3(?), dP4, P4, M1–2(?), M3(?)) with R v.4.0.3. [63] (S1 and S2 Files). First, we tested statistical differences between potential taxonomic groups mentioned by Walton (*M. paradoxalis nom. nud.*, *M. cribriphilus nom. nud.*, *M. ?paradoxalis*, and *Microscleromys* sp.) [37] and *Microscleromys* from TAR-31 using Kruskal-Wallis, Permutation Analysis of Variance, Student and Wilcoxon tests, depending to what preconditions were satisfied, with the 'dunn.test', 'RVAide-Memoire', 'stats' and 'trend' packages [63–66]. Second, partitioning analyses were conducted until five (dp4), nine (m3) or fifteen (other loci) groups depending the number specimens for each loci with the 'cluster' package [67]. Finally, we performed several Bayesian models on the most numerous loci including several specimens previously attributed to *M. cribriphilus* and *M. paradoxalis* (i.e., p4, m1–2(?), and M1–2(?)) in order to obtain the most probable partioning with the 'glm' function of 'MASS' and 'stats' packages [63, 68]. We tested 39 models for the M1–2(?) and 51 models for the p4 and m1–2(?) including the partitionings following the previous systematic hypotheses and those obtained with the partitioning analyses (see S1 File).

## Abbreviations

**Institutional abbreviations.** **FMNH,** Field Museum of Natural History, Chicago, USA; **IGM,** Servicio Geológico Colombiano (before Instituto Nacional de Investigaciones en Geociencias, Mineria y Química [INGEOMINAS]), Museo Geológico José Royo y Gómez, Bogotá, Colombia; **IGM-DU,** Field numbers from expeditions by the INGEOMINAS in cooperation with Duke University. Specimens deposited at the Museo Geológico José Royo y Gómez, Bogotá, Colombia; **INBIAL-CV,** Instituto de Biología de la Altura, San Salvador de Jujuy, Argentina; **ING-KU,** Field numbers from expeditions by the INGEOMINAS in cooperation with Kyoto University. Specimens deposited at the Museo Geológico José Royo y Gómez, Bogotá, Colombia; **LACM,** Los Angeles County Museum, Los Angeles, USA; **MCZ,** Museum of Comparative Zoology, Cambridge, USA; **MACN,** Museo Bernardino Rivadavia, Buenos Aires, Argentina; **MLP,** Museo de La Plata, La Plata, Argentina; **MMP,** Museo Municipal De Ciencias Naturales Lorenzo Scaglia, Mar del Plata, Argentina; **MNHN,** Museum National d'Histoire Naturelle, Paris, France; **MPEF-PV,** Museo Paleontológico Egidio Feruglio, Trelew, Argentina; **MUSM,** Museo de Historia Natural de la Universidad Nacional Mayor San Marcos, Lima, Peru; **UCMP,** University of California Museum of Paleontology, Berkeley, USA; **UFAC,** Coleção de Paleovertebrados do Laboratório de Pesquisas Paleontológicas, Universidade Federal do Acre, Rio Branco-AC, Brazil; **UNC,** Departamento de Geociencias, Universidad Nacional de Colombia, Bogotá, Colombia.

**Other abbreviations.** **CVP,** locality belongs to the Fish Bed in La Venta area collected by screenwashing during the expeditions by the INGEOMINAS in cooperation with Duke University; **IGM-DU loc.,** fossiliferous locality found during the expeditions by the INGEOMINAS in cooperation with Duke University in La Venta area; **UCMP loc.,** fossiliferous locality found during the expeditions by the University of California in La Venta area.

## Systematic paleontology

Nomenclatural Remark: The new species and genera described below must be referred to Boivin & Walton, 2021, following the article 50.1 and the "recommendation 50A concerning multiple authors" of the International Code of Zoological Nomenclature (p. 182 [38]). This is of particular interest for *nomina nuda* that were previously used by Walton [18, 37] and made available here, following the International Code of Zoological Nomenclature [38].

Rodentia Bowdich, 1821

Ctenohystrica Huchon, Catzeflis & Douzery, 2000

Hystricognathi Tullberg, 1899

Caviomorpha Wood, 1955

Erethicavioi Boivin, 2019 in Boivin, Marivaux & Antoine, 2019

Erethizontoidea Bonaparte, 1845

*Microsteiromys* Boivin & Walton gen. nov. urn:lsid:zoobank.org:act:70F9B9F7-51EB-4FFF-913C-2E89EBF5F9DB

## Type species

*Microsteiromys jacobsi* gen. et sp. nov.

## Species content

Only the type species.

## Derivation of name

Based on the original description of the species [37]: 'In reference to its small size and general resemblance to the erethizontoid *Steiromys* Ameghino, 1887'.

## Geographical and stratigraphical distribution

La Venta, Baraya and Cerro Colorado members, Villavieja Fm. (Laventan SALMA, late middle Miocene), Huila Department, Colombia.

## Generic diagnosis

As for the type and only known species.

*Microsteiromys jacobsi* Boivin & Walton sp. nov. urn:lsid:zoobank.org:act:1DF51272-C949-4105-BC5C-299F88252206

Fig 3 and S1 Table

*Nom. nud. Microsteiromys jacobsi* Walton, 1990, Fig 16A–16C, p. 56.

*Nom. nud. Microsteiromys jacobsi* Walton, 1997, Figs 24.1B and 24.2B, p. 394, 395.

## Holotype

FMNH PM 54672, right mandibular fragment bearing dp4–m3 (Walton, 1990: Fig 16A; Walton, 1997: Figs 24.1B and 24.2B; Fig 3A).

## Referred material

IGM-DU 89–249, left mandibular fragment bearing p4–m1 (Walton, 1990: Fig 16C; Fig 3B); IGM-DU 88–034, right posterior fragment of lower tooth (Walton, 1990: Fig 16B; Fig 3C).

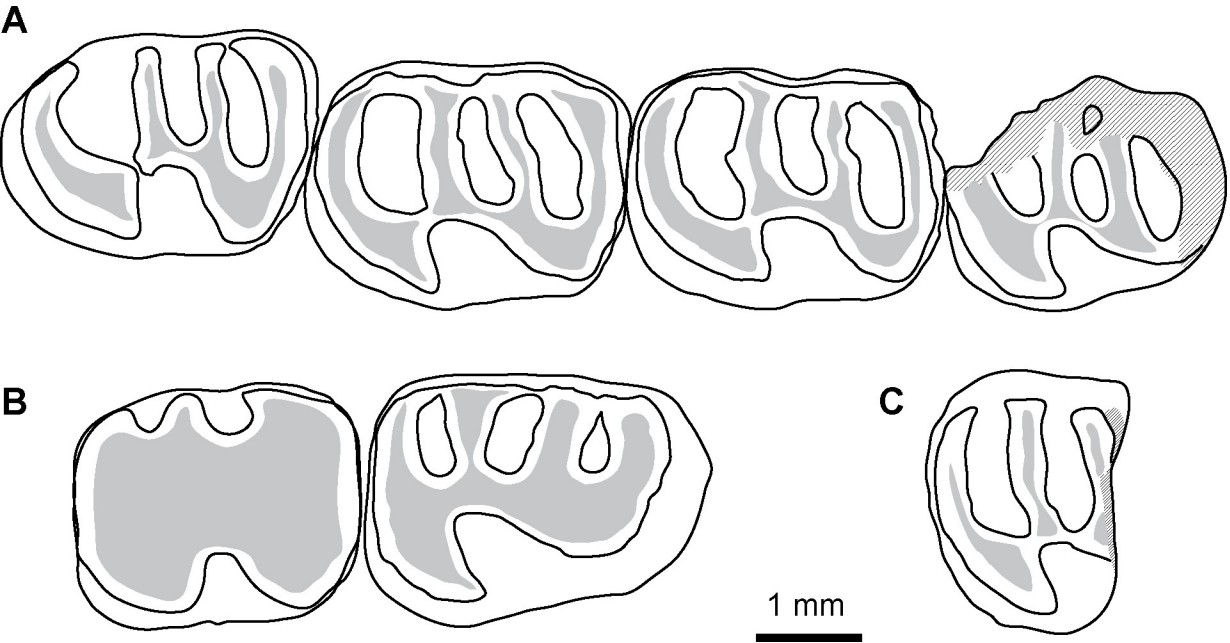

**Fig 3. _Microsteiromys jacobsi_ gen. et sp. nov. from La Venta, Colombia (late middle Miocene, Laventan SALMA).** (A) FMNH PM 54672***, right mandibular fragment bearing dp4–m3. (B) IGM-DU 89–249, left mandibular fragment bearing p4–m1 (reversed), (C) IGM-DU 88–034, right posterior fragment of m1 or m2. Occlusal views. The asterisks appoint the holotype material. Based on Walton (Fig 16A-16C [37]). The illustrations of the fossil specimens are computerised schematic line drawings.

### Derivation of name

Based on the original description of the species [37]: 'In honour to Dr. Louis L. Jacobs, palaeontologist and educator'.

### Type locality

IGM-DU loc. 022 or UCMP loc. V4536 (only for the type), Monkey Beds, La Venta Badlands, upper Magdalena Valley, Huila Department, Colombia.

### Other localities

CVP 10 (for IGM-DU 88–034), Fish Bed and screewashed locality 032–5 (for IGM-DU 89–249), El Cardón Red Bed, La Venta Badlands, upper Magdalena Valley, Huila Department, Colombia.

### Formation and age

Baraya and Cerro Colorado members, Villavieja Fm., late middle Miocene (i.e., Laventan SALMA).

### Diagnosis

Rodent characterised by a small size and brachydont teeth. Lower teeth with a tetralophodont and non-taeniodont pattern, complete metalophulid I, and oblique and complete ectolophid. dp4s with a mesiodistally short and linguolabially wide trigonid and a curved metalophulid I. Large p4s with a linguolabially wide trigonid. Mesiodistally elongated lower molars with a narrow mesoflexid and well-extended metaflexid. Differs from _Eopululo_, _Protosteiromys_,

*Branisamyopsis*, *Hypsosteiromys*, *Parasteiromys*, *Eosteiromys*, *Steiromys*, and *Neosteiromys* in having a smaller size; from *Eosteiromys annectens* and *Steiromys duplicatus* in having non-taeniodont lower teeth; from *Hypsosteiromys* in having brachydont lower teeth with transverse lophids and a complete metalophulid I, and less elongated dp4s; from *Shapajamys*, *Branisamyopsis*, *Eosteiromys homogidens*, *Steiromys*, and *Coendou insidiosus* in having tetralophodont dp4s; from *Eosteiromys segregatus* in having a longer p4 with a long anterior arm of the entonconid; from *Shapajamys*, *Palaeosteiromys*, *Protosteiromys*, *Branisamyopsis*, *Eosteiromys annectens*, *Steiromys*, and *Neosteiromys* in having more rectangular lower molars; from *Palaeosteiromys*, *Branisamyopsis australis*, *Branisamyopsis praesigmoides*, *Steiromys duplicatus*, and *Neosteiromys* in having always tetralopodont lower molars; from *Shapajamys*, *Branisamyopsis praesigmoides* and *Steiromys duplicatus* in having an ectolophid always complete and oblique on lower molars; from *Shapajamys* in having an ectolophid always oblique on lower molars; from *Palaeosteiromys* in having lower molars with a metaflexid more extended; from *Protosteiromys*, *Branisamyopsis*, *Eosteiromys homogidens*, *Steiromys detentus*, and *Neosteiromys* in having a narrower mesoflexid on lower molars.

## Description

FMNH PM 54672 (Fig 3A) and IGM-DU 89–249 (Fig 3B) are two mandibular fragments bearing dp4–m3 and p4–m1, respectively. The body of the FMNH PM 54672 mandible is anteriorly broken at the level of the anterior part of the lower diastema, which makes visible the lower incisor alveola. Posteriorly, the angular apophysis and most of the ascending ramus, including the mandibular condyle, are missing. The coronoid process is broken at its base. Labially, a mental foramen is located at the midpoint of the body of the mandible height (i.e., dorsoventrally). The notch for the insertion of the tendon of the *zygomatico-mandibularis pars infraorbitalis* is medium-sized, below dp4–m1, and centrally situated on the labial edge of the mandible. The anterior tip of the masseteric crest ends below the m1, and links the notch for the insertion of the tendon of the *zygomatico-mandibularis pars infraorbitalis*, at the level of its posteroventral region. The masseteric crest is posteriorly broken. It is posteroventrally directed and prominent in its anterior part.

The dp4 of the FMNH PM 54672 mandibular fragment (Fig 3A) is mesially and distolingually broken at the level of the metaconid, metalophulid I, mesostylid, entoconid, and posterolophid. This tooth is more worn and shorter than the molars (m1–3) beared by FMNH PM 54672. It is characterised by a short trigonid, which has an equal width with the talonid. This dp4 is brachydont and non-taeniodont. Despite its damaged state, no supplementary lophids are visible in addition to the metalophulid I, second transverse cristid, hypolophid and posterolophid, which suggests that the FMNH PM 54672 dp4 was tetralophodont. As this tooth is at an advanced stage of wear, the cuspids are not well defined to the lophids. On the lingual margin of the tooth, the anteroflexid and mesoflexid are closed, thereby suggesting that the mesostylid (or its arms) was connected to the metaconid (or its posterior arm) and its posterior arm reaches the entoconid (or its anterior arm). The metalophulid I, mesially convex, seems to be complete and strongly connected to the protoconid labially (hypothesis 1 on Fig 4A1; see the remark below). This cuspid is crestiform and faintly linked to the second cristid. The latter is slightly oblique (i.e., mesiodistally directed). It bears posteriorly a tiny and horrizontal cristulid, which joins the mesial margin of the hypolophid, dividing the mesoflexid into two small and round fossettids. The labiodistal aspect of the second cristid links to a transverse hypolophid via a very short and oblique ectolophid. The hypoconid is mesiolingually-distolabially compressed. It displays strong anterior arm, anterior outgrowth and posterolophid. The anterior outgrowth is mesiolabially oriented and

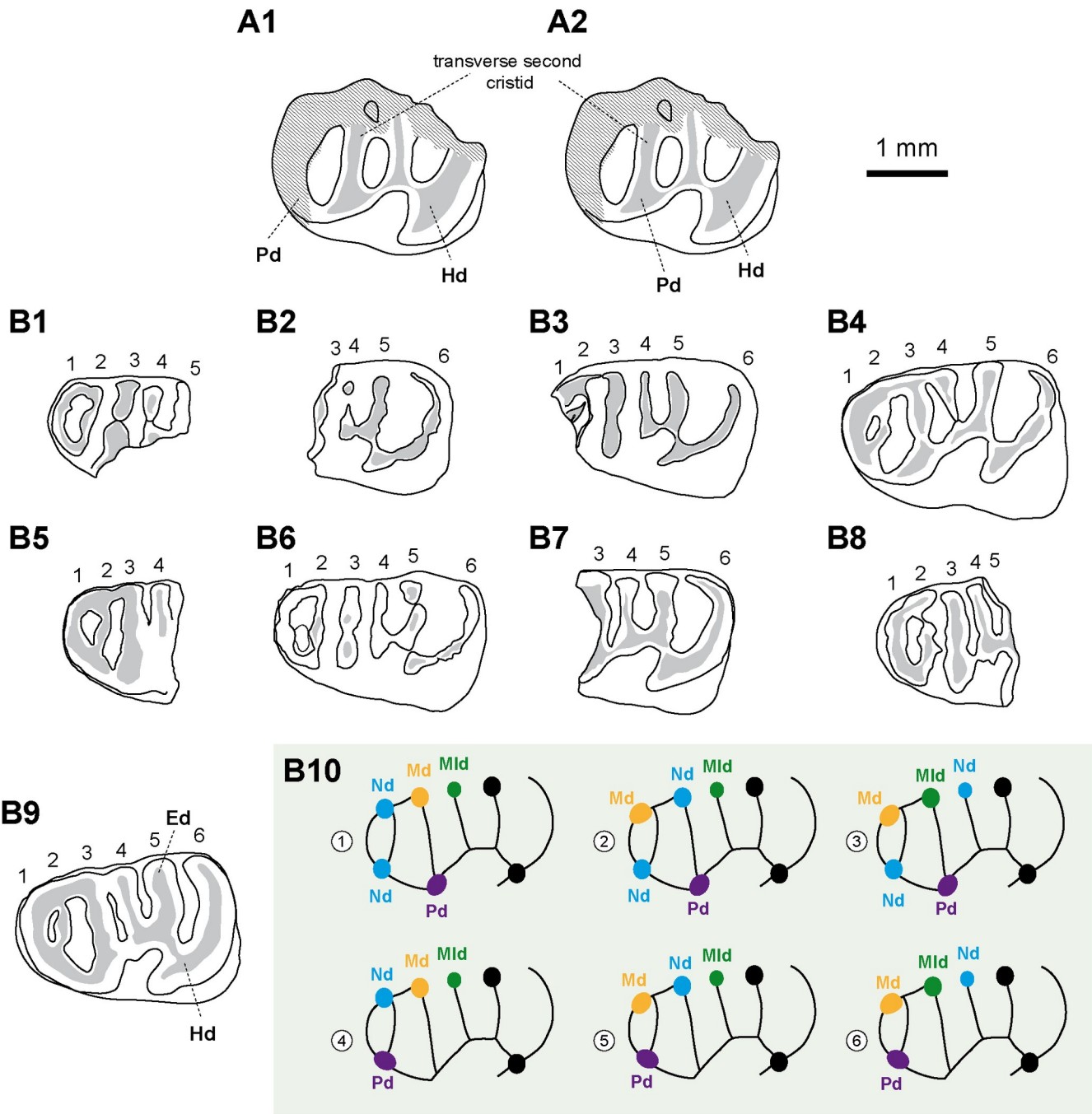

**Fig 4.** Hypotheses regarding the homology of structures on dp4 of the erethizontoid *Microsteiromys jacobsi* (A) and *Nuyuyomys chinqaska* (B). (A1) Hypothesis 1: protoconid in anterior position, aligned with the metalophulid I (followed here). (A2) Hypothesis 2: protoconid in posterior position, aligned with the second transverse cristid. (B1–9) Equivalent transversal cristids (and their associated cuspids/stylids) indicated by numbers (1–6) on all dp4s of *N. chinqaska*. (B10) Different hypotheses regarding the position of the anterior cuspids/stylids based on the morphology of MUSM 4333. For the transversal cristids 1 and 2, the position and number of the structures can slightly vary depending the considering specimen (e.g., MUSM 4332). In the case of the hypotheses 1–4, the neoconid(s) can be present on the cristid 1 or not (correspond solely to thickenings of this cristid). The hypotheses 1 and 3 are the most consistent with the morphology and development of the dental structures. (A) IGM-DU 89–249, (B1) MUSM 4326, (B2) MUSM 4327, (B3) MUSM 4328, (B4) MUSM 4329, (B5) MUSM 4325, (B6) MUSM 4332, (B7) MUSM 4331, (B8) MUSM 4330, and (B9) MUSM 4333. Ed, entoconid; Hd, hypoconid; Md, metaconid; Mld, mesostylid; Nd, neoconid; Pd, protoconid. The illustrations of the specimens are computerised schematic line drawings.

delimits with the anterior arm of the hypoconid, hypolophid, ectolophid, second transverse cristid and protoconid, a triangular-shaped hypoflexid, oriented distolingually, and showing a wide labial opening. Although the lingual part of the posterolophid is missing, the metaflexid appears lingually closed as is the case as well for the anteroflexid and the mesoflexid.

The p4 of the IGM-DU 89–249 mandibular fragment (Fig 3B) is much longer and moderately worn with respect to the associated m1, which shows a very advanced state of wear characterised by a fusion all lophids and cuspids/stylids. As the FMNH PM 54672 dp4, it is brachydont, tetralophodont and non-taeniodont. However, it is much longer than the dp4 and has a mesial margin more straight. The protoconid corresponds to a large dentine plateform on the mesiolabial corner of the tooth. It is mesially connected to a metalophulid I, reaching the metaconid on the lingual margin, and distally to the second cristid and to an oblique and long ectolophid. The labial part of the metalophulid I is mesiolingually directed, while its lingual part is slightly directed distolingually. The metaconid is faintly linked to the mesostylid, which appears to have an equal size to the former. Due to the moderate state of wear of the tooth, the presence of a posterior arm of the metaconid cannot be assessed, but if it would be present, the latter would be short. The mesostylid bears short anterior and posterior arms. Comparatively, the entoconid has longer arms, especially its anterior arm, reaching the posterior arm of the mesostylid. The second cristid is complete and slightly mesiolingually directed, whereas the hypolophid and the posterolophid are transverse. There is a connection between the latter and the posterior arm of the entoconid, which lingually closes the metaflexid. Thus, all the lingual flexids are closed and form round-oval fossettids. The anterior arm of the hypoconid is strong and mesiolingually oriented, whereas its anterior outgrowth is long and mesiolabially oriented. These cristids, with the ectolophid and protoconid, delimit a widely open and retroverse hypoflexid (triangular-shaped and distolingually directed).

The m1–3 of the FMNH PM 54672 specimen (Fig 3A) have a similar/compatible size with the unique and very worn molar (m1) borne by the IGM-DU 89–249 mandibular fragment (Fig 3B). On FMNH PM 54672, all molars are roughly equal in length. The m1 and m2 have a similar width, whereas m3 is narrower. The latter is characterised by a shorter posterolophid due to a linguolabial compression of its posterior part. The morphology of lower molars has the same pattern than that of dp4 and p4. As dp4 and p4, molars are brachydont, tetralophodont, non-taeniodont with a strong anterior arm of the hypoconid, and they have an oblique ectolophid and a triangular-shaped hypoflexid showing a wide labial opening. This flexid has a general distolingual direction, but the latter varies depending on the locus: it is slight on m3, more oblique on dp4 and m1, and the most oblique on p4 and m2. As dp4 and p4, m1–2 display all their lingual flexids closed, whereas m3 has a mesoflexid opened by a shallow and thin furrow, and a posterolophid well-separated from the entoconid. On m1–2, this cuspid appears to have a long posterior arm On m1–3, the metalophulid I is complete, transverse to slightly arched. The metaconid bears a long posterior arm strongly connected to the mesostylid (or its anterior arm). The second cristid and hypolophid are close and separated by a narrow mesoflexid (i.e., furrow-like flexid). These two transverse cristids are straight and roughly parallel on m1 and m3. On m2, most part of the second cristid (i.e., lingual part) is slightly distolingually directed. Because of a thinning on its central part on m1 and its orientation change on m2, the second cristid may be composed of two cristids on these lower molars: a neomesolophid lingually and a posterior arm of the protoconid labially. On all molars, the posterolophid is curved, especially on m3. The metaflexid/metafossettid is mesiodistally well extended.

## Remark

Considering the pattern of dp4s of some erethizontoids (*Steiromys detentus*, MLP 15–339; *Erethizon dorsatum*, MCZ 51367) [69], another hypothesis can be proposed with respect to the position of the protoconid on the FMNH PM 54672 dp4 (hypothesis 2 on Fig 4A2): this cuspid could be situated more posterolingually than in hypothesis 1 and aligned with the second transverse cristid. The discovery of a pristine dp4 of this species would make it possible to decide between these two proposed hypotheses.

## Comparisons

Brachydonty, non-taeniodonty, extended and moderately narrow flexids, weak obliquity of lophids, and p4 with a wide trigonid suggest erethizontoid affinities for these specimens from La Venta. They are markedly smaller than most of erethizontoid species, except for *Kichkasteiromys raimondii*, *Shapajamys labocensis* [70], *Palaeosteiromys amazonensis* [71], *Noamys hypsodonta* [72] and the material from the early middle Miocene of Madre de Dios, Peru formerly attributed to cf. *Microsteiromys* sp. (MD-67; [31] and see below). *Kichkasteiromys raimondii*, *N. hypsodonta* and the taxon from MD-67 being documented only by upper teeth, direct morphological comparisons with the material of La Venta cannot be made. The lower molars from La Venta differ from those of *Shapajamys* and *Palaeosteiromys* in being more rectangular. They are characterised by an ectolophid always complete and oblique, contrary to *Shapajamys*, and by a metaflexid more extended than in *Palaeosteiromys*. These specimens from La Venta are always tetralophodont, contrary to those of *Palaeosteiromys*, *Branisamyopsis australis*, *Branisamyopsis praesigmoides*, *Steiromys duplicatus* and *Neosteiromys*, which can exhibit a pentalophodont pattern by the addition of a mesial neolophid [71, 73–77]. By its rounded shape as well as its short and wide trigonid, the FMNH PM 54672 dp4 is close to the dp4s of the extant species *Coendou insidiosus*. The FMNH PM 54672 dp4 is tetralophodont, as dp4s of *Hypsoteiromys* [74, 78], but the latter are clearly more elongated, with a longer trigonid and more extended lingual flexids. Contrary to FMNH PM 54672, dp4s of other erethizontoids are elongated with a pentalophodont (*Eosteiromys homogidens*, *Eosteiromys annectens* and *C. insidiosus*) or hexalophodont (*Shapajamys*, *Branisamyopsis*, ?*Eosteiromys* sp. nov. *sensu* Candela [69], and *Steiromys*) pattern. MNHN 1903-3-18 (Fig 32 [79]) was originally determined as a m3 of *Protosteiromys asmodeophilus*. However, its trigonid is clearly narrower than its talonid, a feature better characterising a premolar (p4 or dp4). MNHN 1903-3-18 is tetralophodont and non-taeniodont as p4s and dp4s from La Venta, but has lingual flexids much more extended. The material from La Venta differs from *Eopululo*, *Protosteiromys*, *Branisamyopsis*, *Hypsosteiromys*, *Parasteiromys*, *Eosteiromys*, *Steiromys* and *Neosteiromys* [73–76, 78–82] in having a smaller size; from *Hypsosteiromys* in having brachydont lower teeth with transverse lophids and a complete metalophulid I; from *E. annectens* and *S. duplicatus* in having non-taeniodont lower teeth; from *Branisamyopsis*, *S. duplicatus*, *Steiromys verzii sensu* Candela [74], and some p4s of *S. detentus* in having a p4s with tetralophodont pattern; and from *Eosteiromys segregatus* in having a longer p4 with a long anterior arm of the entonconid; from *Protosteiromys*, *Branisamyopsis*, *E. annectens*, *Steiromys* and *Neosteiromys* in having more rectangular lower molars; from *Protosteiromys*, *Branisamyopsis*, *E. homogidens*, *Steiromys detentus* and *Neosteiromys* in having a narrower mesoflexid on lower molars; from *B. praesigmoides* and *S. duplicatus* in having an ectolophid always complete on lower molars. To sum up, these few specimens from La Venta show a set of distinctions (i.e., small size; brachydonty; tetralophodont and non-taeniodont lower teeth; a complete metalophulid I, and an oblique and complete ectolophid on lower teeth; round dp4s with a short and wide trigonid; mesiodistally elongated lower molars with narrow mesoflexid and well-extended metaflexid) that allows us to refer

them to a new genus and new species. Thus, we validate *Microsteiromys jacobsi* Walton, 1990, which had remained until now as a *nomen nudum*.

*Nuyuyomys* Boivin & Walton gen. nov. urn:lsid:zoobank.org:act:360EBF95-C067-4C81-81F4-8FAD41067E7E

## Type species

*Nuyuyomys chinqaska* gen. et sp. nov.

## Species content

Only the type species.

## Derivation of name

From *Nuyuy*, flooding in Quechua language, referring to the location of both fossil localities yielding this genus (at the water level only during the driest season and below it most of the year) and *-mys*, mouse in Greek, a classical suffix for rodents.

## Geographical and stratigraphical distribution

Pilcopata (MD-67), unnamed Fm. (Colloncuran-Laventan, middle Miocene), Madre de Dios Department, Peru; Tarapoto/Juan Guerra (TAR-31), lower member, Ipururo Fm. (Laventan, late middle Miocene), San Martín Department, Peru.

## Generic diagnosis

As for the type and only known species.

*Nuyuyomys chinqaska* Boivin & Walton sp. nov. urn:lsid:zoobank.org:act:353264FD-EB4C-428F-9F6D-EFC6DC1D7A04

Fig 5 and S1 Table

2013 cf. *Microsteiromys* sp. Antoine et al., Fig 3M, 3N, p. 94.

## Holotype

MUSM 4308 (Fig 5R), right M1?.

## Referred material

MUSM 4325–4329 (Fig 5F), left dp4s; MUSM 4330–4333 (Fig 5A), right dp4s; MUSM 4334 (Fig 5B), left p4; MUSM 4335–4339, left lower molars; MUSM 4340–4342, right lower molars; MUSM 4343, 4344, left m1s; MUSM 4345–4348 (Fig 5C and 5G), right m1s; MUSM 4349–4357 (Fig 5H, 5J and 5K), left m2s; MUSM 4358–4363 (Fig 5D and 5L–5N), right m2s; MUSM 4364 (Fig 5I), left m3; MUSM 4365, 4366 (Fig 5E and 5O), right m3s; MUSM 4284–4288 (Fig 5Z), left dP4s; MUSM 4289–4294 (Fig 5P, 5U, 5C' and 5D'), right dP4s; MUSM 4295 (Fig 5Q), left P4; MUSM 4296, 4297, right upper molars; MUSM 4299, right M1 or M2?; MUSM 4300–4306, left M1?s; MUSM 4307–4314 (Fig 5A' and 5B'), right M1?s; MUSM 4315, right M2 (or M3)?; MUSM 4316–4320 (Fig 5S and 5W), left M2?s; MUSM 4321, 4322, right M2?s; MUSM 4323 (Fig 5X), left M3; MUSM 4324 (Fig 5T), right M3.

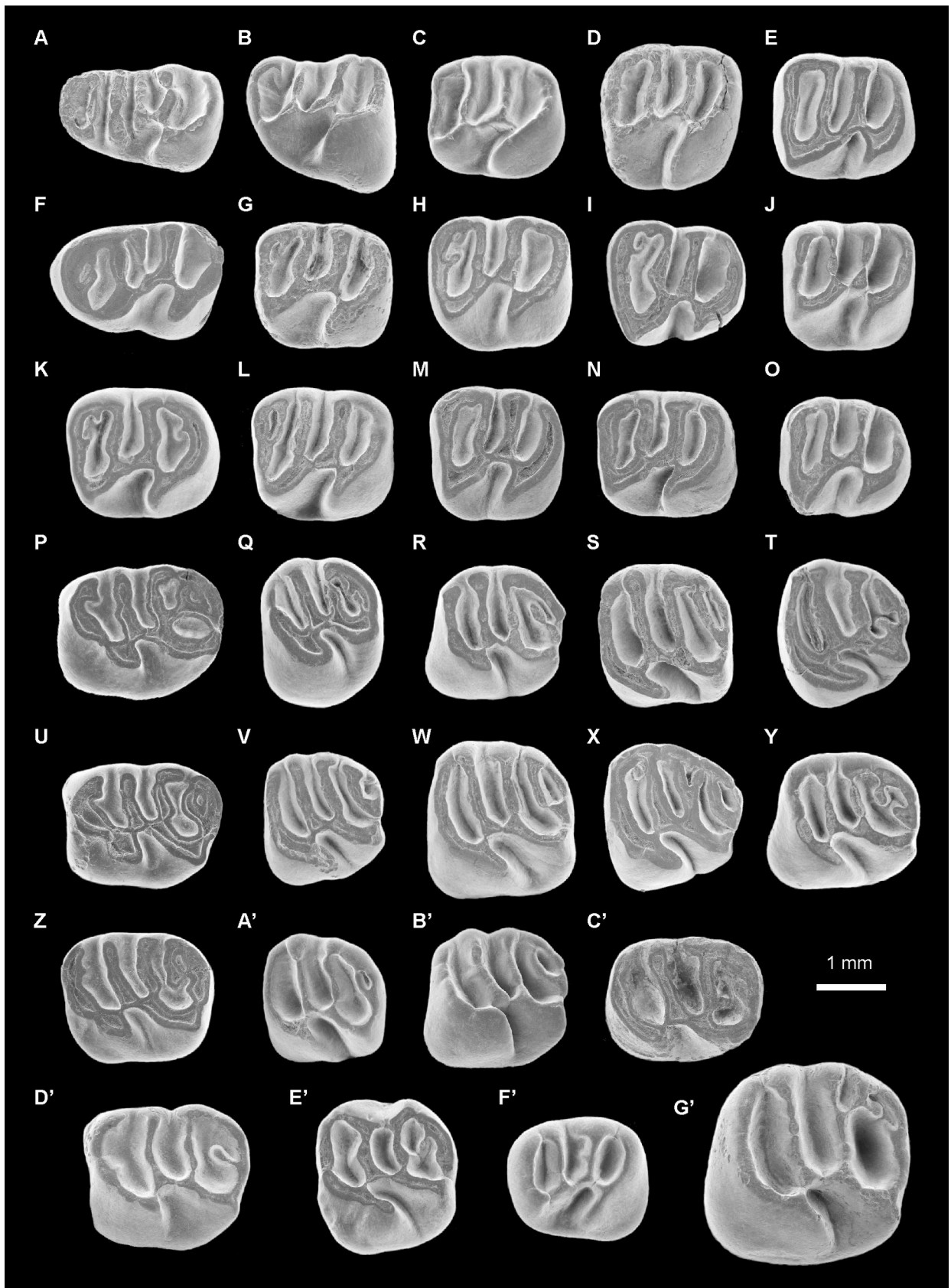

**Fig 5. Erethizontoid rodents from TAR-31, Peru (late middle Miocene, Laventan SALMA).** (A-E') *Nuyuyomys chinqaska* gen. et sp. nov. (F') Erethizontoidea gen. et sp. indet. 1. (G') Erethizontoidea gen. et sp. indet. 2. (A) MUSM 4332, right dp4 (reversed). (B) MUSM 4334, left p4. (C) MUSM 4347, right m1 (reversed). (D) MUSM 4359, right m2 (reversed). (E) MUSM 4366, right m3 (reversed). (F) MUSM 4329, left dp4. (G) MUSM 4348, right m1 (reversed). (H) MUSM 4355, left m2. (I) MUSM 4364, left m3. (J) MUSM 4356, left m2. (K) MUSM 4352, left m2. (L) MUSM 4362, right m2 (reversed). (M) MUSM 4363, right m2 (reversed). (N) MUSM 4360, right m2 (reversed). (O) MUSM 4365, right m3 (reversed). (P) MUSM 4290, right dP4 (reversed). (Q) MUSM 4295, left P4. (R) MUSM 4308***, right M1? (reversed). (S) MUSM 4319, left M2?. (T) MUSM 4324, right M3 (reversed). (U) MUSM 4292, right dP4 (reversed). (V) MUSM 4306, left M1?. (W) MUSM 4320, left M2?. (X) MUSM 4323, left M3. (Y) MUSM 4303, left M1?. (Z) MUSM 4285, left dP4. (A') MUSM 4309, right M1? (reversed). (B') MUSM 4314, right M1? (reversed). (C') MUSM 4293, right dP4 (reversed). (D') MUSM 4294, right dP4 (reversed). (E') MUSM 4298, M1 or 2?. (F') MUSM 4367, left p4. (G') MUSM 4368, right M1 or M2 (reversed). The asterisks appoint the holotype material. The illustrations of the fossil specimens are scanning electron photomicrographs.

## Tentitavely referred material

MUSM 1974 (Antoine et al., 2013: Fig 3M, p. 94), left P4; MUSM 1975 (Antoine et al., 2013: Fig 3N, p. 94), right upper molar; MUSM 4298 (Fig 5E'), left M1 or M2?.

## Derivation of name

From *chinqaska*, vanished in Quechua language, referring to the disappearance of both MD-67 and TAR-31 outcrops since their discovery, due to water erosion and paleontological exploitation, respectively.

## Type locality

TAR-31, Tarapoto/Juan Guerra, Mayo River, Shapaja road, San Martín Department, Western Amazonia, Peru.

## Other locality

MD-67, Pilcopata, Rinconadera and Madre de Dios rivers, Madre de Dios Department, Western Amazonia, Peru.

## Formation and age

Unnamed Fm., middle Miocene (i.e., Colloncuran-Laventan SALMA); lower member, Ipururo Fm., late middle Miocene (i.e., Laventan SALMA).

## Diagnosis

Rodent characterised by small size, brachydont lower teeth and dP4s, brachydont-mesodont P4s and upper molars, and a weak obliquity of loph(-id)s. dp4s, p4s and upper teeth with a non-taeniodont pattern. Lower molars with a non-taeniodont or pseudo-taeniodont pattern at early stages of wear, the former pattern being the most frequent. Elongated dp4s with a hexalophodont pattern. p4s are longer than wide, with a large trigonid and tetralophodont pattern. Lower molars with a tetralophodont or pentalophodont pattern, the latter corresponding to the addition of a neolophid between the metalophulid I and the second transverse cristid. dP4s with a pentalophodont or tetralophodont pattern, the former pattern being the most frequent. The metaloph (when present) can be lingually free or connected to the posteroloph or to both the third crest and posteroloph on dP4s. Round and tetralophodont P4s with an anterior arm of the protocone thicker and mesiolabially directed connected to an anteroloph lower, thinner and roughly transverse, and without endoloph. Pentalophodont upper molars with a metaloph always long and joining the metacone to the posteroloph on M2?s and complete or reduced on M1?s and M3?s. *Nuyuyomys* is markedly smaller than all erethizontoid species except

*Kichkasteiromys*, *Shapajamys*, *Palaeosteiromys*, *Microsteiromys* and *Noamys*. Differs from *Eosteiromys homogidens*, *Eosteiromys annectens*, *Hypsosteiromys*, *Microsteiromys* and *Coendou insidiosus* in having hexalophodont dp4s; from *Shapajamys*, *Protosteiromys*, *Eosteiromys* and *Microsteiromys* in having no neolophid on the anteroflexid on lower molars; from *Shapajamys*, *Branisamyopsis praesigmoides* and *Steiromys duplicatus* in having an ectolophid always complete. Differs from *Kichkasteiromys*, *Shapajamys* and *Palaeosteiromys* in having a metaloph disconnected labially to the metacone on some upper molars; from *Shapajamys* in having no distinct metaloph on P4s, metaloph more developed on ?M2s and an ectolophid always oblique on lower molars; from *Kichkasteiromys* in having a lingual protoloph less lingually situated and a posterior outgrowth of the protocone more oblique on upper molars; from *Palaeosteiromys* in having a mesoflexid more extended on m3s; from *Microsteiromys* in having more elongated dp4s with a narrower trigonid, the mesoflexid still lingually open at advanced stage of wear on m1–2s and less rectangular lower molars. Differs from *Hypsosteiromys* in having loph(-id)s more transverse, round P4s, pentalophodont M1–2s, and a complete metalophulid I on lower teeth; from *Noamys* in having pentalophodont and sligthly smaller dP4 and M1 without oval-shaped occlusal contour and flexi less extended. Differs from *Eosteiromys annectens* and *Steiromys duplicatus* in having non-taeniodont lower teeth; from *Neosteiromys* and *Parasteiromys* in having always a complete mure on dP4s and upper molars; from *Eopululo*, *Protosteiromys pattersoni* and *Neosteiromys* in having a P4 without endoloph; from *Eosteiromys homogidens* in having a mesial mesoflexus less mesiodistally extended on P4s; from *Eosteiromys annectens*, *Branisamyopsis* and *Neosteiromys* in having non-taeniodont dP4s and upper molars; from *Eopululo* and *Protosteiromys pattersoni* in having upper molars always with a metaloph (which is not reduced, far from being a relictual structure); from *Protosteiromys*, *Parasteiromys*, *Branisamyopsis*, *Eosteiromys*, *Steiromys* and *Neosteiromys* in having some elongated upper molars (potential M1s) and an incomplete metaloph on some upper molars; from *Protosteiromys medianus* in having a complete third crest on M3s.

## Description

dp4s are brachydont and much longer than wide, with a talonid slightly wider than the trigonid. They are non-taeniodont and hexalophodont. In addition to six complete transverse lophids, one dp4 (MUSM 4332; Fig 5A) has a tiny low cristulid stemming from the metalophulid I on the anteroflexid. Owing to its low elevation and shortness, it is interpreted here as a neolophid. The entoconid and hypoconid and their associated lophids (i.e., hypolophid [cristid 5 in Fig 4B1–4B9], anterior outgrowth of the hypoconid, anterior arm of the hypoconid, and posterior arm of hypoconid/posterolophid [cristid 6 in Fig 4B1–4B9]), distally positioned on the tooth, are well recognisable. The entoconid is the largest lingual cuspid. It is clearly isolated on almost all dp4s. On MUSM 4331, its isolation is less important because it is separated from the posterolophid by a thinner and shallower notch. The hypolophid is transverse or slightly arcuate (i.e., mesially concave). The crestiform hypoconid displays an anterior arm, which is mesiolingually oriented, thin and linked to the labiodistal extremity of the hypolophid. On MUSM 4332 (Fig 5A), the mesialmost part of the anterior arm of the hypoconid is lower than its distal part, but it is connected to the hypolophid and clearly separates the hypoflexid from the metaflexid (i.e., non-taeniodont pattern). The hypoconid exhibits a strong and short anterior outgrowth. The posterolophid is particularly long and curved, notably in its lingual part. On the less worn dp4s, it is composed of two (MUSM 4329; Fig 5F) or three (MUSM 4327 and 4332; Fig 5A) interconnected parts, a posterior arm of the hypoconid and posterolophid included. Mesially, the recognition of the main cuspids (i.e., protoconid, metaconid, and mesostylid) and cristids (i.e., posterior arm of the protoconid, metaconid cristid, mesolophid,

and neomesolophid) is ambiguous by the addition of some neostructures (i.e., neoconids and neolophids). Several hypotheses of homology can be proposed (Fig 4B10). The mesialmost cristid (cristid 1 in Fig 4B1–4B9) is curved. It (mainly) corresponds to the metalophulid I. It shows one or several cuspate thickenings in its central part and/or extremities (MUSM 4325, 4326, 4329, and 4330, 4332, 4333; Figs 4B1, 4B4–4B6, 4B8, 4B9 and 5A and 5F). They can correspond to the main cuspids (i.e., protoconid, metaconid, mesostylid) and/or neoconids (Fig 4B10). The second cristid (cristid 2 in Fig 4B1–4B9) is highly variable. It can be constituted of one or two linked cristulids, which can be associated to the thickenings of the metalophulid I. They can be connected to the central part of the cristid 1 (MUSM 4329 and 4333; Fig 4B4 and 4B9) and/or to one (MUSM 4325; Fig 4B5) or two of its extremities (MUSM 4326, 4330, and 4332; Fig 4B1, 4B6 and 4B8). The next transverse cristid (cristid 3 in Fig 4B1–4B9) is thick and long, stemming from a cuspate structure on the lingual margin of the tooth to connect to a large another cuspid-shaped structure on the labial margin. These cuspid-like structures are clearly isolated at early stages of wear (MUSM 4326, 4330, and 4332; Figs 4B1, 4B6, 4B8 and 5A) and can link the structures located mesially (metalophulid I and annexes) at more advanced staged of wear (MUSM 4329 and 4333; Figs 4B4, 4B9 and 5F). On MUSM 4326 and 4329, this transverse cristid has two parts: a lingual cristid connected to a labial one, equally sized (MUSM 4326; Fig 4B1) or shorter (MUSM 4329; Figs 4B4 and 5F). Owing to the important development of this cristid and of its cuspate structures on all dp4s, they are probably not neostructures (hypotheses 1 and 3 in Fig 4B10). The transverse cristid appearing in fourth position (cristid 4 in Fig 4B1–4B9) is labially connected to the hypolophid via the distal ectolophid on all dp4s. This cristid can be continuous (MUSM 4327, 4328, and 4330–4333; Fig 4B2, 4B3, 4B6–4B9) or divided into two parts, which are joined (4326 and 4329; Fig 4B1 and 4B4). In addition, it can be high (MUSM 4326 and 4330–4333; Fig 4B1, 4B6–4B9), entirely low (MUSM 4327; Fig 4B2) or only its labial part can be low (MUSM 4329; Figs 4B4 and 5F). On MUSM 4327 (Fig 4B2), it is slightly connected to a bulbous cuspid clearly distinct on the lingual margin of the tooth.

MUSM 4334 is an almost pristine p4 (Fig 5B). It is longer than wide, with a trigonid narrower than the talonid. It is brachydont and tetralophodont. On its mesiolingual corner, the crestiform metaconid is convex forward. Distally, this cuspid displays a very short posterior arm. There is a tiny spur-like cristulid on the anteroflexid, labially free and stemming from the distal extremity of the posterior arm of the metaconid. Mesially, the metaconid has a transverse and long anterior arm. From its central part, a short cristulid runs distally on the anteroflexid. The anterior arm of the metaconid is linked to a faintly curved anterior arm of the protoconid. The latter cristid is clearly sloped with a high distal part stemming from the triangular-shaped protoconid. The second transverse cristid is essentially composed of a long neomesolophid joining a very short posterior arm of the protoconid. The ectolophid is mainly oblique and S-shaped. It reaches the distal slope of the protoconid. The entoconid is high and it has a sharp posterior arm. Labially, this cuspid is connected to a hypolophid, strong and distolingually oblique. On the labiodistal margin of the ectolophid-hypolophid junction, a tiny cristulid is present. Its top is separate from the anterior arm of the hypoconid by a very shallow notch. However, the contact between these structures remains high despite the presence of this notch (i.e., non-taeniodont pattern). The anterior outgrowth of the hypoconid is short and directed mesiolabially. The posterolophid is transverse with curved lingual and labial extremities. On the lingual margin, all structures (i.e., metaconid, mesostylid, entoconid and lingual extremity of the posterolophid) are merged together at their base, but they remain disconnected at their apex. The mesostylid is discreet and isolated from the posterior arm of the metaconid and the entoconid by wide and deep furrows. The notch separating the entoconid from the lingual extremity of the posterolophid is thinner and shallower. The anteroflexid and metaflexid are

more expanded than the mesoflexid. The hypoflexid is triangular-shaped with a wider labial openning.

The lower molars are brachydont and rectangular-shaped. m3s (MUSM 4364–4366; Fig 5E, 5I and 5O) show a talonid slightly narrower than the trigonid, notably lingually. m1s and m2s usually have a trigonid and a talonid roughly of same width and m2s are larger than m1s and m3s. On all lower molars, the metaconid, protoconid and hypoconid are crestiform, whereas the mesostylid and entoconid are more bulbous. The metalophulid I is transverse and complete, joining the metaconid to the protoconid. On the lower molars at the earliest stage of wear (MUSM 4342, 4347, and 4359; Fig 5C and 5D), this cristid corresponds to a long anterior arm of the metaconid, whose lingual extremity is low and linked a short anterior arm of the protoconid. On most lower molars, the mesostylid is linked to the posterior arm of the metaconid, this connection can be slight or tight. These two structures can be separated by a wide furrow (e.g., MUSM 4336, 4357, and 4365; Fig 5O) or a thin notch (e.g., MUSM 4356, 4347, and 4348; Fig 5C, 5G and 5J). Several lower molars exhibit two short cristulids on their mesiolingual area: the first cristulid is situated on the distolingual part of the metalophulid I, and the second is located distally to the metaconid, on its posterior arm (MUSM 4344, 4352, 4355, 4364, and 4362; Fig 5H, 5I, 5K and 5L). On MUSM 4344, 4364 and 4362 (Fig 5I and 5L), these two cristulids are joined, and form a mesial neolophid on the anteroflexid (i.e., pentalophodont pattern). On MUSM 4352 and 4355, they remain separate (Fig 5H and 5K). On MUSM 4351, there is only the cristulid of the metalophulid I, whereas on MUSM 4348 (Fig 5G), there is only the cristulid distally to the metaconid. The second transverse cristid is a long neomesolophid directly connected to the protoconid or via its very short posterior arm. It can be slightly oblique, transverse, or sometimes sinuous (S-shaped). The ectolophid is always oblique, either aligned with the protoconid or a little more longitudinal. The entoconid can display strong anterior and posterior arms. On MUSM 4336 and 4351, the posterior arm is particularly long. The latter can be connected (e.g., MUSM 4336, 4352, 4364, and 4359; Fig 5D, 5I and 5K) or disconnected (e.g., MUSM 4351, 4354, 4356, 4357, 4347, and 4365; Fig 5C, 5J and 5O) to the posterolophid. The hypolophid is well marked and often parallel to the second transverse cristid. The anterior arm of the hypoconid is complete and high in most cases (i.e., nontaeniodont pattern). Two lower molars, MUSM 4356 and 4347 (Fig 5C and 5J), show a pseudo-taeniodont pattern. On MUSM 4356 (Fig 5J), the anterior arm of the hypoconid is complete, but moderately low. On MUSM 4347 (Fig 5C), there is a very low cristulid stemming from the hypolophid-ectolophid junction and joining the mesial slope of the hypoconid, thereby involving a partial confluence between the metaflexid and the hypoflexid.

dP4s are longer than wide with a rectangular or slightly trapezoidal occlusal outline. On all dP4s (except on MUSM 4293; Fig 5C'; see below), the protocone is somewhat bulbous, a morphology accentuated by a thinning at the level of its connection with the anteroloph. In addition, the lingual extremity of the latter is sometimes thickened (MUSM 4284, 4287, and 4289–4292; Fig 5P and 5U). The anteroloph is mesially convex, and can have an enamel thickening on its labial part, that we interpret as a parastyle, associated with a spur-like crestule (MUSM 4285, 4287, 4289, 4290, and 4292; Fig 5P, 5U and 5Z). This crestule remains lingually free on the paraflexus. On the labial margin of dP4s, the paracone is the largest cusp. It is clearly separated mesially from the labial extremity of the anteroloph and distally from the mesostyle by wide and deep furrows in most cases. On MUSM 4290 (Fig 5P), the paracone is separate from the anteroloph by a shallower notch. The labial protoloph can be transverse (MUSM 4291), oblique (i.e., mesiolabially directed; MUSM 4293; Fig 5C') or with an oblique lingual part and a labial part more transverse (on the other dP4s). On MUSM 4287, in its middle, the labial protoloph displays a tiny crestule on its mesial part. Its ends close to the distal extremity of the crestule stemming from the parastyle, but does not connect to it. The mure is oblique (e.g.,

MUSM 4290 and 4293; Fig 5P and 5C') or more longitudinal (i.e., parallel to the mesiodistal long axis of the tooth; e.g., MUSM 4291 and 4292; Fig 5U). Its junction with the labial proto-loph is linked to the protocone via its thin anterior arm (i.e., non-taeniodont pattern). Labially, the mesostyle is large, whereas the metacone is faintly visible from the posteroloph. On MUSM 4290–4292 and 4294 (Fig 5P, 5U and 5D'), the mesostyle shows a long posterior arm. The mesostyle or its posterior arm is either linked to the metacone-posteroloph complex (MUSM 4287, 4290, 4291, and 4293; Fig 5P and 5C') or is separated from this cusp by a thin and shallow notch (MUSM 4285, 4286, 4292, and 4294; Fig 5U, 5Z and 5D'). The third crest is complete and S-shaped. On MUSM 4285 and 4292 (Fig 5U and 5Z), this crest consists of a long mesoloph stemming from the mesostyle and joining lingually the mure-anterior arm of the hypocone junction directly (MUSM 4292; Fig 5U) or via a short mesolophule (MUSM 4285; Fig 5Z). The posteroloph is mesially concave and links the metacone to the hypocone. The metaloph shows a variable length, direction and lingual connection. On MUSM 4285, 4290 and 4292 (Fig 5P, 5U and 5Z), the metaloph is long, turned backwardly (either curved or L-shaped), and linked to the posteroloph via (or not) a posteroloph spur. On MUSM 4286, the metaloph is short, mesiolingually directed and lingually free. On that tooth, there is also a cres-tule, close to the metaloph and connected lingually to the posteroloph spur and labially to the posteroloph. Due to its position and configuration, it is probably a rest of metaloph. On some worn dP4s (MUSM 4287, 4291, and 4293; Fig 5C'), the metacone and metaloph are likely sub-sumed within the posteroloph (+ its spur), thereby forming a thick platform distally. Neocres-tules connect the third transverse crest to the metacone or to the distal platform on MUSM 4289–4291 (Fig 5P). On MUSM 4294 (Fig 5D'), there is no metaloph or distinct metacone (which is probably fused with the posteroloph). On that tooth, the thin posteroloph bears a long spur, which runs mesiolabially on the posterior flexus and remains free. The hypoflexus forms a U-shaped structure, mesiolabially directed.

P4s are round, MUSM 4295 shows an oval outline (Fig 5Q), whereas MUSM 1974 is longer and more circular-shaped (Fig 3M [31]). MUSM 4295 is higher-crowned with respect of the other teeth attributed to *Nuyuyomys chinqaska* (Hling = 0.89; Hlab = 0.62; Fig 5Q). On MUSM 4295, the area of the protocone and that of the hypocone are large and conical, corresponding to oval regions of dentine delimited by a very thick exterior enamel layer (Fig 5Q). On the two P4s, the mesialmost crest is divided into two parts, which are connected together. Its lingual part, the anterior arm of the protocone, is thicker, mesiolabially directed, longer on MUSM 4295 (Fig 5Q), but shorter on MUSM 1974 (Fig 3M [31]). This crest is in the continuity with the protocone and its short posterior outgrowth. The labial part of the mesialmost crest, the anteroloph, is thinner, roughly transverse, shorter on MUSM 4295, but longer on MUSM 1974. It should be lower than the anterior arm of the protocone (and the other occlusal struc-tures) at early stage of wear, because it is barely worn contrary to all other structures of the occlusal surface. The anteroloph is separate from a strong paracone (or the mesial extremity of its sloped anterior arm on MUSM 4295) by a thin furrow. The labial protoloph is oblique. It is connected, with the mure, to a strong and complete posterior arm of the protocone. On MUSM 4295 (Fig 5Q), the mure is very short and longitudinal (i.e., parallel to the mesiodistal axis of the tooth). On MUSM 1974 (Fig 3M [31]), the mure is absent or reduced to a very tiny crestule from the third crest-anterior arm of the hypocone jonction that remains mesially free. The third crest is almost parallel to the oblique labial protoloph, and is composed of a long mesolophule sligthly linked to a short mesoloph. The mesostyle is strongly connected to a curved and thick posteroloph. There is no clear distinct metacone or metaloph. Hence, the posterior flexus is fully closed. However, the posteroloph appears to have a thickenning, which could correspond to a relictual metaloph, on MUSM 4295 (Fig 5Q). The latter and paraflexus are long, narrow mesiolabially-distolingually extended and oriented on MUSM 4295 (Fig 5Q),

whereas they are more mesiodistally expanded on MUSM 1974 (Fig 3M [31]), due to the higher length of that tooth. The mesial mesoflexus and hypoflexus are clearly open.

M3s of *Nuyuyomys chinqaska* (MUSM 4323 and 4324; Fig 5T and 5X) are well recognisable by their distal part, which is labiolingually pinched (notably labially), and their hypocone, being clearly smaller and with a lingual margin occupying a more labial position than the protocone. Among other upper molars attributed to this taxon, some are small-sized and subsquare or slightly mesiodistally elongated, and other larger can be also subsquare but they are usually more transverse, being wider than long. We interpret them as possibly M1s and M2s, respectively (i.e., M1? and M2?). The height of crown is widely variable. Some upper molars at early stages of wear have a higher crown (e.g., the MUSM 4303 M1? and 4320 M2?; Hling = 0.74–0.89) than other, which are less worn (e.g., the MUSM 4313 and 4314 M2?s; Hling = 0.60–0.69). Because there are teeth with intermediary crown height between these two extremes and no morphological character allowing to differentiate them, we consider that all the material reported here documents a single taxon. On all upper molars, the crests are thin. The paracone and the mesostyle are well differentiated, whereas the protocone and hypocone are more crestiform, as the metacone in most cases. The latter is more cuspate on two upper molars (MUSM 4309 and 4320; Fig 5W and 5A'). The protocone and hypocone are mesiolingually-distolabially and mesiodistally compressed, respectively. As on the MUSM 4295 P4, the anterior arm of the protocone is mesiolabially directed and in continuity with the protocone, its posterior outgrowth and sometimes a part of the anteroloph, like on MUSM 4314 (Fig 5B'). The anteroloph (or its labial part) is more transverse. At early stages of wear, labially the paraflexus can be either closed (MUSM 4298 and 4303; Fig 5Y and 5E') or opened (e.g., MUSM 1975, 4306, 4308, 4309, 4311, and 4320; Fig 3N [31]; Fig 5R, 5V, 5W and 5A'), separate from the paracone by a thin notch (e.g., MUSM 4308, 4309, and 4320; Fig 5R, 5W and 5A') or a wider furrow (MUSM 1975, 4306 and 4311; Fig 3N [31]; Fig 5V). The posterior outgrowth of the protocone can be very short, particularly on M1?s, or moderately longer. The posterior arm of the protocone is complete and thin. On MUSM 4303, 4307 and 4314 (Fig 5Y and 5B'), this crest is slightly lower than the protocone, but remains high (i.e., non-taeniodont pattern). Although the paracone and mesostyle are close in position, they do not share connection in most cases. On MUSM 4298, 4320 and 4323 (Fig 5W, 5X and 5E'), however they are linked via their arms (i.e., posterior arm of the paracone and anterior arm of the mesostyle). The labial protoloph is usually straight and mesiolabially directed, but it can have a lingual part oblique, whereas its labial part is more transverse such as on MUSM 4298, 4318 and 4322 (Fig 5E'). Two upper molars (MUSM 4311 and 4317) have one crestule on the mesial margin of the labial protoloph, which is mesially free. MUSM 4323 does not have that crestule, but it shows a tiny enamel swelling on the labial part of the paraflexus (Fig 5X). The third crest is often parallel to the labial protoloph, and is usually composed of a long mesoloph linked to a mesolophular spur. The mure is short but complete, oblique and linked to a strong and long anterior arm of the hypocone. As dP4s, some upper molars have a long posterior arm of the mesostyle (MUSM 4303, 4307, 4319, and 4323; Fig 5S, 5X and 5Y). On other upper molars, this arm is usually short, but it can also be undeveloped (MUSM 1975 and 4314; Fig 3N [31]; Fig 5B'). The metacone can be connected to the mesostyle or its posterior arm (e.g., MUSM 4303, 4308, and 4319; Fig 3N [31]; Fig 5R, 5S and 5Y), or separated from these structures by a very tiny notch (e.g., MUSM 4306, 4307 and 4320; Fig 5V and 5W), except on MUSM 1975 and 4314, where the notch is thicker (Fig 3N [31] and Fig 5B'). On all upper molars, a metaloph is present, either long or short. It is often connected lingually to the posteroloph (via a posteroloph spur on MUSM 4315 and 4320; Fig 5W), but it can also be lingually free (MUSM 4298, 4303, and 4312; Fig 5Y and 5E'). On MUSM 1975, 4308 and 4307 (Fig 3N [31] and Fig 5R), it is reduced to a crestule labially disconnected to the metacone, and lingually linked to the

posteroloph spur. Specifically, the metaloph is always long and joining the metacone to the posteroloph on teeth identified as M2?s, whereas it is complete or reduced on M1?s and M3?s. The MUSM 4323 M3 displays a neocrest stemming from the anterior arm of the metacone, and connecting lingually to the third crest (Fig 5X). The hypocone has a short posterior arm slightly connected to the posteroloph (MUSM 4308, 4309, 4311, 4315, 4318, 4319, and 4324; Fig 5R, 5S, 5T and 5A') or separate from it by a minute and shallow notch (MUSM 4298, 4303, 4306, 4313, 4320, and 4323; Fig 5V, 5W, 5X, 5Y and 5E'). This notch is wider on MUSM 4314 (Fig 5B'). Some upper molars have a mesiodistal constriction at the base of the hypoflexus (MUSM 4303, 4307, 4312, 4319, and 4324; Fig 5S, 5T and 5Y).

## Remark 1

The MUSM 4293 and 4294 (Fig 5C' and 5D') dP4s have size, a crown height and global occlusal morphology similar than the rest of dP4s attributed to *Nuyuyomys chinqaska* (i.e., rectangular or slightly trapezoidal occlusal outline, pentalophodonty-tetralophodonty, non-taeniodonty, and U-shaped hypoflexus). Nevertheless, they differ from other dP4s attributed to this taxon in some characters. MUSM 4293 (Fig 5C') has a protocone strongly linked with the anteroloph without any thinning of this crest. On MUSM 4294 (Fig 5D'), the protocone is less bulbous than on other dP4s of *N. chinqaska*, and the anteroloph shows a more sligthly thinning. Finally, there is no distinct metacone and metaloph on MUSM 4294 (Fig 5D'). We interpret all these differences between MUSM 4293, 4294, and the rest of dP4s attributed to *N. chinqaska* as documenting intra-specific variation.

## Remark 2

We tentatively assigned the MUSM 4298 upper molar (Fig 5E') to *Nuyuyomys chinqaska* because it shows a particular pattern, not really shared with other upper molars attributed to *N. chinqaska*. On that tooth, the apices of the paracone and mesostyle appear to be lingually displaced and they bear very long arms with unusual orientations due to a kind of unusual invagination of the labial crown margin: the anterior arm of the paracone is mesiolabially directed, its posterior arm is very slightly distolabially directed, and the anterior arm of the mesostyle is mesiolingually directed. Owing to the orientation of these arms and the connections anteroloph-anterior arm of the paracone, posterior arm of the paracone-anterior arm of the mesostyle, and mesostyle-metacone, the dental structures of the tooth labially form a "M". In addition, the extremity of the posterior outgrowth of the protocone is bulged appearing as a cusp-like structure. Nevertheless, MUSM 4298 matches with the morphology of the other upper molars of *N. chinqaska* by other characters (i.e., size, crown-height, pentalophodonty, non-taeniodonty, crestules on the paraflexus, oblique mure, long mesoloph connected to a mesolophular spur, and tiny and shallow furrow separating the posterior arm of the hypocone from the posteroloph). As far we know, such M-shaped pattern for the labial dental structures on upper molars is not found in caviomorphs or very rare. Consequently, the peculiar characters of MUSM 4298 may result to some abnomalies or extreme intra-specific variation of this taxon.

## Remark 3

In addition, the specimens of TAR-31 have a small size and general pattern (i.e., non-taeniodonty; complete mure on upper teeth; presence of a metaloph on dP4s and upper molars) similar to those of the material from the early middle Miocene of Madre de Dios formerly attributed to cf. *Microsteiromys* sp. (MD-67, Peru) [31]. The latter is documented by only two upper teeth: MUSM 1974 P4 and MUSM 1975 upper molar (Fig 3M and 3N, p. 94 [31]). By

their mesiodistally-elongated shape, ?M1s from TAR-31 are close to MUSM 1975. Particularly, MUSM 4307 and 4308 (Fig 5R) from TAR-31 show a labial protoloph and a third crest slightly oblique and a short metaloph labially unconnected to the metacone and lingually linked to the posteroloph spur, as on MUSM 1975. Due to the absence of lower teeth attributed to the taxon from MD-67 and differences on P4 (i.e., longer; absence or strong reduction of the mure; flexus shape), we prefer to tentatively refer this material (MUSM 1974 and 1975) to as *Nuyuyomys chinqaska* for the time being.

## Comparisons

By its non-taeniodonty, extended flexi(-ds), weak obliquity of loph(-id)s, p4 with a large trigonid, rounded P4, pentalophodont upper molars, and a well-defined, and slightly oblique (faintly backwardly oriented) metaloph on upper molars, the material from TAR-31 exhibits strong erethizontoid affinities. Moreover, like on some lower molars from TAR-31, a subpentalophodont pattern (i.e., presence of a short neolophid in the anteroflexid) is otherwise found in some erethizontoids (e.g., *Palaeosteiromys amazonensis*, *Branisamyopsis australis*, *Branisamyopsis praesigmoides*, *Steiromys duplicatus* and *Neosteiromys pattoni*) [71, 74–77]. The material from TAR-31 shows brachydont lower teeth and submesodont-mesodont upper teeth, a rather rare crown condition found among the erethizontoids, which are usually brachydont. The only extinct erethizontoids so far showing high tooth crowns are *Hypsosteiromys* (Sarmiento Fm., Argentina; early Miocene) [74, 78, 83] and *Noamys* (site 3 of Río Chico Locality, Guanaco Fm., Argentina; late Miocene and/or early Pliocene) [72]. However, *Nuyuyomys* is clearly distinct from the latter taxa. *Hypsosteiromys* has a larger size, loph(-id)s more oblique, square P4s, tetralophodont M1–2s, tetralophodont-pentalophodont dp4s, and a metalophulid I reduced to an anterior arm of the metaconid disconnected to the protoconid on lower teeth. *Noamys* has tetralophodont and sligthly longer dP4 and M1 with oval-shaped occlusal contour and flexi more extended, especially anteroposteriorly. The specimens from TAR-31 are markedly smaller than all erethizontoid species except *Kichkasteiromys raimondii*, *Shapajamys labocensis*, *Palaeosteiromys amazonensis*, *Microsteiromys jacobsi* gen. et sp. nov., and *Noamys* [18, 37, 70–72] (this work). Contrary to *Shapajamys*, the P4s from TAR-31 and MD-67 (MUSM 4295 and MUSM 1974) do not have any distinct metaloph. Unlike the material from MD-67 and TAR-31, *Kichkasteiromys*, *Shapajamys* and *Palaeosteiromys* do not display a metaloph disconnected labially to the metacone on upper molars. One M1 of *Shapajamys* (MUSM 2993) has a reduced metaloph that is lingually free, as on some M1?s from TAR-31. However, ?M2s from TAR-31 have a metaloph more developed than in *Shapajamys*. Upper molars from TAR-31 differ from the MUSM 2925 M1 or M2 of *Kichkasteiromys* (Fig 2D, p. 92 [70]) in having a lingual protoloph less lingually situated and a posterior outgrowth of the protocone more oblique. For lower teeth, like on the material from TAR-31, some erethizontoids (*Shapajamys*, *Branisamyopsis*, ?*Eosteiromys* sp. nov. *sensu* Candela [69], and *Steiromys*) show hexalophodont dp4s with accessory structures. By its elongated shape and tetralophodont pattern, the MUSM 4334 p4 is particularly close to the p4 of *M. jacobsi* (IGM-DU 89–249 in Fig 16C, p. 56 [37]; Fig 3B), whereas dp4s from TAR-31 are much more elongated, and with a narrower trigonid than in *M. jacobsi* (FMNH PM 54672 in Fig 24.2B, p. 395 [18]; Fig 3A). In addition, dp4s from TAR-31 are hexalophodont, while the FMNH PM 54672 dp4 appears tetralophodont. The lower molars of *M. jacobsi* (FMNH PM 54672) and of *Shapajamys* do not have any neolophid on their anteroflexid, contrary to several specimens from TAR-31 and *Palaeosteiromys*. The material from TAR-31 differs from *M. jacobsi* in having the mesoflexid still lingually open at advanced stage of wear on m1–2s and less rectangular lower molars; and from *Shapajamys* in having an ectolophid always complete and oblique on lower molars. Compared to the MUSM

2861 m3 of *Palaeosteiromys* (Fig 4L, p. 77 [71]), m3s from TAR-31 have no neolophid or have a shorter one, and show a mesoflexid more extended. The material from TAR-31 differs from *Eosteiromys annectens* and *Steiromys duplicatus* in having non-taeniodont lower teeth; *Eosteiromys homogidens*, *Eosteiromys annectens*, and *Coendou insidiosus* in having hexalophodont dp4s; from *Protosteiromys* and *Eosteiromys* in having no neolophid on the anteroflexid on lower molars; from *Branisamyopsis praesigmoides* and *Steiromys duplicatus* in having an ecto-lophid always complete; from *Neosteiromys* and *Parasteiromys* in always having a complete mure on dP4s and upper molars; from *Eopululo*, *Protosteiromys pattersoni* and *Neosteiromys* in having a P4 without endoloph; from *Eosteiromys homogidens* in having a mesial mesoflexus less mesiodistally extended on P4s; from *Eosteiromys annectens*, *Eosteiromys kramarzi sensu* Candela [74], *Branisamyopsis* and *Neosteiromys* in having non-taeniodont dP4s and upper molars; from *Eopululo*, *Protosteiromys pattersoni* and the three Erethizonthoidea indet. from Acre region (Brazil; Late Miocene) [84, 85] in having upper molars always with a metaloph (which is not reduced, far from being a relictual structure); from *Protosteiromys*, *Parasteir-omys*, *Branisamyopsis*, *Eosteiromys*, *Steiromys* and *Neosteiromys* in having some elongated upper molars (potential M1s) and an incomplete metaloph on some upper molars; from *Brani-samyopsis pardinas sensu* Candela [74] in having a complete third crest on M1–2s; from *Proto-steiromys medianus* in having a complete third crest on M3s [74–77, 80–83]. To summarise, the erethizontoid teeth from TAR-31 are morphologically close to those of *Kichkasteiromys*, *Shapajamys laboensis*, *Palaeosteiromys*, and *M. jacobsi*. In having elongated dp4s, with a hexa-lophodont pattern, the material from TAR-31 clearly differs from that of *M. jacobsi*. Hence, we consider the dental material from TAR-31 as documenting a new genus and species: *Nuyuy-omys chinqaska*.

Erethizontoidea gen. et sp. indet. 1

Fig 5F' and S1 Table

### Referred material

MUSM 4367, left p4 (Fig 5F').

### Locality

TAR-31, Tarapoto/Juan Guerra, Mayo River, Shapaja road, San Martín Department, Western Amazonia, Peru.

### Formation and age

Lower member, Ipururo Fm., late middle Miocene (i.e., Laventan SALMA).

### Description

MUSM 4367 is a pristine p4, with rounded corners and trigonid slightly narrower than the talonid (Fig 5F'). It is brachydont, tetralophodont and non-taeniodont. The entoconid is the highest and largest cuspid. The metaconid and hypoconid are mesiodistally compressed, whereas the mesostylid is more bulbous. On the mesiolabial corner of MUSM 4367, the meta-lophulid I is connected to a cuspate structure with a size similar to the mesostylid, that we interpret as the protoconid due to its position and connection with the metalophulid I. The lat-ter is complete and continuous. It has a transverse and long lingual part connecting to the poorly distinct metaconid, whereas its labial one is shorter and curved. On its distal slope, the metaconid has a short and quite transverse posterior arm, which is disconnected to the

mesostylid. The second transverse cristid is straight and well-marked. It appears to be formed by a long neomesolophid, connecting the mesostylid to a shorter and sloping posterior arm of the protoconid. The latter is labially linked to a labial cuspid-shaped structure. This structure is elongated, mesiolabially directed. It is aligned with a cristid stemming from the hypolophid-anterior arm of the hypoconid junction (= the ectolophid or a part of the ectolophid), but remains separate from it, as well as from the protoconid mesiolabially, by notches. This elongated cuspid-shaped structure could correspond to a part of the ectolophid fused with the protoconid cristid (see [4]). The entoconid is somewhat bulbous and displays thick, but faintly defined, arms. Its anterior arm is clearly separate from the mesostylid, whereas its posterior arm almost connects to the lingual extremity of the posterolophid, a very thin and shallow notch separating both cristids. The hypolophid is slightly oblique and delimits, with the entoconid, the curved posterolophid and the strong and high anterior arm of the hypoconid, a well-extended and oval metafossettid. The anteroflexid and mesoflexid are rectangular-oval and have a similar extension, smaller than that of the metafossettid. The U-shaped hypoflexid with a distolingual direction is linguolabially well-expanded.

## Comparisons

MUSM 4367 is brachydont with a large trigonid, non-taeniodont pattern, extended flexi(-ds), and a weak obliquity of loph(-id)s, which are characters found in erethizontoids. In addition to its size, which is similar to that of p4s of *Microsteiromys jacobsi* (IGM-DU 89–249) and *Nuyuyomys chinqaska* (MUSM 4334), MUSM 4367 has a similar morphology (i.e., brachydonty, tetralophodonty, non-taeniodonty, and oblique ectolophid). However, contrary to p4s of *M. jacobsi* and *N. chinqaska*, as well as the MNHN 1903-3-18 premolar of *Protosteiromys asmodeophilus* (Fig 32 [79] and see above), MUSM 4367 is more round shaped and less elongated, with a shorter trigonid and a straight mesial crown margin. In particular, MUSM 4367 differs from MUSM 4334 of *N. chinqaska* in having a wider trigonid comparing to the width of the talonid, a higher labial part of the metalophulid I, and a protoconid more mesially canted; from MNHN 1903-3-18 of *P. asmodeophilus* in having smaller lingual flexid/fossettids and a larger hypoflexid. Unlike p4s of *Hypsosteiromys*, MUSM 4367 has more straight lophids and a complete metalophulid I connected to the protoconid [74, 83]. The MLP 93-XI-18-1 p4 of *Eosteiromys homogenidens* [74] appears to have a similar mesiolabial morphology to MUSM 4367 in having a second cristid labially connected to an elongated and oblique structure, which may correspond to a part of the ectolophid + protoconid cristid, disconnected to the rest of the ectolophid. This structure is slightly connected to the protoconid on MLP 93-XI-18-1, contrary to MUSM 4367, but it was probably disconnected to this cuspid at an earlier stage of wear. MUSM 4367 differs from p4s of *E. homogenidens*, *Eosteiromys annectens* [77], *Branisamyopsis australis* [74, 75], *Branisamyopsis praesigmoides* [77], and *Steiromys detentus* [74] in being more square-shaped; of *E. homogenidens* in having a straight metalophulid I and less extended anteroflexid; of *E. annectens* in having a non-taeniodont pattern and a less oblique hypolophid and posterolophid; of *Eosteiromys segregatus* [74] in having no posterior arm of the mesostylid and a more extended metaflexid; of *Branisamyopsis*, *Steiromys duplicatus* [74], *Steiromys verzii sensu* Candela [74], and some p4s of *S. detentus* in having a tetralophodont pattern; of *S. detentus* in having a less extended anteroflexid and more oblique ectolophid; and of *S. duplicatus* in having a straight metalophulid I. MUSM 4367 is smaller than *Hypsosteiromys*, *Branisamyopsis*, *Eosteiromys* and *Steiromys*. Given the paucity of the material, we provisionally assign MUSM 4367 to Erethizontoidea gen. et sp. indet. 1.

Erethizontoidea gen. et sp. indet. 2

Fig 5G' and S1 Table

## Referred material

MUSM 4368, left M1 or M2 (Fig 5G').

## Locality

TAR-31, Tarapoto/Juan Guerra, Mayo River, Shapaja road, San Martín Department, Western Amazonia, Peru.

## Formation and age

Lower member, Ipururo Fm., late middle Miocene (i.e., Laventan SALMA).

## Description

The unique M1 or 2 (MUSM 4368; Fig 5G') documenting this taxon is large, brachydont, and non-taeniodont. It has a subsquare to rounded occlusal outline. The main cusps/styles (paracone, mesostyle, protocone, and hypocone) are large and well defined. The labial cusps/styles are also well defined, whereas the lingual ones are somewhat compressed labiodistally-linguomesially, especially the protocone, which has a strongly crestiform aspect. The worn part of the posterior outgrowth of the protocone is very short at this early stage of wear of the tooth. The anteroloph is long and curved. It is lingually connected to the protocone and labially separated from the paracone by a thin notch. The lingual protoloph, stemming from the protocone, is high and reaches the labial protoloph–mure junction (i.e., non-taeniodont pattern). The labial protoloph and the third crest are faintly oblique. The mure, linking these two transverse crests lingually, is longitudinal (i.e., on the mesiodistal axis of the tooth). Labially, a large furrow separates the mesostyle from the paracone. The metacone is visible because it does not display tight connections with the other labial dental structures. A thin and shallow groove separates it from the mesostyle, and a very slight connection is present between the tiny posterior arm of the metacone and the posteroloph. The third transverse crest is complete, but labially thinner and shorter, thereby indicating that it could be formed primarily by a long mesolophule and a short labial mesoloph. The metacone displays a straight and very short metaloph. At the level of this lingual part, the metaloph exhibits: (i) a very short crestule mesially, which connects to the labial extremity of the mesolophule via a longer crestule; and (ii) an alignement of crestules distally, which remain free. This connection between the metaloph and the third crest generates a small and narrow flexus labially located. The hypocone has mesially a long anterior arm, which is strongly linked to the mure-third crest junction, and distally a posterior arm slightly curved and connected to the straight posteroloph. In addition, a third short crest runs distolingually stemming from the hypocone, and can be recognised as a posterior outgrowth of this cusp. In its middle part, the posteroloph bears a spur, labially directed towards the distal crestules of the metaloph but without connection with them. All the main flexi are well extended and deep.

## Comparisons

Among the four superfamilies of caviomorphs, MUSM 4368 exhibits strong erethizontoid affinities in being brachydont, non-taeniodont, pentalophodont, and in showing extended flexi and straight lophs. Erethizontoid molars have usually a better-defined metaloph than MUSM 4368, but some species can also be characterised by a reduced metaloph (e.g., *Protosteiromys pattersoni*, *Eosteiromys kramarzi*, *Parasteiromys uniformis*, and *Steiromys detentus*) or the absence of this crest (e.g., *Eopululo wigmorei*, *P. pattersoni*, *Hypsosteiromys*, *Eosteiromys segregatus*, *Steiromys detentus*, and *Noamys hypsodonta* [72, 74, 78, 80–83, 86]. However, these species with a reduced metaloph do not show a configuration similar to that of MUSM 4368, with crestules on the

lingual extremity of the metaloph, some of which are connected to the third transverse crest. MUSM 4368 is smaller than the upper molars of *P. pattersoni*, *Hypsosteiromys*, *P. uniformis*, *E. kramarzi*, *E. segregatus*, and *Steiromys*. Moreover, MUSM 4368 differs from the upper molar attributed to *E. wigmorei* (LACM 143269 in p. 111 [81]) in having a more quadrate shape; from upper molars of *Hypsosteiromys* in having lower crown and lophs less oblique; of *P. uniformis* in having a complete mure; and of *E. kramarzi* in having a lingual protoloph. By its size larger than MUSM 4367 and the material attributed to *N. chinqaska*, MUSM 4368 probably documents another erethizontoid at TAR-31, provisionally identified as Erethizontoidea gen. et sp. indet. 2.

Cavioidea Fischer, 1817

Caviidae Fischer, 1817

Caviidae gen. et sp. indet.

Fig 6 and S1 Table

## Referred material

MUSM 4283, right p4 (Fig 6).

## Locality

TAR-31, Tarapoto/Juan Guerra, Mayo River, Shapaja road, San Martín Department, Western Amazonia, Peru.

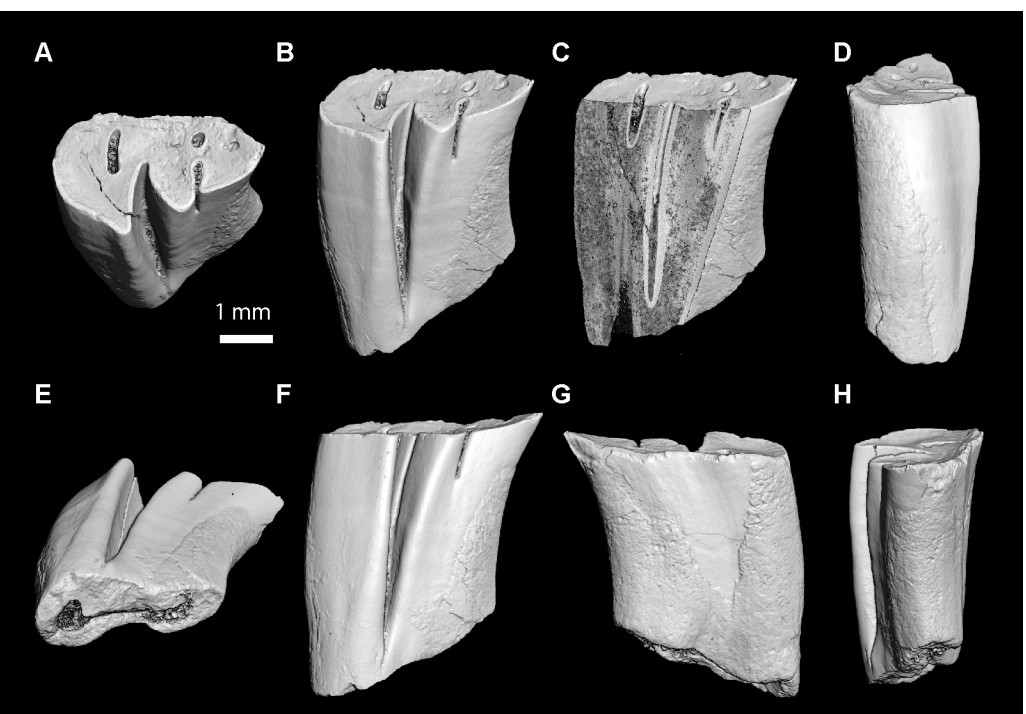

**Fig 6. Caviidae gen. et sp. indet. from TAR-31, Peru, MUSM 4283, right p4.** (A) Occlusal, (B) occluso-labial, (C) cross-sectional, (D) posterior, (E) ventral, (F) labial, (G) lingual, (H) anterior views. The illustrations of the fossil specimen are 3D digital models obtained by X-ray μCT surface reconstruction. The image in C is a 2D orthoslice of the tooth (cross-section).

## Formation and age

Lower member, Ipururo Fm., late middle Miocene (i.e., Laventan SALMA).

## Description

The MUSM 4283 p4 is slightly damaged at the level of its mesiolingual margin (Fig 6A, 6G and 6H). It also shows a crack on its labiodistal part. The enamel and dentine are damaged and eroded in several regions of the crown and occlusal surface. MUSM 4283 is clearly high crowned (Hling = 1.15; Hlab = 1.27; Fig 6B, 6D, 6F and 6G) and, although the presence of roots is not directly visible, the closing of the hypostriid suggests the closing of the crown too (Fig 6E and 6F). Hence, the tooth was seemingly protohypsodont. The cuspids/stylids and lophids are not visible because they are subsumed within three enlarged lobes, corresponding (from anterior to posterior) to the anterior projection, anterior lobe and posterior lobe. The dentine is predominating, surrounded by a continuous and thin enamel layer. The tooth has a triangular mesial margin. The anterior projection is narrower than the anterior lobe, which itself is narrower than the posterior lobe. The anterior projection is roughly tear-drop shaped, with a labial extremity short and distolabially directed. In its lingual part, the projection has an oval structure, which is hollow and shallow (Fig 6A and 6B). Nevertheless, it is not a fossettid because it is not surrounded by enamel layer. It is probably a dentine scar owing to an external impact during the mastication or a taphonomic process. The limit between the anterior projection of p4 and the anterior lobe is marked by the presence of: (i) labially, a flexid almost labially closed; and (ii) lingually, an isolated fossettid (Fig 6A). This fossettid is small, round and shallow. It has a thinner enamel layer than that of the margins of the tooth. As the mesialmost lobe of the tooth is determined as the anterior projection, the flexid almost closed would be the homologous of the interprismatic furrow *sensu* Pérez [51] separating the anterior projection from the anterior lobe in some stem-Caviidae (*Guiomys* and "*Prodolichotis pridiana*") [51, 87–90], and which is variably present in Caviidae [1, 55, 91–98]. The enamel layer of the distal margin of this flexid is thinner than that of its mesial margin and of the rest of the exterior enamel layer of the tooth. On the labial margin, the striid corresponding to the opening of the interprismatic furrow is mesiodistally very thin. Its depth represents less than 20% of the crown height (Fig 6B and 6C). Due to their position, the fossettid and/or the interprismatic furrow could be rests of the anteroflexid. The anterior lobe is quite oval and labially separated from the posterior lobe by a triangular hypoflexid. This flexid is linguolabially extended and its lingual extremity reaches more than the half of the width of the tooth. It displays a continuous and homogenous enamel layer. The hypostriid is mesiodistally large and almost reaches the neck of MUSM 4283. On the lingual margin of the tooth, there is an inward fold faintly developed at the transition between the anterior and posterior lobes. It could correspond to a very slight longitudinal furrow opposite to the hypoflexid (= primary internal flexid [hpi]). The posterior lobe is tear-drop shaped, with its labial apex short and mesiolabially directed. In its mesiolingual part, there is a fossettid linguolabially extended, with a thin enamel layer. Its depth is slightly shallower than the striid corresponding to the opening of the interprismatic furrow (Fig 6C). This fossettid probably corresponds to the metafossettid. Because the hypoflexid and metafossettid are not confluent, the tooth is non-taeniodont. The lingual fossettids and the hypoflexid (at its base) show a white deposit, which by its aspect, does not seem to be coronary cement, but more probably sediments that filled them after the death of the animal.

## Comparisons

The protohypsodonty, formation of lobes, linguolabial extension of the hypoflexid, and non-taeniodonty primarily suggest caviid affinities. The very slight development of a possible

longitudinal furrow opposite to the hypoflexid (= hpi), a slight development of the interprismatic furrow (= secondary external flexid [hse]), and a relatively narrow anterior projection exclude hydrochoerine affinities for this material from TAR-31. Moreover, MUSM 4283 does not have any additional flexids as observed in capybaras (i.e., internal flexids, including secondary internal flexids and supernumerary internal flexid; see [99]). MUSM 4283 shares characters, which would be ancestral during the evolution of the caviid family [89, 100], with stem-Caviidae: protohypsodonty (*Chubutomys*, *Luantus minor*, *Luantus propheticus*, *Luantus toldensis*, *Phanomys*, and *Eocardia*), presence of lingual fossettids on worn teeth (*Asteromys*, *Chubutomys*, *Luantus*, *Phanomys*, *Eocardia fissa*, and *Schistomys erro*), and a hypoflexid that does not reach the lingual margin of the tooth (*Asteromys*, *Chubutomys*, and *Luantus*) [52, 73, 79, 86, 100–104]. The presence of an anterior projection, which can be associated with an interprismatic furrow, is otherwise found in some derived stem-Caviidae (*Guiomys* and "*Prodolichotis*" *pridiana*) [51, 87–90] and crown-Caviidae [1, 55, 91–98]. However, *Guiomys*, "*Prodolichotis*" and the crown caviids do not have any lingual fossettids on worn teeth contrary to MUSM 4283. p4s of *L. toldensis* (MACN Pv SC4033 in Fig 5F [101]) and *Phanomys* [100] have a lingual thickening of the metalophulid I/their mesial margin, which could correspond to an incipient anterior projection. If the latter is really present in these taxa, it is less developed than on MUSM 4283. In all Caviidea with an interprismatic furrow associated with an anterior projection on p4s, the interprismatic furrow is never almost labially close, unlike on MUSM 4283. This tooth differs from p4s of *Asteromys* in having a hypoflexid more linguolabially extended; of *Asteromys* and *Luantus initialis* in having a higher crown; of *Asteromys*, *Chubutomys*, *Eocardia*, and *L. propheticus* in having an anterior projection; of *Phanomys* in having a thinner enamel; and of *Eocardia* in having a hypoflexid less linguolabially extended. The stem-caviid *Guiomys* is euhypsodont unlike MUSM 4283, and shows a p4 with a hypoflexid more linguolabially extended, a more developed anterior projection and longitudinal furrow opposite to the hypoflexid [51, 105]. MUSM 4283 differs from p4s of *Schistomys erro* and *Microcardiodon* in being protohypsodont, and in having an anterior projection and a well-marked hypoflexid; of *Microcardiodon* in having fossettids; and of *Schistomys erro* in having two persistent lobes [52, 87]. The size of MUSM 4283 is similar with that of *Chubutomys simpsoni*, *Chubutomys leucoreios*, *Luantus minor*, and *Guimoys unica*. As Dolichotinae gen. 2 large from La Venta (IGM 183425 in Fig 24.7A, p. 401 [18]), MUSM 4283 shows a very slight longitudinal furrow opposite to the hypoflexid, and a lingual extremity of the hypoflexid exceeding the half of the width of the tooth, but without reaching its lingual margin. However, MUSM 4283 is characterised by the presence of fossettids and an anterior projection, contrary to Dolichotinae gen. 2 large from La Venta. MUSM 4283 differs from p4s of "*P.*" *pridiana*, *Neocavia*, *Prodolichotis prisca*, *Dolicavia*, *Orthomyctera*, and *Dolichotis* in having a less developed longitudinal furrow opposite to the hypoflexid; of *Paleocavia*, *Cavia*, *Galea* and *Microcavia* in having a clearly less developed longitudinal furrow, which is opposite to the hypoflexid; of *Cavia*, *Galea* and *Microcavia* in having a hypoflexid without cement; of "*P.*" *pridiana*, *Neocavia*, *Paleocavia*? *mawka*, *Paleocavia impar*, *P. prisca*, *Dolicavia*, *Orthomyctera*, *Cavia*, *Dolichotis*, and *Galea* in having a linguolabially narrower hypoflexid; of *Cavia* in having a more developed anterior projection [28, 54, 55, 90, 94, 95, 106–113]. In conclusion, the morphology of MUSM 4283 appears particular by a mosaic association of characters, combining primitive characters (i.e., protohypsodonty, presence of fossettids on worn teeth, and a hypoflexid, which do not reach the lingual margin of the tooth) and more derived ones (i.e., development of a relatively large anterior projection); and also by an unique configuration of the fossettids and the interprismatic furrow. Thus, it is possible that MUSM 4283 documents a new species among the stem-Caviidae, but given the scarce available material, we provisionally identified it as Caviidea gen. et sp. indet.

Octochinchilloi Boivin, 2019 in Boivin, Marivaux & Antoine, 2019

Octodontoidea Waterhouse, 1839

Adelphomyidae Patterson & Pascual, 1968, nov. stat.

*Ricardomys* Boivin & Walton gen. nov. urn:lsid:zoobank.org:act:46B7B00E-E14D-4123-BF6D-9AB7AFF4AAD4

## Type species

*Ricardomys longidens* gen. et sp. nov.

## Species content

Only the type species.

## Derivation of name

Based on the original description of the species [37]: 'In honor of Richard Kay, professor of Evolutionary Anthropology at Duke University, who discovered the holotype of this species'.

## Geographic and stratigraphic distribution

La Venta (IGM-DU loc. 032), Cerro Colorado Member, Villavieja Fm. (Laventan, late middle Miocene), Huila Department, Colombia; Tarapoto/Juan Guerra (TAR-31), lower member, Ipururo Fm. (Laventan, late middle Miocene), San Martín Department, Peru.

## Generic diagnosis

As for the type and only known species.

*Ricardomys longidens* Boivn & Walton sp. nov. urn:lsid:zoobank.org:act:83062BD3-9006-4C73-BB02-37BF66BF6663

Fig 7 and S1 Table

*Nom. nud. Ricardomys longidens* Walton, 1990, Fig 9A, 9B, p. 31.

*Nom. nud. Ricardomys longidens* Walton, 1997, Figs 24.1C and 24.2D, p. 394, 395.

## Holotype

IGM 183847, left mandibular fragment bearing dp4–m2 (Walton, 1990: Fig 9A, 9B; Walton, 1997: Figs 24.1C and 24.2D; Fig 7A).

## Referred material

MUSM 4397–4401 (Fig 7B and 7F), left dp4s; MUSM 4402–4406 (Fig 7J), right dp4s; MUSM 4407, right lower molar; MUSM 4408, left m1 or m2?; MUSM 4409 (Fig 7C), left m1?; MUSM 4410 (Fig 7G), right m1?; MUSM 4411–4413 (Fig 7D and 7H), left m2?s; MUSM 4414, 4415 (Fig 7K), right m2?s; MUSM 4416–4419 (Fig 7E and 7I), left m3?s; MUSM 4370–4373, 4391–4394 (Fig 7U and 7V), left upper teeth; MUSM 4374, 4395, 4396 right upper teeth; MUSM 4375 (Fig 7P–7R), left fragment of maxillary bearing dP4 (or M1)? and M1 (or M2)?; MUSM

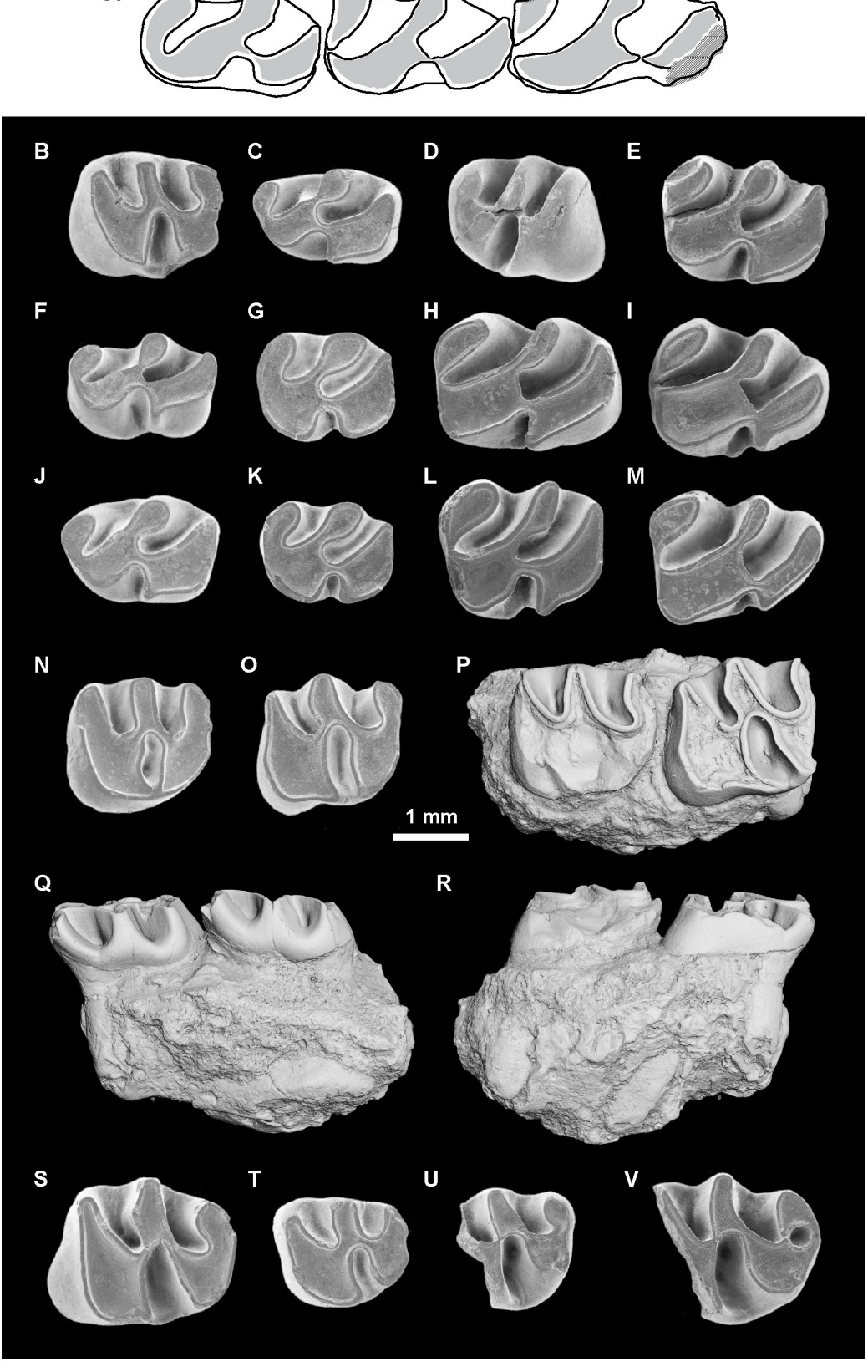

**Fig 7.** *Ricardomys longidens* **gen. et sp. nov.** (A) Material from La Venta, Colombia. (B–V) Material from TAR-31, Peru. (A) IGM 183847***, left mandibular fragment bearing dp4–m2. (B) MUSM 4401, left dp4. (C) MUSM 4409, left m1?. (D) MUSM 4411, left m2?. (E) MUSM 4416, left m3?. (F) MUSM 4400, left dp4. (G) MUSM 4410, right m1? (reversed). (H) MUSM 4413, left m2?. (I) MUMS 4418, left m3?. (J) MUSM 4406, right dp4 (reversed). (K) MUSM 4415, right m2? (reversed). (L) MUSM 4377, left dP4 (or M1)?, morph 1. (M) MUSM 4380, right dP4 (or M1)?, morph 1 (reversed). (N) MUSM 4376, left dP4 (or M1)?, morph 1. (O) MUMS 4384, left M1 or M2?, morph 2. (P–R) MUSM 4375, left maxilla fragment bearing dP4–M1 (or M1–2)?, morph 1 and 2. (S) MUSM 4388, right M1 or M2?, morph 2 (reversed). (T) MUSM 4390, right M3?, morph 3. (U) MUSM 4392, left fragmentary upper molar, morph 4. (V) MUSM 4393, left fragmentary upper molar, morph 4. (A-P, S-V) Occlusal, (Q) labial, and (R) lingual views. The asterisks appoint the holotype material. The illustrations of the fossil specimens are computerized schematic line drawings based on Walton (Fig 9A [37]). (A), scanning electron photomicrographs (B-O and S-V), and 3D digital models obtained by X-ray µCT surface reconstruction (P-R).

4376, 4377 (Fig 7L and 7N), left dP4s (or M1s)?; MUSM 4378–4380 (Fig 7M), right dP4s (or M1s)?; MUSM 4381, right dP4?; MUSM 4382–4384 (Fig 7O), left M1 or M2?s; MUSM 4385–4388 (Fig 7S), right M1 or M2?s; MUSM 4389, left M2?; MUSM 4390 (Fig 7T), right M3?.

### Derivation of name

Based on the original description of the species [37]: 'from the latin *longidens*, in reference to the length and slenderness of the tooth row of the holotype of this species'.

### Type locality

IGM-DU loc. 032 (only for the holotype), El Cardón Red Bed, La Venta Badlands, upper Magdalena Valley, Huila Department, Colombia.

### Other locality

TAR-31, Tarapoto/Juan Guerra, Río Mayo, Shapaja road, San Martín Department, Western Amazonia, Peru.

### Formation and age

Cerro Colorado Member, Villavieja Fm.; lower member, Ipururo Fm.; late middle Miocene (i.e., Laventan SALMA).

### Diagnosis

Octodontoid rodent characterised by a small size and non-taeniodont teeth with thick and oblique lamellae. Lower teeth with a trilophodont pattern characterised by the absence of metaconid cristid, posterior arm of the protoconid, neomesolophid and mesolophid. Thick lingual parts of the first and second lamellae on lower teeth, and a first lamella divided into two parts by a thin and shallow furrow on lower molars. Upper teeth with a transversal extension of the hypoflexus, prompt loss of the third transverse crest with wear and the absence of metaloph. Differs from *Eodelphomys* in being smaller and in having lower molars with a thinner furrow dividing the first lamella, more expanded lingual flexids, metaflexid always open at a more advanced stage of wear, less transversely extended hypoflexid, an entoconid area with thickening and without discernable entoconid arms. Differs from *Xylechimys* in having less transversely extended hypoflexid, an ectolophid which is not discernable because fused with the hypolophid, hypoconid and the protoconid, an indistinct posterior arm of the metaconid, and no posterior arm of the protoconid on lower molars. Differs from *Ethelomys*, *Paradelphomys*, *Quebradahondomys*, *Deseadomys*, *Adelphomys*, *Prostichomys*, *Stichomys*, *Prospaniomys*, and *Spaniomys* in having a thickening of the lingual parts of the two first lamellae on

lower teeth; from *Ethelomys* in having thicker cristids and a less-transversely extended hypo-flexid on lower molars; from *Paradelphomys* in having a non-taeniodont and trilophodont dp4s with a lingual part of the first lamella less distally extended and the first and second lamellae very close in position, and no furrow-shaped flexids on lower teeth; from *Quebradahond-omys* in having more oblique cristids on lower molars; from *Deseadomys*, *Adelphomys*, *Prostichomys*, *Stichomys*, *Prospaniomys*, and *Spaniomys* in always having trilophodont dp4s, m1s, and m2s; from *Deseadomys* and *Prospaniomys* in having no recognisable cusp(-id)s and more oblique and thicker lamellae; from *Deseadomys* and *Spaniomys* in having a less trans-versely extended hypoflexid on lower teeth; from *Prospaniomys* in having a more transversely extended hypoflexus on upper teeth; from *Adelphomys* in having posterior fossette probably less persistent on upper teeth, a metaflexid always open at a more advanced stage of wear on lower teeth, and a first lamella which can be divided into two parts on lower molars; from *Spa-niomys* in having more oblique lamellae, and a straight and distolingually directed second lamella on m3s; and from *Prostichomys* in having thicker lamellae, posterior fossette less per-sistent on upper teeth, and no mesostylid and an indistinct posterior arm of the metaconid on lower molars.

## Description

IGM 183847 is a left mandibular fragment preserving dp4–m2 (Fig 7A). The body of the man-dible is anteriorly broken at the level of the anterior root of the dp4. Posteriorly, the angular apophysis and all the ascending ramus, including the mandibular condyle, are missing. The coronoid process is broken at its base. Labially, the notch for the insertion of the tendon of the *zygomatico-mandibularis pars infraorbitalis* is small, below m1, and situated ventrally on the labial edge of the mandible. The anterior tip of the masseteric crest and that of the lateral crest end below the m1, and they appear to be not linked to the notch for the insertion of the tendon of the *zygomatico-mandibularis pars infraorbitalis*, at the level of its posteroventral and poster-odorsal regions, respectively. The masseteric crest is posteriorly broken. It is posteroventrally directed and poorly prominent in its anterior part. The lateral crest is posterodorsally directed and almost undistinct.

The teeth referred to this taxon are brachydont. The less worn tooth from TAR-31 (MUSM 4377) presents a fairly high crown, but its hypsodonty indexes stay below 1 (Hling = 0.62; Hlab = 0.63). This tooth was probably submesodont at earlier wear stages. All lower and upper teeth are non-taeniodont. The cusp(-id)s and styles (stylids) are not visible even on the tooth at the earliest stage of wear (MUSM 4377). They are subsumed within loph(-id)s, thereby forming flat-topped and thick crestiform structures (lamellae *sensu* Patterson & Pascual [56]). Each lamella displays a continuous enamel layer, coating a dentine layer, which is increasingly thicker with the dental wear.

The dp4s are longer than wide (Fig 7A, 7B, 7F and 7J). Their trigonid is wide but narrower than the talonid. These teeth are characterised by two roots and a trilophodont pattern with oblique lamellae. Mesiolabially, the area of the protoconid is well expanded and usually trian-gular-shaped. From this area, emerge mesially the first lamella and distally the second lamella. The first lamella is mesially convex. Its lingual part is short and would correspond to the meta-conid area (without its posterior arm?). It can be thick with respect of the thin central part of the first lamella (metalophulid I; IGM 183847, MUSM 4400, 4401, 4405, and 4406; Fig 7A, 7B, 7F and 7J). On the worn MUSM 4397 dp4, the protoconid area is not recognisable from the rest of the first lamella, these structures are fully fused, forming a round and large structure of dentine at the mesial part of the tooth. On all dp4s, the second lamella is a central cristid disto-lingually directed. It would result of the fusion between the entoconid + the hypolophid + the

mesial and distal ectolophids + a part of the protoconid. As the lingual part of the first lamella on some specimens, the lingual part of the second lamella (the entoconid area), by its thickness, is well discernable from the other structures. There would be neither metaconid cristid, neomesolophid, posterior arm of the protoconid, nor mesolophid. The third lamella is cristiform on the dp4 at the earliest stage of wear (MUSM 4401; Fig 7B) and diamond/parallelogram-shaped on other dp4s. It would include the hypoconid + its anterior outgrowth + the posterolophid. On MUSM 4400 and 4401, it bears a small spur in its mesiolabial aspect (i.e., anterior outgrowth; Fig 7B and 7F). It is linked to the second lamella by a complete and thin anterior arm of the hypoconid. On IGM 183847, the anterior arm of the hypoconid reaches a short cristulid stemming from the second lamella (Fig 7A). The anterior arm of the hypoconid is mesiolingually directed, strongly oblique on IGM 183847 and MUSM 4399–4401 (Fig 7B and 7F), and more longitudinal on MUSM 4404–4406 (Fig 7J). Owing to the absence of the metaconid cristid, posterior arm of the protoconid, neomesolophid and mesolophid, the anterolingual flexid corresponds to the confluence of the anteroflexid with the mesial and distal mesoflexids. This flexid and the metaflexid are widely open lingually and more extended than the U-shaped hypoflexid.

Lower molars have a pattern similar to that characterising dp4s (i.e., non-taeniodonty, trilophodonty, and obliquity, global shape and composition of lamellae; Fig 7A, 7C–7E, 7G–7I and 7K). Comparatively to dp4s, lower molars have three roots (i.e., two mesial and one distal) and a more rectangular shape with a transverse mesial margin. Four lower molars (MUSM 4416–4419; Fig 7E and 7I) show a distal part of the talonid labiolingually pinched, and as such they are probably m3s (determined as m3?s). MUSM 4416 (Fig 7E) also has a posterior root directed distally, a character typically found on m3s. Among the isolated lower molars without distal pinch, some are small-sized, and as such could be m1s (determined as m1?s; MUSM 4409, and 4410; Fig 7C and 7G). On IGM 183847 and MUSM 4417, 4419 and 4415 (Fig 7K), the lingual part of the first lamella is separated from the protoconid area by a thin and shallow furrow open on the mesial margin of the tooth (MUSM 4415 is broken at the level of the furrow giving the impression that it is larger and deeper; Fig 7K). On MUSM 4417, the margins of the furrow are composed of two sloped cristulids, one lingual and one labial. The lingual cristulid stems from the metaconid area (the ?anterior arm of the metaconid), while the labial one stems from the protoconid area (the ?anterior arm of the protoconid). They are connected at the base of the furrow. On molars with a more advanced stage of wear (MUSM 4408–4414 and 4418; Fig 7C–7E and 7G–7I), this furrow is fully closed. The morphology of the second and third lamellae is very similar of that of dp4s, except they are longer.

Regarding upper teeth (Fig 7L–7V), the absence of complete tooth row and a documentation of the whole stages of wear make it difficult to recognise the dental intra-specific variation related to the difference of loci, wear, or polymorphism. Four morphs can be distinct. The unique fragment of maxillary found at TAR-31 and attributed to this species, MUSM 4375 (Fig 7P–7R), bears two upper teeth. This specimen and the comparison with upper teeth of other adelphomyines or closely related forms (e.g., *Deseadomys*, *Adelphomys*, *Prostichomys*, and *Spaniomys*) allow us to propose hypotheses regarding the locus determination for these morphs:

- Morph 1: small upper teeth longer than wide or subsquare, and with a quadrangular occlusal outline and three transverse lophs (MUSM 4375–4381; Fig 7L–7N and 7P–7R). MUSM 4381 seems to have three roots (i.e., two labial and one large lingual), whereas other teeth have not conserved their roots. This morph could correspond to the morphology of dP4s (or M1s);

- Morph 2: upper teeth larger than those of the morph 1, either subsquare or rectangular, and with quadrangular occlusal outlines, three transverse lophs and four roots (MUSM 4375 and

4382–4389; Fig 7O–7S). This morph could correspond to the morphology of M1s or M2s. MUSM 4389 has a very short posterior crest as MUSM 4390 (see below morph 3). This shortness may be related to a strong labial displacement of the hypocone. Nevertheless, MUSM 4389 shows typical characters of morph 2: large size, quadrangular occlusal outline with a straight distal margin and four roots. It could correspond to a M2;

- Morph 3: this morph is represented by one small upper tooth (MUSM 4390; Fig 7T) characterised by three roots, a more round outline than teeth of the morphs 1 and 2, distal pinch, and three transverse lophs. This morph could correspond to the morphology of M3s;

- Morph 4: this morph is represented by six mesially-broken upper teeth (MUSM 4391–4396; Fig 7U and 7V). These fragments of teeth have different size, some are small-sized (MUSM 4391 and 4392; Fig 7U), medium-sized (MUSM 4393 and 4395; Fig 7V), while other are larger (MUSM 4396). They have a more rounded occlusal outline than teeth of morphs 1 and 2, and a distal part slightly labiolingually narrower than the mesial part. They seem to be mesiodistally elongated. Because they are broken, the hypsodonty index cannot be calculated. However, they have thinner crests and deeper flexi than teeth of other morphs, and three (MUSM 4392 and 4395; Fig 7U) or four (MUSM 4391, 4393, 4394, and 4396; Fig 7V) transverse lophs. Owing to the latter characters, this morph likely clusters upper molars at less advanced wear stages than those of morphs 2 and 3.

The first lamella is diamond/parallelogram-shaped with a large and rounded apex at the mesiolingual corner of the tooth. Two other apices, the first at the mesiolabial corner of the tooth and the second in the center of its lingual edge, are usually narrower and tapered. This lamella would include the protocone + the posterior outgrowth of the protocone + the anteroloph. The second lamella, which would correspond to the paracone + the labial protoloph, is either transverse or oblique (i.e., mesiolabially directed) or lingually oblique and labially transverse. It is short due to a strong transversal extension of the hypoflexus. The lingual protoloph is thin, distolabially directed and complete, thus linking the first lamella to the second (i.e., non-taeniodont pattern). On the trilophodont upper teeth (MUSM 4373, 4375–4389, 4390, and 4395; Fig 7L–7U), the third lamella is usually bean-shaped, but can be suboval (MUSM 4389 and 4390; Fig 7T). On the tetralophodont MUSM 4394, this third lamella presents a round flexus still distolabially open. On other tetralophodont upper teeth (MUSM 4391–4393 and 4396; Fig 7V), this flexus is fully closed, thereby forming a round fossette. This flexus/fossette separates the third transverse crest from the posteroloph. On all tetralophodont upper teeth, these two transverse lophs are very thin and short, except on MUSM 4394, in which the third transverse crest is thicker. On MUSM 4391 and 4393 (Fig 7V), the third crest is composed of a very short mesoloph, whose lingual extremity is connected to a mesolophular spur. The third lamella on trilophodont upper teeth would include the mesostyle and/or the metacone (see [50]) + the third transverse crest + the posteroloph + the hypocone + its anterior arm. Near the center of the tooth, the second lamella and the third lamella are connected via a mure mesiolabially directed (strongly oblique).

## Comparisons

A small size, non-taeniodont teeth with thick and oblique lamellae, and a transverse extension of the hypoflexus, the prompt loss of the third transverse crest with wear and the absence of metaloph on upper teeth are dental traits found in adelphomyine octodontoids [4, 56], thereby suggesting adelphomyine affinities for this material from La Venta and TAR-31. By their size and morphology, the lower teeth from TAR-31 are close to those of IGM 183847 from La Venta, previously attributed to *Ricardomys longidens nom. nud.* (Figs 24.1C and 24.2D [18];

Fig 9A, 9B [37]; Fig 7A). As lower teeth from TAR-31, the IGM 183847 lower teeth have a trilo-phodont pattern characterised by the absence of metaconid cristid, posterior arm of the proto-conid, neomesolophid and mesolophid. These lower teeth also show oblique lamellae, thick lingual parts of the first and second lamellae, and a first lamella divided into two parts by a thin and shallow furrow on lower molars. However, the lower molars from TAR-31 slightly differ from IGM 183847 in having the first lamella more oblique. A similar pattern to dp4s from La Venta and TAR-31 is also found on the MUSM 2856 dp4 attributed to Adelphomyi-nae indet. 2 from CTA-61 (Peru, late Oligocene) [71]. Unlike dp4s from La Venta and TAR-31, MUSM 2856 does not have a hypoflexid transversely oriented, but the latter is distolin-gually oriented. Other adelphomyines or closely related forms have strictly trilophodont lower molars: *Eodelphomys* (Santa Rosa, Peru; ?late Eocene/early Oligocene) [81], *Ethelomys* (Cabeza Blanca, Argentina; late Oligocene) [114], *Paradelphomys* (Sarmiento Fm., Argentina; early Miocene) [56], and *Quebradahondomys* (Quebrada Honda, Bolivia; late middle Miocene) [105]. *Eodelphomys* is only known from a single m3 (LACM 144298 in p. 130 [81]), which shows a first lamella divided into two parts by a furrow. However, the latter is larger than that on lower molars from La Venta and TAR-31 (when the furrow is present). The lower molars from La Venta and TAR-31 also differ from LACM 144298 in having lingual flexids more expanded mesiodistally, always open at a more advanced stage of wear, less transversely extended hypoflexid, and an entoconid area with thickening and without discernable entoco-nid arms. In comparison with the material form La Venta and TAR-31, *Ethelomys*, *Paradel-phomys*, and *Quebradahondomys* do not have any thickening of the lingual parts of the two first lamellae (or at least of the second lamella) on lower teeth; *Ethelomys* has thinner cristids and a more transversely extended hypoflexid on lower molars; *Paradelphomys* has furrow-shaped flexids on lower teeth, a taeniodont and tetralophodont dp4 with a lingual part of the first lamella more distally extended (with posterior arm of the metaconid?) and the first and second lamellae very close in position; and *Quebradahondomys* has less oblique lamellae on lower molars. As the lower teeth from La Venta and TAR-31, *Xylechimys* (Laguna de los Machos, Argentina; late Oligocene) [56] appears to develop a lingual thickening of the second lamella on lower molars with the dental wear (visible on the m1 of the holotype). Nevertheless, *Xylechimys* has a trigonid of lower molars clearly different characterised by: (i) a posterior arm of the protoconid linked to the metaconid area, thus delimiting a small and rounded anterofos-settid, (ii) discernable ectolophid and posterior arm of the metaconid, and (iii) a more trans-versely extended hypoflexid. The general pattern of the upper teeth from TAR-31 is similar to that of adelphomyines (or closely-related forms) documented by upper teeth: *Deseadomys* (Cabeza Blanca and La Flecha, Argentina; late Oligocene) [79], *Adelphomys* (Pinturas Fm. and Santa Cruz Fm., Argentina; early Miocene) [73, 77], *Prostichomys* (Sarmiento Fm. and Pin-turas Fm., Argentina; early Miocene) [77, 115, 116], *Stichomys* (Santa Cruz Fm.; late early Mio-cene) [73], *Prospaniomys* (Sarmiento Fm.; early Miocene) [11, 78, 117], and *Spaniomys* (Pinturas Fm. and Santa Cruz Fm.; Pampa Castillo, Chile; late early Miocene) [16, 73, 77, 118, 119]. Contrary to the material from La Venta and TAR-31, all these taxa display no thickening of the lingual parts of the two first lamellae (or at least of the second lamella) on lower teeth, and can have tetralophodont lower teeth (usually dp4s, m1s or m2s). The tetralophodont pat-tern of these lower teeth is characterised by the presence of a posterior arm of the protoconid + metaconid cristid and/or neomesolophid (and mesolophid on some dp4 of *Prostichomys*). The material from La Venta and TAR-31 also differs from *Deseadomys* and *Prospaniomys* in having no recognisable cusp(-id)s and more oblique and thicker lamellae; from *Deseadomys* and *Spaniomys* in having a less transversely extended hypoflexid on lower teeth; from *Prospa-niomys* in having a more transversely extended hypoflexus on upper teeth; from *Adelphomys* in having a posterior fossette probably less persistent on upper teeth, always open at a more

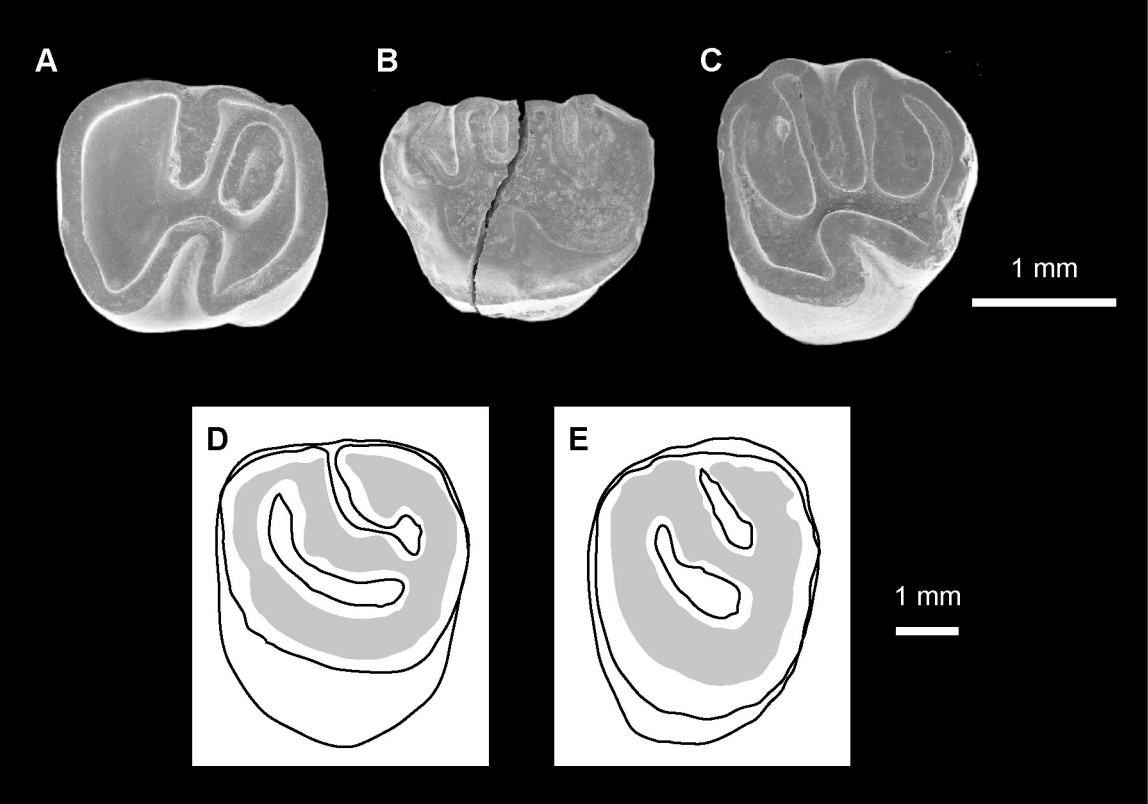

**Fig 8.** Octodontoidea gen. et sp. indet. from TAR-31, Peru (A–C) and *Microscleromys* sp. 1 from La Venta, Colombia (D, E). (A) MUSM 4662, left m1 or m2. (B) MUSM 4660, left dP4. (C) MUSM 4661, right M3 (reversed). (D) UNC unnumbered specimen, left P4. (E) IGM-DU 85–411, left P4. Occlusal views. The illustrations of the fossil specimens are scanning electron photomicrographs (A-C) and computerized schematic line drawings based on Walton (Fig 15 [37]) (D-E).

advanced stage of wear on lower teeth, and a first lamella which can be divided into two parts on lower molars; from *Spaniomys* in having more oblique lamellae, and a straight and distolingually directed second lamella on m3s; and from *Prostichomys* in having thicker lamellae, a posterior fossette less persistent on upper teeth, and no mesostylid and no discernable posterior arm of the metaconid on lower molars. To summarize, due to the morphological similarity between the lower teeth from La Venta and TAR-31, we assign them to the same taxon. By its particular set of characters, the entire material from these both areas is determinable as a new species, hence we validate *Ricardomys longidens*.

Octodontoidea gen. et sp. indet.

Fig 8 and S1 Table

### Referred material

MUSM 4662, left m1 or m2 (Fig 8A); MUSM 4660, left dP4 (Fig 8B); MUSM 4661, right M3 (Fig 8C).

### Locality

TAR-31, Tarapoto/Juan Guerra, Mayo River, Shapaja road, San Martín Department, Western Amazonia, Peru.

## Formation and age

Lower member, Ipururo Fm., late middle Miocene (i.e., Laventan SALMA).

## Description

Due to the very advanced stage of wear of the MUSM 4662 m1 or m2 (Fig 8A), its occlusal pattern is faintly discernable, especially on the mesial area of the tooth, where a broad dentine platform is present. There is a thick, continuous enamel layer on the entire periphery of the crown. It is thinner at the level of the lingual parts of the hypoflexid and of the mesoflexid. Only three transverse lophids are visible, delimited by a very thin enamel layer, which is absent in some regions. The mesialmost lophid is transverse and connects the area of the posterior arm of the metaconid/mesostylid to the junction of the protoconid (or the protoconid cristid) and the ectolophid. This lophid is recognised as a second transverse cristid (i.e., neomesolophid and/or posterior arm of the protoconid). Mesially, it is partially fused with the mesial dentine platform, which probably corresponds to the fusion at least of the metalophulid I with the anteroflexid. The transverse lophid that is second in position, is the hypolophid, running slightly labiomesially from the area of the entoconid to link the posterior part of the ectolophid. The hypolophid is almost parallel to the second cristid. The ectolophid is short and gently oblique. The transverse lophid appearing in third position is a curved posterolophid, lingually connected to a strong posterior arm of the entoconid. The area of the hypoconid is enlarged and oval, formed by the fusion of the hypoconid, its anterior outgrowth and its posterior arm (if present) and/or the labial part of the posterolophid. There is a thick and complete anterior arm of the hypoconid (i.e., non-taeniodont pattern). The mesoflexid and metaflexid are lingually closed, whereas the hypoflexid remains labially open. The metafossettid is oval and smaller than the mesofossettid.

The MUSM 4660 dP4 (Fig 8B) has a trapezoidal crown outline in occlusal view (i.e., labial marging longer than the lingual margin). This tooth shows an advanced stage of wear. Labially, the dental structures have a thick enamel layer, while the lingual structures are surrounded by a clearly thinner enamel layer. The posteroloph area is very enlarged due to the wear, and forms a dentine platform, in which a metacone, a metaloph or a rest of this loph may be included. The hypoflexus is triangular and poorly extended linguolabially. It separates lingually the area of the protocone from that of the hypocone. The areas of the protocone, mure and of the hypocone are also strongly enlarged and broadly merged together, as well as with the platform of the posteroloph, forming a S-shape dentine structure on the distolingual part of the tooth. From this dentine structure, three main lophs run labially: the anteroloph, the labial protoloph and the third transverse crest. The anteroloph is curved with a lingual part thicker than its labial part. The labial protoloph and third crest are transverse and parallel. The paraflexus is still open labially, separating the anteroloph from the labial protoloph. In contrast, the other transverse lophs are connected labially, forming a small oval mesial mesofosette and a more linguolabially extended distal mesofosette.

As MUSM 4662, the MUSM 4661 M3 (Fig 8C) is much worn, but it documents a slightly earlier stage of wear. A thick enamel layer surrounds the tooth as well, but the latter is internal at the level of the mesial mesoflexus, which is still opened labially contrary to MUSM 4662. Moreover, all the transverse lophs and flexi are still discernable. The hypocone is clearly more labially situated than the protocone, which indicates that the specimen would be a M3. MUSM 4661 is tetralophodont and non-taeniodont. The area of the protocone corresponds to a broad dentine platform, including the protocone, its posterior outgrowth and its anterior arm (if present) and/or the lingual part of the anteroloph. From this platform, the anteroloph runs labially and links the paracone. The lingual part of the anteroloph is transverse, whereas its

labial part is curved. Due to the connection between the anteroloph and the paracone, a paraf-ossette is formed and reduced to a thin and straight structure. The enamel layer that surrounds it is thick, notably on its mesial margin. On the labial margin of the tooth, the paracone and mesostyle are strong and of similar size. The posterior arm of the protocone is thick and obli-que. It connects to the mure, which is linguodistally oriented and in the continuity with the anterior arm of the hypocone. The labial protoloph and the third transverse crest (i.e., meso-loph and/or mesolophule) are roughly straight and parallel. There is neither metaloph nor discernable metacone. A well-curved posteroloph is labially connected to the mesostyle and lingually to the area of the hypocone, which is triangular-shaped. The mesial mesoflexus is very thin. The posterior fossette (i.e., confluence of the distal mesoflexus and the metaflexus) is reduced to a very small and rounded structure. As the parafossette, the mesial mesoflexus and the posterior fossette are surrounded by a thick enamel layer.

## Comparisons

By its small size, weak loph(-id) obliquity, non-taeniodonty, tetralophodont upper molars (i.e., absence of metaloph), and a strong mesostyle on upper molars, these three teeth from TAR-31 likely document a single and distinctive octodontoid. Indeed, many Palaeogene and Neogene octodontoids also display these characters. They are usually characterised by a reduction or the absence of the second transverse cristid, and a reduction of the posterior arm of the metaconid on lower molars, and a metacone more distally positioned on upper molars (e.g., *Acarechimys*, *Deseadomys*, *Dudumus*, and *Caviocricetus*) [30, 79, 120, 121], characters absent on the speci-mens from TAR-31. However, *Paulacoutomys* and some lower molars of *Platypittamys*, *Scia-mys principalis*, *Prostichomys* and *Plesiacarechimys* show a complete second transverse cristid and a well-developed posterior arm of the metaconid [77, 115, 116, 122–126], as it is observed on the material from TAR-31. Among the latter taxa, the material from TAR-31 is closer to *Paulacoutomys* and *Plesiacarechimys* in having transverse loph(-id)s and a similar extension of the flexi(-ds). In addition, these genera can have an ectolophid slightly oblique on lower molars, as it does on MUSM 4662. The material from TAR-31 differs from *Paulacoutomys* and *Plesiacarechimys* in having a smaller size; from *Platypittamys* and *Prostichomys* in having a more longitudinal ectolophid on lower molars; from *Platypittamys* in having M3 with third crest that is not curved, a mesial mesoflexus less extended, a posterior flexus/fossette more extended; from *Prostichomys* and *S. principalis* in having a mesostyle less connected to the pos-teroloph on dP4s and upper molars; from *Prostichomys* in having a more transverse hypolo-phid on lower molars, a more transverse labial protoloph on dP4s and upper molars, and more transverse M3 with a wider distal part; and from *S. principalis* in having a mesial part of the lower molars (= fusion of the metalophulid I with the second cristid) more mesiodistally long at its lingual and labial extremities, a hypoflexid less extended labiolingually, and a mesoflexid which is lingually closed although a metafossettid is still present on lower molars, and a hypo-flexus less extended labiolingually and a mesial mesoflexus which is strongly reduced although the posterior fossette is still present on upper molars. In sum, given the condition and scarcity of the material, we provisionally refer these teeth to an octodontoid indet.

Chinchilloidea Bennett, 1833

*Microscleromys* Boivin & Walton gen. nov. urn:lsid:zoobank.org:act:785C5F6A-DCCD-4C02-B136-A2B49702E6DA

## Type species

*Microscleromys paradoxalis* gen. et sp. nov.

## Species content

The type species and *Microscleromys cribriphilus* gen. et sp. nov.

## Derivation of name

Based on the original description of the species [37]: 'In reference to its small size and general resemblance to *Scleromys* Ameghino, 1887'.

## Geographic and stratigraphic distribution

La Victoria Fm. and Baraya Member of the Villavieja Fm. (Laventan SALMA, late middle Miocene), Huila Department, Colombia; Tarapoto/Juan Guerra (TAR-31), lower member, Ipururo Fm., (Laventan SALMA, late middle Miocene), San Martín Department, Peru.

## Generic diagnosis

Chinchilloid rodent characterised by a small or moderate size. Relative size of m3s/M3s smaller in average than that of m1–2s/M1–2s. Brachyodont to mesodont teeth. P4s and upper molars with noticeable high crowns. Upper teeth with a higher lingual crown and lower molars with a higher distolabially crown. Absence of cement or in small quantity. Homogeneous thickness of the enamel layer of dental strutures. Poorly distinct cusp(-id)s from the loph(-id)s, which do not form lamina. At early stages of wear, lower teeth with pseudo-taeniodont or taeniodont pattern, lower molars with a pseudo-taeniodont pattern more frequently present, taeniodont P4s, dP4s and upper molars with a taeniodont pattern more frequently present. dp4s with a metalophulid I not connected to the metaconid and protoconid at early stages of wear, a posterolophid well-separated from the entoconid, a mesostylid mesially and distally isolated in most cases, and a tendency to the reduction or absence of the third crisitid. Lower molars with usually a groove on the metalophulid I at early stages of wear. Suboval dP4s, with well-arcuate anteroloph and posteroloph as well as a bulbous protocone slightly or strongly separate from the anteroloph. Round/oval P4s with a connection between the posterior outgrowth of the protocone and the hypocone. Upper molars with usually a mure oblique and aligned with the anterior arm of the hypocone. Important variation in terms of development, configuration and orientation of the second transverse cristid on p4s and lower molars, and of the third transverse crest and metaloph on upper teeth, but with a tendency to their reduction and absence. Trilophodont lower molars in most cases. Among upper teeth, P4s with stronger reduction of the third crest and metaloph. Upper molars with a third crest more frequently reduced to a mesoloph and without metaloph. Persistent flexids/fossettids on lower molars and flexi/fossettes except the posterior ones (the distal mesoflexus and metaflexus if present) on upper molars. Differs from *Chambiramys* in having slightly higher crowns, and more quadrangular upper molars; from *Eoincamys*, *Incamys* and *Scleromys*/"*Scleromys*" in having more square-shaped upper molars and a more important reduction of the third transverse crest on upper molars (i.e., more frequent trilophodont pattern than in these genera); from *Eoincamys ameghinoi*, *Eoincamys pascuali* and *Incamys* in having a second transverse cristid usually more reduced on lower molars; from *Eoincamys pascuali* in having a thinner furrow (when it is present) on the metalophulid I, which can be situated at the middle of this lophid; from *Saremmys* in having lower-crowned teeth, thinner loph(-id)s, a metaconid postionned more distally and strongly connected to the protoconid by a thick second cristid (metaconid cristid and/or posterior arm of the protoconid) on dp4s, a less oblique hypolophid on p4s and lower molars, a more bulbous protocone and a hypocone mesiolabially-distolingually compressed on dP4s, and some upper molars with a distinct metacone and a long metaloph and

mesoloph; from *Saremmys*, *Scleromys*/"*Scleromys*", *Garridomys* and *Drytomomys* in having an anterior arm of the hypoconid, which can be present on p4s and lower molars, and in a more curved posterolophid on lower molars; from *Incamys*, *Saremmys*, *Scleromys*/"*Scleromys*", *Garridomys* and *Drytomomys* in having a smaller size; from *Garridomys* and *Drytomomys* in having a second transverse cristid usually more reduced on lower molars; from *Garridomys* in having a homogeneous thickness of the enamel layer, thinner loph(-id)s, and strong and high protoconid and ectolophid on p4s; from "*Scleromys*" *colombianus* and *Drytomomys aequatorialis* in having a separation between the hypocone and the posteroloph thinner and less persistent at early wear stages on upper molars; and from *Drytomomys* in having a less oblique hypolophid on lower molars.

*Microscleromys paradoxalis* Boivin & Walton sp. nov. urn:lsid:zoobank.org:act:38631BE9-32E6-4E5D-B5CE-469C37B98D8C

Fig 9 and S1 Table

*Nom. nud. Microscleromys cribriphilus* Walton, 1990, Fig 12F, 12G-12O, p. 45.

*Nom. nud. Microscleromys paradoxalis* Walton, 1990, Fig 11A-11J, p. 39.

*Nom. nud. Microscleromys* ?*paradoxalis* Walton, 1990, Fig 13B, 13C p. 49.

*Microscleromys* sp. Walton, 1990, Fig 13A, p. 49.

*Nom. nud. Microscleromys cribriphilus* Walton, 1997, Fig 24.2.N, p. 295.

*Nom. nud. Microscleromys paradoxalis* Walton, 1997, Fig 24.2.G-I, p. 295.

## Holotype

IGM 250321, right M1 or M2? (Fig 9P).

## Referred material

MUSM 4643 (Fig 9A–9E), left fragment of mandible bearing incisor and dp4–m1; ING-KU 8604, right fragment of mandible bearing p4–m2; MUSM 4640–4642 (Fig 9F), left dp4s; IGM 250306 (Fig 9G), left p4; IGM 250305 and 250308 (Fig 9K), right p4s; MUSM 4644, left lower molar; IGM 250265, MUSM 4645, right lower molar; IGM 250288, 250307, IGM-DU 88–814, 89–675, MUSM 4646–4650 (Fig 9H, 9I, 9M and 9N), left m1 or m2; IGM 250319, MUSM 4651–4656 (Fig 9L), right m1 or m2s; IGM 250274, 250285, 250304, MUSM 4657 (Fig 9J), left m3s; IGM 251020 (Fig 9O), right m3; IGM 250302, IGM-DU 88–815 (Fig 9Z), ING-KU unnumbered specimen (Fig 9U), MUSM 4623 (Fig 9Q), left dP4s; MUSM 4624 (Fig 9A'), left P4; MUSM 4625 (Fig 9V), right P4; MUSM 4626, left upper molar; MUSM 4627, right upper molar; IGM 250283, IGM-DU 89–673, MUSM 4628–4631 (Fig 9X, 9B' and 9C'), left M1 or M2?s; MUSM 4632–4636 (Fig 9R, 9S and 9W), right M1 or M2?s; IGM 250284, MUSM 4637 (Fig 9T), left M3?s; IGM-DU 85–410, MUSM 4638, 4639 (Fig 9Y and 9D'), right M3?s.

## Derivation of name

Based on the original description of the species [37]: 'from the latin *paradoxalis*, in reference to the strange and confusing morphology of this species'. In light of the last discoveries of caviomorph faunas in Peru, the morphology of this species does not appear so strange or confusing, but we prefer maintaining this name for the sake of both usage and courtesy.

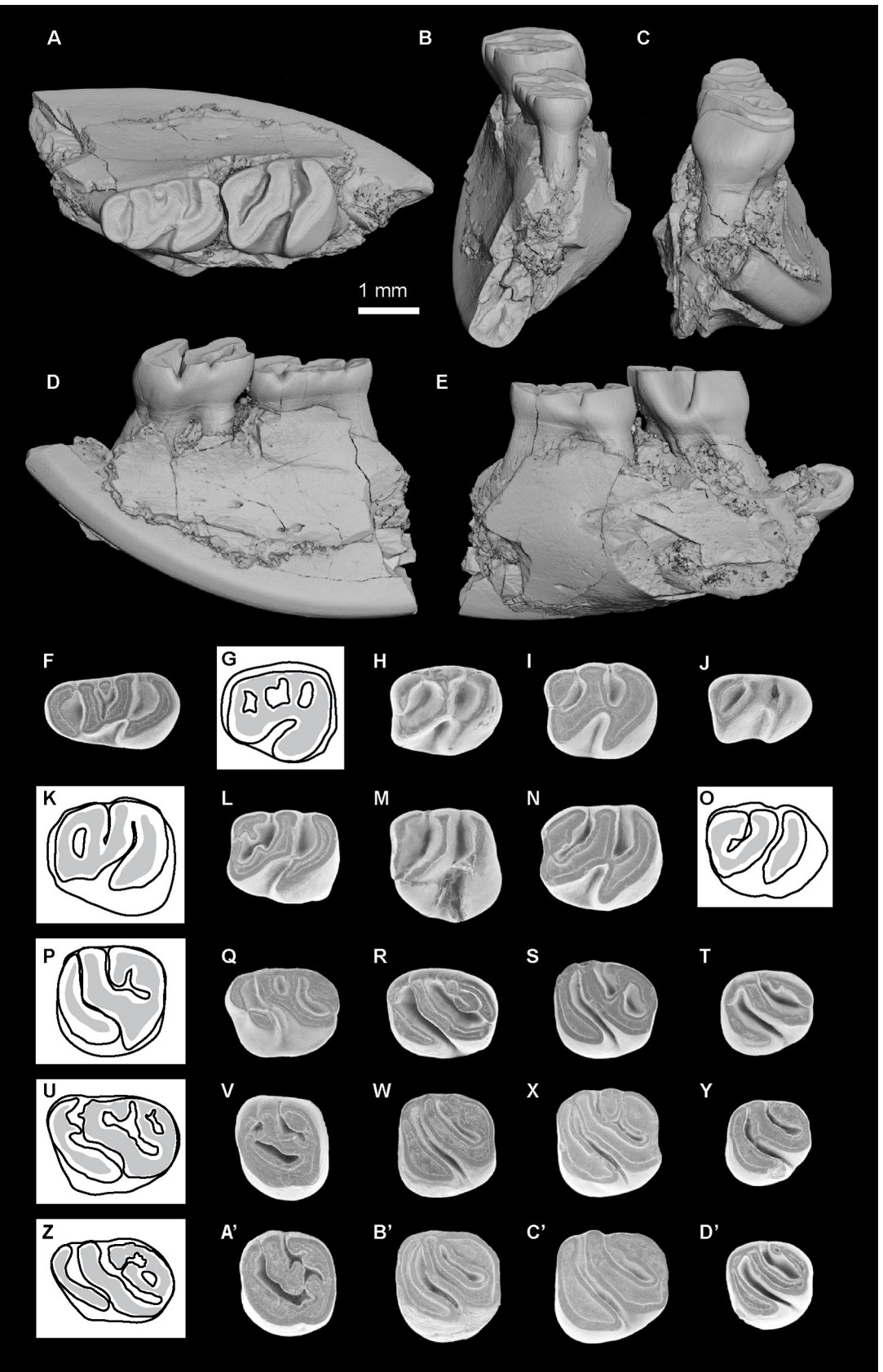

**Fig 9.** *Microscleromys paradoxalis* **gen. et sp. nov.** Mateial from TAR-31, Peru (black background) and from La Venta, Colombia (white background). (A–E) MUSM 4643, left mandibular fragment bearing dp4–m1. (F) MUSM 4641, left dp4. (G) IGM 250306, left p4. (H) MUSM 4647, left m1 or m2. (I) MUSM 4648, left m1 or m2. (J) MUSM 4657, m3. (K) IGM 250308, right p4 (reversed). (L) MUSM 4654, right m1 or m2 (reversed). (M) MUSM 4650, left m1 or m2. (N) MUSM 4649, left m1 or m2. (O) IGM 251020, right m3 (reversed). (P) IGM 250321\*\*\*, right M1 or M2? (reversed). (Q) MUSM 4623, left dP4. (R) MUSM 4634, right M1 or M2? (reversed). (S) MUSM 4633, right M1 or M2? (reversed). (T) MUSM 4637, left M3?. (U) ING-KU unnumbered specimen, left dP4. (V) MUSM 4625, right P4 (reversed). (W) MUSM 4632, right M1 or M2? (reversed). (X) MUSM 4631, left M1 or M2?. (Y) MUSM 4638, right M3? (reversed). (Z) IGM-DU 88–815, left dP4. (A') MUSM 4624, left P4. (B') MUSM 4630, left M1 or M2?. (C') MUSM 4629, left M1 or M2?. (D') MUSM 4639, right M3? (reversed). (A, F-D') Occlusal, (B) anterior, (C) posterior, (D) lingual, and (E) labial views. The asterisks appoint the holotype material. The illustrations of the fossil specimens are 3D digital models obtained by X-ray µCT surface reconstruction (A-E), scanning electron photomicrographs (F, H-J, L-N, Q-T, V-Y, and A'-D'), and computerized schematic line drawings based on Walton (Fig 24.2 [18]; Fig 13 [37]) (G, K, O, P, U, and Z).

## Type locality

CVP 13B locality, Fish Bed near Casa Cuzco, La Venta Badlands, upper Magdalena Valley, Huila Department, Colombia.

## Other localities

localities unknown (for ING-KU unnumbered specimen and ING-KU 8604), IGM-DU loc. 075, between the Chunchullo and the Tatcoa Sandstone Beds (only for IGM 250265), and localities of the Fish Bed (including CVP 5, 8, 9, 10, and 10A) and IGM-DU loc. 006–1, Monkey Beds (only for IGM-DU 88–814), La Venta Badlands, upper Magdalena Valley, Huila Department, Colombia; TAR-31, Tarapoto/Juan Guerra, Río Mayo, Shapaja road, San Martín Department, Western Amazonia, Peru.

## Formation and age

La Victoria Fm. and Baraya Member of the Villavieja Fm.; lower member, Ipururo Fm.; late middle Miocene (i.e., Laventan SALMA).

## Diagnosis

Species with teeth approximately 25–30 per cent larger than those of *M. cribriphilus*. dP4s can have a pseudo-taeniodont or taeniodont pattern.

## Description

MUSM 4643 is a right fragment of mandible bearing a part of the lower incisor and dp4–m1 (Fig 9A–9E). The incisor is mesially and distally broken, but the preserved portion testifies to a long radius of curvature of this tooth. The preserved part of the dentary corresponds to the ventrally broken body below the dp4–m1. It is very damaged, showing the base of the roots of the cheek teeth and the entire surface of the preserved fragment of incisor. The anterior root of dp4 is mesiodistally large, perpendicular to the plan of the neck, but clearly oriented mesioventrally in the dentary. Conversely, the posterior root of m1 is slightly oriented distoventrally. The morphology of the notch for the insertion of the tendon of the *zygomatico-mandibularis pars infraorbitalis*, masseteric crest or lateral crest is not accessible.

In addition to the dp4 preserved on MUSM 4643 (Fig 9A), three other dp4s are attributed to *Microscleromys paradoxalis* (Fig 9F). MUSM 4641 is complete (Fig 9F), whereas MUSM 4642 is a posterior fragment of dp4, in which the metaconid and metalophulid I are missing. MUSM 4640 is worn and abraded especially at the level of the metalophulid I, which is reduced to its distal part. The two complete dp4s are longer than wide, with a trigonid narrower than

the talonid. The metalophulid I is distally well arced, forming the mesial edge of the tooth. It is separate from the metaconid and protoconid by narrow notches on the MUSM 4643 dp4 (Fig 9A). On MUSM 4641 (Fig 9F), a similar configuration is present for its lingual extremity, whereas its labial extremity reaches the base of the protoconid. A continuous second cristid links the metaconid to the protoconid. Hence, it would be formed by a metaconid cristid and/ or posterior arm of the protoconid. It is mesiolingually directed and faintly curved on MUSM 4641 (Fig 9F) and MUSM 4643 dp4 (Fig 9A). It appears more transverse on MUSM 4640. On the MUSM 4643 dp4, MUSM 4641 and 4642 (Fig 9A and 9F), a mesostylid is present on the lingual margin. It can be mesially and distally isolated (MUSM 4641; Fig 9F), mesially isolated (MUSM 4643 dp4; Fig 9A) or connected to both the metaconid and entoconid (MUSM 4642). From that stylid, a neomesolophid runs labially and connects to a mesolophid reduced to a low spur. The two complete dp4s show neither neomesolophid nor mesolophid. The mesial and distal ectolophids are distolingually directed and aligned. They are not discernable from each other on the MUSM 4640, 4642 and 4643 dp4 (Fig 9A). On MUSM 4641 (Fig 9F), they are constituted of a short and thin mesial part (?mesial ectolophid) connecting to a longer and thicker distal part (?distal ectolophid). The labial part of the hypolophid is slightly oblique (i.e., distolingually directed), whereas its lingual part is more transverse. At early stages of wear (MUSM 4641; Fig 9F), the hypoconid displays a very short anterior outgrowth, which becomes longer with wear (MUSM 4642 and 4643 dp4; Fig 9A). The dp4s at advanced stages of wear have a non-taeniodont pattern, whereas MUSM 4641 shows a low and thin anterior arm of the hypoconid (i.e., pseudo-taeniodont pattern). The posterolophid is strong, well curved and separate from the entoconid by a large furrow. The hypoflexid is V-shaped with an anterior margin longer than the distal one.

The p4s are slightly longer than wide with the trigonid smaller than the talonid (Fig 9G and 9K). They are tetralophodont. Due to the moderate-advanced stage of wear of all p4s, the conids and stylids are not discernable from the lophids. The mesial margin of p4s is constituted by a transverse metalophulid I. Although it cannot be confirmed, a mesostylid may be present because the area of the posterior arm of the metaconid is long, especially on IGM 250306 (Fig 9G) and 250308 (Fig 9K), and could be fused with this stylid distally. The second cristid is complete on IGM 250306 (Fig 9G) and 250308 (Fig 9K), transverse on the former and mesiolingually directed on the latter. Moreover, IGM 250306 has a tiny spur on its lingual margin between the second cristid and the entonconid (Fig 9G). This spur could be a very short neomesolophid. On IGM 250305, the second cristid is reduced to a lingual cristid, labially free. It woud be a metaconid cristid or a neomesolophid depending on if it is stemming from the metaconid or the mesostylid, respectively. The metaconid area is thick on the mesiolingual corner of the p4 of the mandibular fragment ING-KU 8604, and could include a metaconid cristid. On that tooth, the protoconid has a very short posterior arm, and posteriorly to the metaconid area, there is a spur similar to that of IGM 250306 (Fig 9G), which is probably a rest of neomesolophid. The posterior arm of the metaconid (or mesostylid) is slightly connected to the entoconid on IGM 250305, 250306 (Fig 9G) and 250308 (Fig 9K). It is strongly linked to this cuspid on the most worn p4 (ING-KU 8604 p4). The ectolophid is oblique and connected mesially to the protoconid on all p4s. It connects distally to the hypolophid on IGM 250305, 250306 (Fig 9G) and 250308 (Fig 9K). Contrary to the other specimens, the ING-KU 8604 p4 shows the labial extremity of the hypolophid linked to the posterolophid instead of the ectolophid. The hypolophid is roughly transverse on all p4s. The hypoconid is fused with its anterior outgrowth, and the posterolophid is curved, with its lingual extremity reaching the entoconid. The anterior arm of the hypoconid is strong and complete on all p4s, except on IGM 250308 in which this arm is absent and the hypoflexid is fully confluent with the metaflexid (Fig 9K).

Lower molars are brachydont (Hling = 0.47–0.57; Hlab = 0.49–0.72; Fig 9A, 9H–9J and 9L–9O). The talonid of m3s is slightly narrower than the trigonid and with a distal part pinched labiolingually. The cuspids are very cristiform and thus, faintly differentiated from the cristids. At early stages of wear, the lower molars display a groove at the level of the metalophulid I separating it into an anterior arm of the metaconid and an anterior arm of the protoconid. The position of this groove varies: either in the middle part of the metalophulid I (MUSM 4647 and 4648; Fig 9H and 9I), lingual to the protoconid (IGM 250319, IGM-DU 88–814, 89–675, MUSM 4650 and 4655; Fig 9M), or closer to the metaconid (MUSM 4654; Fig 9L). On some other lower molars, the anterior arms of the metaconid and protoconid are slightly linked, either equally-sized (MUSM 4649–4653; Fig 9N) or the anterior arm of the protoconid can be shorter (MUSM 4657; Fig 9J). The posterior arm of the metaconid is strong and long (MUSM 4647, 4650, and 4655; Fig 9H and 9M), except on MUSM 4654 in which it is short (Fig 9L). On that tooth, there is a short posterior arm of the protoconid and metaconid cristid, which are both labially free. IGM 250304 has a metaconid cristid, which runs distolabially and remains free. On some lower molars at an advanced stage of wear (e.g., IGM 250274, 250319, 251020, IGM-DU 88–814, 89–675, ING-KU 8604, MUSM 4652, and 4653; Fig 9O), there is a dentine plateform at their mesiolingual corner, formed by the fusion of the metaconid, its arms, and possibly a metaconid cristid (+ ?neomesolophid + ?mesostylid). On MUSM 4644 and 4646, the ectolophid shows a small expansion, which could be a relic of the posterior arm of the protoconid. The ectolophid is oblique, aligned with the crestiform protoconid in most cases, and sometimes with the labial part of the hypolophid, its lingual part being more transverse (IGM-DU 88–814, 89–675, IGM 250319, 251020, and MUSM 4657; Fig 9J and 9O). The entoconid occupies a mesial position with respect to the hypoconid. The former usually has long and thick anterior and posterior arms, which are separated mesially to the posterior arm of the metaconid and distally to the posterolophid by tiny furrows (MUSM 4647, 4652, and 4653; Fig 9H). On MUSM 4650, the entoconid, having shorter and thinner arms, is mesially and distally isolated from these structures by larger notches (Fig 9M). The anterior arm of the hypoconid is either incomplete (MUSM 4654 and 4657; Fig 9J and 9L), notched (MUSM 4651 and 4653) or absent (e.g., IGM 250274, 250285, 251020, ING-KU 8604, MUSM 4643, and 4655; Fig 9A and 9O). On some lower molars at advanced stage of wear, the anterior arm of the hypoconid is, however, connected to the hypolophid-ectolophid junction (i.e., non-taeniodont pattern). The anterior outgrowth of the hypoconid is thin and becomes longer with wear. In addition, the hypoconid is strongly connected to the posterolophid. The latter is thick and C-shaped or hook-shaped.

The dP4s have a suboval/subtrapezoidal occlusal outline, with rounded corners (Fig 9Q, 9U and 9Z). The cusps are quite bulbous. Mesially, the anteroloph is curved and separate labially from the paracone. Lingually, it is strongly (IGM-DU 88–815 and ING-KU unnumbered specimen; Fig 9U and 9Z) or slightly (IGM 250302 and MUSM 4623; Fig 9Q) connected to a massive protocone or its anterior arm. The posterior outgrowth of the protocone is very short. On IGM 250302 and ING-KU unnumbered specimen (Fig 9U), a short crestule (i.e., neoformation) runs distally from the anteroloph and remains free. On all dP4s, the lingual protoloph is absent (i.e., taeniodont pattern). The labial protoloph is strong and either transverse (IGM 250302, ING-KU unnumbered specimen, and MUSM 4623; Fig 9Q and 9U) or oblique (IGM-DU 88–815; Fig 9Z). The mure is complete, oblique, and tends to be aligned with the anterior arm of the hypocone. On the labial margin, the paracone and mesostyle have a similar size, whereas the metacone remains poorly distinct (not well-defined) as it is fused with the curved posterolophid. The mesostyle is connected to the metacone solely (or its anterior arm) on IGM-DU 88–815 (Fig 9Z), whereas it joins the paracone and metacone or its anterior arm on ING-KU unnumbered specimen and MUSM 4623 (Fig 9Q and 9U). On IGM 250302, the

mesostyle is mesially and distally isolated. The third transverse crest corresponds to a meso-loph, which is lingually free (IGM 250302) or slightly connected to a mesolophular spur stemming from the anterior arm of the hypocone (MUSM 4623; Fig 9Q), or directly to the anterior arm of the hypocone (ING-KU unnumbered specimen and IGM-DU 88–815; Fig 9U and 9Z). Most of dP4s have a short metaloph backwardly connected to the posteroloph via a longer pos-teroloph spur or not, delimiting a small and oval metafossette. However, it can be more slightly connected to the posteroloph spur (MUSM 4623; Fig 9Q). On IGM-DU 88–815, the metacone does not seem to bear a metaloph (Fig 9Z). On that tooth, a long crest slightly linked to the posteroloph in its lingual part is present, which could be a rest of metaloph and/or the postero-loph spur. In addition, there is a tiny crestule stemming from the labial part of the posteroloph, which connects to the labial extremity of the metaloph/posteroloph spur, thus delimiting a small and oval fossette on IGM-DU 88–815 (Fig 9Z).

MUSM 4624 (Fig 9A') and 4625 (Fig 9V) are two complete P4s at moderate stages of wear. They preserve a part of their roots: a lingual one very massive and two labial much smaller, which externally diverge. These two specimens are oval and high-crowned, especially at the level of their lingual margin, their labial margin being lower (Hling = 0.80–0.87; Hlab = 0.62–0.70). On the labial margin, the paracone is large and connected to a strong and oblique labial protoloph. On MUSM 4625 (Fig 9V), the mure strongly links the labial protoloph to the short anterior arm of the hypocone. On that tooth, the mure is aligned with the anterior arm of the hypocone. On MUSM 4624 (Fig 9A'), the mure is slightly connected to the anterior arm of the hypocone by a thin and short crestule. In addition to the paracone, the rest of the occlusal out-line is composed of a curved and long crest formed by the fusion of the anteroloph, protocone, its anterior arm and its posterior outgrowth, the mesostyle, posteroloph, and hypocone. On MUSM 4624, the paracone is mesially and distally separated from this long crest by a thin fur-row (Fig 9A'), while the anteroloph reaches this cusp on MUSM 4625 (Fig 9V). On MUSM 4625, there are two connected crestules, one stemming from the anteroloph and the other from the labial protoloph (Fig 9V). The lingual protoloph is absent (i.e., taeniodont pattern). On MUSM 4624, the mesostyle is smaller than the paracone, whithout any mesoloph, and it joins the posteroloph (Fig 9A'). On that tooth, a metaloph stemming from a metacone entirely subsumed within the posteroloph, runs mesiolingually. Its lingual extremity is distally con-nected to a spur of the posteroloph, and ends close to a longer mesolophule, without connec-tion to the latter. On MUSM 4625, the posteroloph exhibits a thickening in its labial part, which may correspond to its fusion with the mesostyle/mesoloph and/or metacone/metaloph (Fig 9V).

The upper molars determined as M1 or M2?s are clearly mesiodistally elongated (IGM 250283, MUSM 4633, and 4634; (Fig 9R and 9S) or subsquare (IGM 250321, IGM-DU 89–673, MUSM 4629–4632, and 4635; Fig 9W, 9X, 9B' and 9C'). It seems that these upper molars show a tendency of shape variation, ranging from mesiodistally elongated at early stages of wear to squarer configuration at later stages of wear. On these upper molars, the hypocone is either more labially situated than the protocone, or situated at the same level of this cusp. Other upper molars have a hypocone clearly positioned labially with respect to the protocone, and thus they could be M3s (determined as M3?s). All upper molars have a taeniodont pattern characterised by a paraflexus and hypoflexus fully confluent. On the two upper molars at the earliest stage of wear (MUSM 4634 and 4637; Fig 9R and 9T), there is a short anteroloph (+ parastyle?), which is separate from the long and curved anterior arm of the protocone by a thin (MUSM 4634; Fig 9R) or wider (MUSM 4637; Fig 9T) furrow. The anteroloph is either not connected to the paracone (e.g., IGM-DU 89–673, MUSM 4629, 4632, 4634; Fig 9R, 9W and 9C'), slightly (MUSM 4633, 4638; Fig 9S and 9Y) or more strongly (MUSM 4637, 4639; Fig 9T and 9D') connected to this cuspid. The strong labial protoloph is usually oblique but it

can also be transverse (IGM 250321 and MUSM 4629; Fig 9C'). The anterior arm of the hypocone is oriented mesiolabially, except on MUSM 4634, in which it is more longitudinal. It tends to be aligned with a short mure. The labial protoloph can be in continuity to both the mure and anterior arm of the hypocone with roughly the same orientation (MUSM 4632, 4635, 4637, and 4639; Fig 9T, 9W and 9D'). On MUSM 4635, the labial protoloph bears on its mesial part a short crestule, which remains free on the paraflexus. It likely corresponds to a neocrest. Some upper molars have a well-defined mesostyle (MUSM 4626, 4632, 4634, 4637, and 4639; Fig 9R, 9T, 9W and 9D'). At early stages of wear, a thin and shallow notch separates the paracone from the mesostyle. A mesoloph, stemming from this style, is usually presents with a medium or short length, but it can also be more reduced as on MUSM 4637, in which it is incipient and poorly distinct from the mesostyle (Fig 9T). It can be transverse, either lingually free (IGM 250321 and MUSM 4637; Fig 9T), connected to a short mesolophule (MUSM 4633; Fig 9S), or directly connected to the anterior arm of the hypocone (MUSM 4631; Fig 9X). Some upper molars (IGM 250283, 250284, IGM-DU 89–673, MUSM 4630, and 4634; Fig 9R and 9B') have a mesoloph backarldly directed and linked to the posteroloph (or its spur), thereby delimiting an oval and small posterior fossette. On MUSM 4626, 4632, and 4639 (Fig 9W and 9D'), the third crest is absent (neither mesoloph nor mesolophule). There is no clear metacone on all upper molars, but a metaloph connected lingually to the posteroloph is present on MUSM 4631 (Fig 9X). It is faintly linked to the posterior arm of the mesostyle on that tooth. One upper molar at an early stage of wear, MUSM 4639 (Fig 9D'), has a trilophodont pattern without neither third crest nor metaloph. At more advanced stages of wear, some molars are trilophodont but with a thick posteroloph, which may include a mesoloph and/or a metaloph and their associated cusps/styles (MUSM 4627–4629, 4632, 4635, and 4638; Fig 9W, 9Y and 9C').

*Microscleromys cribriphilus* Boivin & Walton sp. nov. urn:lsid:zoobank.org:act:74B60D5-F-A8B3-4A2C-9483-D22739400EC8

Fig 10 and S1 Table

*Nom. nud. Microscleromys cribriphilus* Walton, 1990, Fig 12A-12E, p. 45.

*Nom. nud. Microscleromys cribriphilus* Walton, 1997, Fig 24.2.K-M,O, p. 295.

*Nom. nud. Microscleromys paradoxalis* Walton, 1997, Fig 24.2.J, p. 295.

### Holotype

MUSM 4491, left M1 or M2? (Fig 10H').

### Referred material

MUSM 4545, right fragmentary dp4 or p4; MUSM 4532–4541 (Fig 10A, 10F and 10K), left dp4s; MUSM 4542–4544, right dp4s; IGM-DU 88–818, MUSM 4546–4550 (Fig 10G, 10L and 10P), left p4s; IGM 251018 (Fig 10Q), MUSM 4551–4558 (Fig 10B), right p4s; IGM-DU 89–676, left m1 or m2 (or p4); MUSM 4558–4561, left lower molars; MUSM 4562–4565, right lower molars; MUSM 4566, left m1 or m2?; IGM 250309, right m1 or m2?; MUSM 4567–4588 (Fig 10H and 10M), left m1?s; MUSM 4589–4600 (Fig 10C), right m1?s; MUSM 4601–4608 (Fig 10D, 10I, 10N and 10S), left m2?s; IGM 250303 (Fig 10R), MUSM 4609–4615, right m2s; MUSM 4621, left m3?; MUSM 4616–4620 (Fig 10E, 10O and 10T), left m3s; MUSM 4622 (Fig 10J), right m3; MUSM 4420–4433 (Fig 10U and 10E'), left dP4s; IGM 250320 (Fig 10J'),

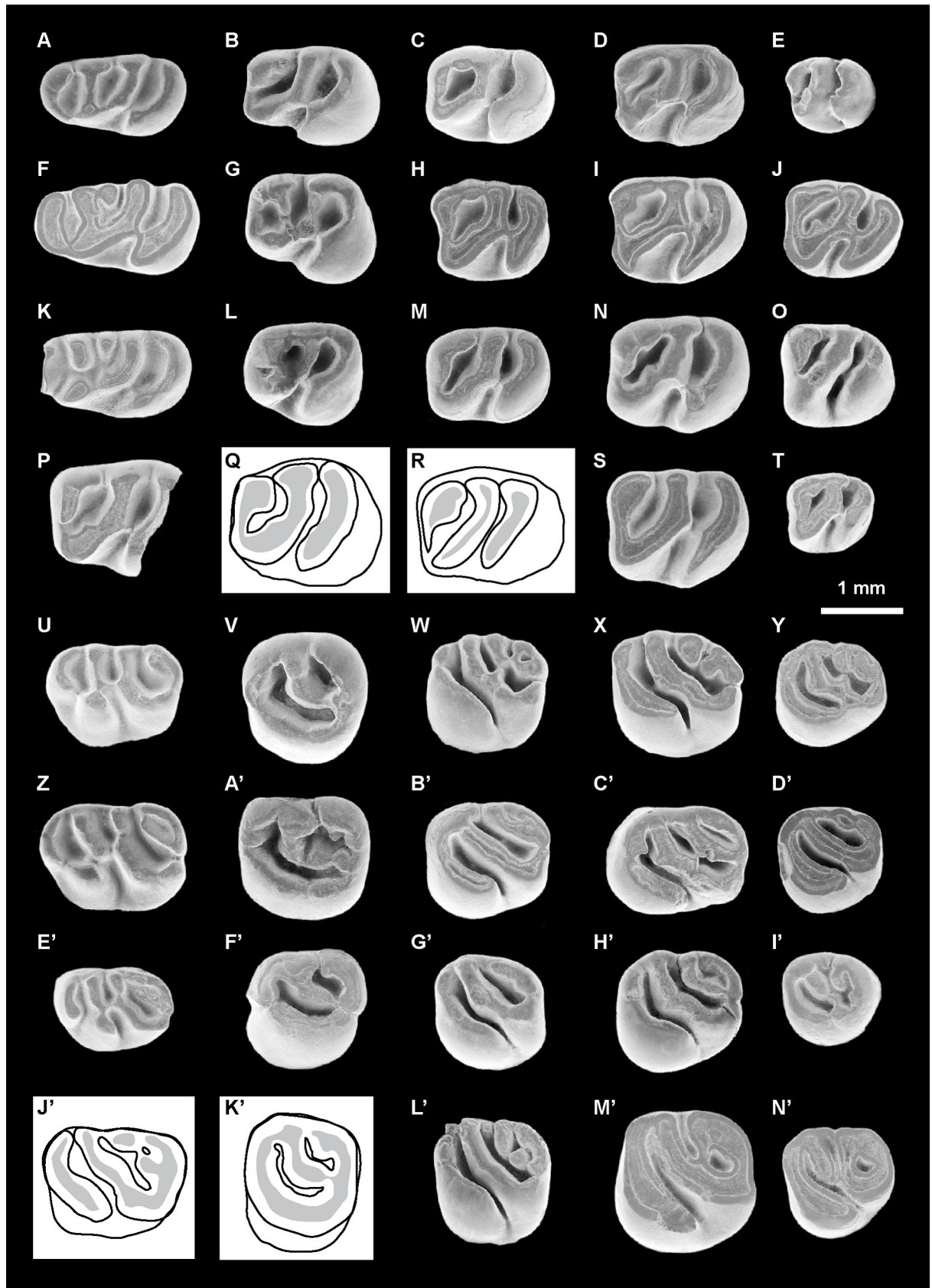

**Fig 10.** *Microscleromys cribriphilus* **gen. et sp. nov.** Material from TAR-31, Peru (black background) and from La Venta, Colombia (white background). (A) MUSM 4532, left dp4. (B) MUSM 4553, right p4 (reversed). (C) MUSM 4599, right m1? (reversed). (D) MUSM 4603, left m2?. (E) MUSM 4619, left m3. (F) MUSM 4540, left dp4. (G) MUSM 4548, left p4. (H) MUSM 4581, left m1?. (I) MUSM 4606, left m2?. (J) MUSM 4622, right m3 (reversed). (K) MUSM 4536, left dp4. (L) MUSM 4546, left p4. (M) MUSM 4583, left m1?. (N) MUSM 4605, left m2?. (O) MUSM 4616, left m3. (P) MUSM 4550, left p4. (Q) IGM 251018, right p4 (reversed). (R) IGM 250303, right m2? (reversed). (S) MUSM 4602, left m2?. (T) MUSM 4620, left m3. (U) MUSM 4425, left dP4. (V) MUSM 4453, left P4. (W) MUSM 4505, right M1 or M2? (reversed). (X) MUSM 4514, right M2? (reversed). (Y) MUSM 4521, left M3?. (Z) MUSM 4445, right dP4 (reversed). (A') MUSM 4466, right P4 (reversed). (B') MUSM 4489, left M1 or M2?. (C') MUSM 4511, left M2?. (D') MUSM 4530, right M3? (reversed). (E') MUSM 4428, left dP4. (F') MUSM 4452, left P4. (G') MUSM 4502, right M1 or M2? (reversed). (H') MUSM 4491***, left M1 or M2?. (I') MUSM 4525, right M3? (reversed). (J') IGM 250320, right dP4 (reversed). (K') IGM 251040, right P4 (reversed). (L') MUSM 4508, right M1 or M2? (reversed). (M') MUSM 4513, left M2?. (N') MUSM 4517, left M3?. Occlusal views. The asterisks appoint the holotype material. The illustrations of the fossil specimens are scanning electron photomicrographs (A-P, S-I', and L'-N') and computerized schematic line drawings based on Walton (Fig 24.2 [18]; Fig 14 [37]) (Q, R, J', and K').

MUSM 4434–4447 (Fig 10Z), right dP4s; MUSM 4469, right P4 or M3?; MUSM 4448–4458 (Fig 10V and 10F'), left P4s; IGM 251040 (Fig 10K'), MUSM 4459–4468 (Fig 10A'), right P4s; MUSM 4470, 4471, left upper molars; MUSM 4472–4478, right upper molars; MUSM 4479–4497 (Fig 10B' and 10H'), left M1 or M2?s; MUSM 4498–4510 (Fig 10W, 10G' and 10L'), right M1 or M2?s; MUSM 4511–4513 (Fig 10C' and 10M'), left M2?s; MUSM 4514–4516 (Fig 10X), right M2?s; MUSM 4517–4524 (Fig 10Y and 10N'), left M3?s; MUSM 4525–4531 (Fig 10D' and 10I'), right M3?s.

### Derivation of name

Based on the original description of the species [37]: from the latin *cribri* (sieve) and *philus* (liking to or attracted to) 'because this rodent is so far only known from isolated teeth recovered by screenwashing'.

### Type locality

TAR-31, Tarapoto/Juan Guerra, Mayo River, Shapaja road, San Martín Department, Western Amazonia, Peru.

### Other localities

Localities of the Fish Bed (including CVP 9, 10A, and 13B), La Venta Badlands, upper Magdalena Valley, Huila Department, Colombia.

### Formation and age

Lower member, Ipururo Fm.; Baraya Member of the Villavieja Fm.; late middle Miocene (i.e., Laventan SALMA).

### Diagnosis

Species with teeth approximately 25–30 per cent smaller than those of *M. paradoxalis*. dP4s can have a non-taeniodont pattern characterised by a complete and high lingual protoloph in continuity with the labial protoloph.

### Description

The pattern of teeth usually follows that of *M. paradoxalis* described above, with few exceptions, but mostly the smaller size allows for a clear distinction of the two species (see S1 File).

dp4s are longer than wide, with a trigonid narrower than the talonid. The protoconid appears to be the largest cuspid. On the dp4 at the earliest stage of wear (MUSM 4532; Fig

10A), the metaconid is the most cristiform cuspid, which is linguolabially pinched. On worn dp4s (MUSM 4539, 4540, and 4544; Fig 10F), the protoconid displays three connections: (i) a mesiolingual one with a curved metalophulid I, which extends lingually to the metaconid; (ii) a distolingual one with the second transverse cristid, which connects labially to the metaconid; and (iii) a distal one with the mesial extremity of the mesial ectolophid. On the dp4s at the earliest stage of wear (MUSM 4532 and 4536; Fig 10A and 10K), the protoconid is separated from the metalophulid I and from the mesial ectolophid by two notches. The notch between the protoconid and the metalophulid I is the deepest and thus, it closes at a latter stage of wear than that between the protoconid and the mesial ectolophid (MUSM 4534). Conservely to MUSM 4532, the protoconid is also isolated from the second transverse cristid, which is reduced to a medium-sized metaconid cristid, on MUSM 4536 (Fig 10K). However, both structures are connected at their base. On MUSM 4532, 4534, and 4536 (Fig 10K), the metalophulid I is not connected to the metaconid (Fig 10A). On all dp4s, the second transverse cristid is slightly directed mesiolingually. The mesial and distal ectolophids are thin, distolingually directed, aligned and often not discernable from each other. A small mesostylid is often present (MUSM 4534, 4536, 4539, 4540, and 4541; Fig 10F and 10K). It is isolated from the entoconid on all dp4s. It is connected to the metaconid on MUSM 4534, 4539 and 4540 (Fig 10F). On MUSM 4532 (Fig 10A), a mesostyle may be present, fused with the long posterior arm of the metaconid. On MUSM 4539, a complete third transverse cristid links the mesostylid with the ectolophids. The other dp4 do not show any development of the third cristid. On all dp4s, a well-marked hypolophid links the large entoconid to the distal ectolophid. In most cases, the labial part of the hypolophid is slightly oblique (i.e., distolingually directed), whereas its lingual part is more transverse. The hypoconid always displays a short anterior outgrowth and a low, but complete anterior arm, reaching the hypolophid-distal ectolophid junction (i.e., pseudo-taeniodont pattern). Besides, the hypoconid has lingually a very short posterior arm, connecting to a strong and well-curved posterolophid on MUSM 4532 and 4534 (Fig 10A). At early stages of wear, the posterolophid is separate from the entoconid or its posterior arm by a large furrow. The three lingual flexids show a similar large expansion. The hypoflexid is less extended, distolingually oriented and V- or U- shaped. Its anterior margin is longer than its distal one, except on MUSM 4533.

p4s are brachydont-submesodont (Hling = 0.45–0.77; Hlab = 0.54–0.81; Fig 10B, 10G, 10L, 10P and 10Q). They are slightly longer than wide, with a talonid clearly wider than the trigonid. By their high elevation and large surface, the protoconid and hypoconid are the most distinct cuspids of the crown at the earliest stage of wear (MUSM 4548; Fig 10G). The protoconid is labiolingually pinched, whereas the hypoconid is mesiodistally compressed. At the mesiolingual corner, the metaconid is well defined during the first stages of wear (MUSM 4548 and 4553; Fig 10B and 10G). It displays a short or moderate posterior arm (MUSM 4548, 4550, and 4553; Fig 10B, 10G and 10P). On MUSM 4548 (Fig 10G), the metalophulid I is continuous and high, joining the protoconid to the metaconid. On MUSM 4546 and 4553 (Fig 10B), the anterior arms of the protoconid and metaconid are short, strongly sloped and separated by a large furrow, which is thinner on MUSM 4547 and 4550 (Fig 10P). The protoconid frequently has a short posterior arm, which can be very short (MUSM 4546–4548; Fig 10G and 10L) or reduced to a spur (IGM-DU 88–818, MUSM 4550, 4551, and 4557; Fig 10P). On IGM-DU 88–818, MUSM 4547 and 4557, it is lingually free. The posterior arm of the protoconid is absent on IGM 251018, MUSM 4553 and 4558 (Fig 10B). A thin and short metaconid cristid can be present (MUSM 4546 and 4548; Fig 10G and 10L). On MUSM 4546 (Fig 10L), it is slightly connected to the posterior arm of the protoconid. On MUSM 4548 (Fig 10G), these two cristids are linked at their base but not connected at their apex. The worn MUSM 4554 p4 shows an oblique second transverse cristid, linking the metaconid to the protoconid. Thus, this cristid

may be formed here by the fusion of a metaconid cristid with a posterior arm of the protoconid. On MUSM 4551, a round and small mesostylid is clearly distinct, separated from the metaconid (or its posterior arm) and entoconid by tiny notches. On that tooth, a cristid, probably a neomesolophid, is faintly connected to the mesostylid and to a very tiny posterior arm of protoconid. In addition, it shares a very slight connection mesially with the mesialmost lophid and distally with the hypolophid. On MUSM 4546, the distal extremity of the posterior arm of the metaconid displays a tiny spur that may be a neomesolophid that would be connected to a mesostylid fused with the posterior arm of the metaconid (Fig 10L). At some advanced stage of wear, the metaconid, its anterior and posterior arms, metaconid cristid, as well as possibly the mesostylid and neomesolophid (if present), fuse and form a small platform of dentine on the mesiolingual corner of the teeth. If the entoconid has anterior and posterior arms, the latter is longer than the former (MUSM 4547, 4548, 4550, and 4556; Fig 10G and 10P), except on IGM 251018 and MUSM 4546, in which they are equally-sized (Fig 10L and 10Q). The entoconid (or its anterior arm) is separated from the mesostylid-posterior arm of the metaconid by a large furrow (MUSM 4548 and 4553; Fig 10B and 10G) or a very thin notch (MUSM 4546, 4547, 4550, and 4556; Fig 10L and 10P). On p4s at a more advanced stage of wear, these structures are connected. Stemming from the entoconid, a hypolophid runs labiomesially (i.e., oblique) and slightly connects to the ectolophid (IGM-DU 88–818, MUSM 4546, and 4548; Fig 10L). The anterior arm of the hypoconid usually reaches the hypolophid-ectolophid junction (i.e., non-taeniodont pattern). On p4s at early stages of wear (MUSM 4546, 4548, and 4553; Fig 10B, 10G and 10L), this cristid is more or less notched but remains elevated. However, on MUSM 4550 (Fig 10P), the notch of the anterior arm of the hypoconid is more pronounced, involving a partial confluence between the metaflexid and the retroverse hypoflexid (i.e., pseudo-taeniodont pattern). IGM 251018 appears to have a taeniodont pattern (Fig 10Q). The posterior arm of the hypoconid is fused with the strong and curved posterolophid, the latter extending lingually and ending directly distal to the entoconid, connecting to its base. The hypoconid displays a long and thin anterior outgrowth.

Lower molars are brachydont-mesodont (Hling = 0.50–0.79; Hlab = 0.54–0.91; Fig 10C–10E, 10H–10J, 10M–10O and 10R–10T). The m3s have a distal part of the talonid labiolingually pinched, and thus are more recognisable than m1s and m2s. Some lower molars appear smaller, and could be m1s (identified as m1?s). The m3s are either roughly equal in size to the lower molars determined as m1?s or smaller. Lower molars differ from p4s in having a wider trigonid comparatively to the talonid. The cristids are thick and incorporate in their ends the cuspids, which are thus faintly differentiated (except for the entoconid). The metalophulid I can show a thin interruption (groove) either in its middle part, separating equally-sized anterior arms of the metaconid and of the protoconid (MUSM 4611), or directly lingual to the protoconid, separating a long anterior arm of the metaconid from a very short and low anterior arm of the protoconid (MUSM 4588 and 4616; Fig 10O). On some other lower molars, both arms are slightly connected (e.g., IGM 250303, MUSM 4573, 4597, 4599, and 4620; Fig 10C, 10R and 10T). MUSM 4586 shows an additional cristid stemming from the metalophulid I, running distally and ending freely. The metaconid has a strong posterior arm, which can be long (e.g., MUSM 4592) or short (e.g., MUSM 4580, 4588, and 4599; Fig 10C). This cristid connects to a distinct mesostylid on some lower molars (MUSM 4580, 4582, 4584, 4588, 4597, 4599, 4601, and 4605; Fig 10C and 10N). A neomesolophid stemming from the mesostylid can be present (e.g., MUSM 4580, 4588, and 4605; Fig 10N) or absent (e.g., MUSM 4582, 4584, 4597, 4599, and 4601; Fig 10C). The neomesolophid is either limited to a very short enamel spur or short cristulid. It is labially free on all specimens. Two lower molars (MUSM 4597 and 4613) exhibit a short lingual cristulid from the metaconid, recognised as a metaconid cristid, which remains labially free. As p4s, some lower molars at an advanced stage of wear have a

mesiolingual dentine platform, formed by the fusion of the metaconid, its arms, and possibly the metaconid cristid and/or neomesolophid + mesostylid. On several lower molars (e.g., MUSM 4560, 4563, 4568, 4571, 4588, 4613, and 4622; Fig 10J), there is a small lingual expansion on the ectolophid that is likely to be a relic of the posterior arm of the protoconid. With the reduction or absence of the neomesolophid and the posterior arm of the protoconid, the mesoflexid is partly or entirely confluent with the anteroflexid. The ectolophid is oblique, frequently aligned with the crestiform protoconid and the hypolophid, thereby forming a long mesiocentral (diagonal) cristid. Lingually, the entoconid is the most differentiated cuspid. It is frequently mesiodistally elongated by developing strong anterior and posterior arms, which are particularly long on MUSM 4582. They tend to be mesially connected with wear to the mesostylid-posterior arm of the metaconid, and distally to the posterolophid. On lower molars at earliest stages of wear, thin furrows usually isolate the entoconid from these other structures mesially and distally. A short but strong hypolophid extends from the entoconid to the distal extremity of the ectolophid. As on p4s, the hypoconid on lower molars is mesiodistally compressed and displays a thin anterior outgrowth, which becomes long with wear. The hypoconid is distal in position to the entoconid. The anterior arm of the hypoconid can be present or absent (e.g., IGM 250303, MUSM 4600, and 4616; Fig 10O and 10R). When it is present, it is faintly developed and runs to the ectolophid–hypolophid junction. It is grooved (e.g., MUSM 4579, 4605, and 4620; Fig 10N and 10T) or deeply notched (e.g., MUSM 4569 and 4602; Fig 10S), showing a suite of intermediate sizes of the notch. Hence, the metaflexid is more or less confluent with the hypoflexid (i.e., from pseudo-taeniodont pattern to taeniodont pattern). However, many lower molars at an advanced stage of wear show a complete anterior arm of the hypoconid, which can be thick (i.e., non-taeniodont pattern; e.g., MUSM 4581, 4606, and 4622; Fig 10H–10J). A massive and hook-shaped posterolophid strongly connects to the hypoconid, both structures being fused and in perfect continuity.

dP4s are longer than wide and with rounded corners (Fig 10U, 10Z, 10E' and 10J'). They have a suboval occlusal outline, with a well-arcuate anteroloph (i.e., mesially convex) and posteroloph (i.e., distally convex). All dP4s are brachydont. The cusps are usually moderately inflated. The anteroloph is curved, short, and located far mesially. At early stages of wear, two notchs separate the anteroloph labially from the paracone (except on MUSM 4445; Fig 10Z) and lingually from the strong and bulbous protocone or its anterior arm. The first notch is wide, while the second is highly variable in terms of development. It corresponds to a very large notch on some dP4s (MUSM 4425, 4427, 4428, 4431, 4432, and 4435; Fig 10U and 10E'), whereas, on other teeth, it is a thinner and shallower furrow (MUSM 4429, 4439, 4440, and 4443) or a tiny groove (MUSM 4430, 4437, 4442, 4445, and 4446; Fig 10Z). On MUSM 4438, the protocone is mesiolingually-distolabially pinched, developing a short anterior arm, which is strongly connected to the anteroloph. The posterior outgrowth of the protocone is very short on the whole dP4s. On MUSM 4422 and 4428 (Fig 10E'), a well-cuspate parastyle is discernable on the mesiolabial corner of the tooth. This style bears short, but strong, anterior and posterior arms, separate by deep furrows from the anteroloph and paracone, respectively. On MUSM 4422, a medium-sized loph (i.e., neoformation) runs lingually from the parastyle, slightly directed distolingually, but with its lingual extremity faintly linked to the anteroloph via a very short and thin crestule. On MUSM 4428 (Fig 10E'), this loph is present but reduced to a short spur, whose base is connected to a peculiar anteroloph, short and labially concave. Hence, the parastyle, its anterior arm, its spur and the anteroloph delimit a small and round fossette. On the labial margin, the paracone and mesostyle are equally sized, and the metacone is smaller and usually still visible. The mesostyle is separate mesially from the paracone by a large furrow, that closes with wear. This style can be connected (MUSM 4422, 4431, and 4438) or disconnected (MUSM 4424, 4425, 4427, 4428, 4430, 4439, and 4445; Fig 10U, 10Z and 10E')

to the metacone (or its anterior arm). Posteriorly, the metacone is strongly connected to the well-curved posteroloph. On dP4s at early stages of wear, the lingual protoloph can be absent (i.e., taeniodont pattern; e.g., MUSM 4422, 4427, and 4428; Fig 10E') or present, either thin and low (i.e., pseudo-taeniodont pattern; MUSM 4437, 4438, and 4445; Fig 10Z) or high (i.e., non-taeniodont pattern; MUSM 4425, 4431, and 4446; Fig 10U). The labial protoloph is thick and usually straight, but it can also be slightly oblique. On IGM 250320, MUSM 4420, 4427, 4428, 4444, and 4445 (Fig 10Z, 10E' and 10J'), it is particularly strongly oblique. On MUSM 4425, 4429, 4431, 4432, 4439, 4440, 4442 and 4446, the labial and lingual protolophs are aligned and form a transverse crest linking the paracone to the protocone (Fig 10U). The mure is complete and high, except on MUSM 4429 in which it is low, involving a partial confluence of the mesial mesoflexus with the hypoflexus. It is usually longitudinal (i.e., parallel to the mesiodistal axis of the tooth), but can be more oblique, aligned with the anterior arm of the hypocone (IGM 250320, MUSM 4420, 4421, 4422, 4427, and 4443; Fig 10J'), and sometimes with an oblique protoloph (MUSM 4420 and 4427). The third transverse crest is parallel with the labial protoloph. It is composed of two crests: a short and thick mesoloph stemming from the mesostyle and a mesolophular spur stemming from the anterior arm of the hypocone. On some dP4s (MUSM 4425, 4438, and 4446; Fig 10U), the third crest is complete, with the mesoloph linked to a short mesolophule, whereas they can be unconnected or faintly connected together on other teeth (MUSM 4420, 4427, 4428 and 4443–4445; Fig 10Z and 10E'). On IGM 250320, MUSM 4428 and 4444 (Fig 10E' and 10J'), the mesoloph seems to be absent or strongly reduced. IGM 250320 appears to not have any mesolophule (Fig 10J'). Most of dP4s have a short metaloph backwardly connected to the posteroloph via a longer posteroloph spur or not, delimiting a small and oval metafossette. However, it can be lingually free (MUSM 4427) or absent (MUSM 4425, 4428, and 4431; Fig 10U and 10E'). On MUSM 4442, there is a short posteroloph spur and a cusp-like structure on the distalmost flexus, which is probably a relic of the metaloph. The latter structures are not linked. On dP4s at the earliest stages of wear, the posteroloph displays a shallow (MUSM 4425, 4430, 4431, 4442; Fig 10U) or deep (MUSM 4427, 4428, 4446; Fig 10E') furrow, which separates it from the hypocone.

P4s are high-crowned, especially at the level of their lingual crown, their labial crown being lower (Hling = 0.81–0.96; Hlab = 0.45–0.79; Fig 10V, 10A', 10F' and 10K'). In occlusal view, their crown is oval, labiolingually wider than longer. Mesiolingually, the anteroloph, protocone, its anterior arm and its posterior outgrowth form a long crest, which is separate from the large and well-cuspate paracone by a thin furrow. Distally, the paracone remains free and separate from a long distal crest including the mesostyle, posteroloph, hypocone and its long anterior outgrowth. Therefore, a usually wider and persistant furrow occurs distal to the paracone. The mesial extremity of the anterior outgrowth of the hypocone is connected to the posterior outgrowth of the protocone, and to the whole of the aforementioned structures, thereby forming most of the peripheral margin of the tooth. There is no lingual protoloph on the whole specimens. Accordingly, the most anterior flexus is a fossette, corresponding to the confluence of the paraflexus with the hypoflexus (i.e., taeniodont pattern). In mostcases, the mure and the short anterior arm of the hypocone are thin, linked, and aligned, forming a longitudinal or oblique crest. On MUSM 4450 and 4455, these two structures are not aligned, the mure being more longitudinal than the anterior arm of the hypocone. On MUSM 4456, the mure is not linked to the anterior arm of the hypocone or to the posteroloph. The mure is connected to the lingual extremity of the labial protoloph, which is almost transverse (IGM 251040, MUSM 4449, 4455, 4457, 4459, and 4466; Fig 10A' and 10K') or oblique (MUSM 4450–4454, 4456, 4461–4464, 4467, and 4468). The mure is absent on MUSM 4463, in which the labial protoloph is not connected to the anterior arm of the hypocone. On MUSM 4449, a longitudinal crest connects the anteroloph to the labial protoloph. Hence, these strutures and the paracone

delimits a very small rounded fossette. The mesostyle is small and crestiform, poorly distinct from the posteroloph. On MUSM 4452 and 4466 (Fig 10A' and 10F'), it displays a very short mesoloph lingually free, longer on MUSM 4466 (Fig 10A') than on MUSM 4452 (Fig 10F'). On P4s in a more advanced stage of wear (MUSM 4451, 4456, and 4468), the mesoloph is longer. On MUSM 4451, its lingual extremity is slightly connected to the mure-anterior arm of the hypocone and to the posteroloph. One P4, MUSM 4453 (Fig 10V), do not show any mesoloph. MUSM 4466 has a very short metalophular spur labially free, as well as a tiny crestule from the posteroloph, which could be a posteroloph spur or a metaloph (Fig 10A'). With wear, the posteroloph often exhibits a thickening in its labial part, which may correspond to its fusion with the mesostyle/mesoloph, metacone/metaloph and/or posteroloph spur (IGM 251040, MUSM 4450, 4455, 4458, 4459, 4463, and 4467; Fig 10K'). This plateform can be free, joined lingually to a tiny mesolophular spur (MUSM 4450) or directly to the mure (MUSM 4455).

The different loci documenting upper molars (i.e., M1 vs M2 vs M3) are not easily recognisable (Fig 10W–10Y, 10B'–10D', 10G'–10I' and 10L'–10N'). Some upper molars are small, with a rounded occlusal outline and a hypocone clearly labially positioned with respect to the protocone, and thus they could be M3s (determined as M3?s; Fig 10Y, 10D', 10I' and 10N'). Other upper molars are square, mesiodistally elongated or transverse. On these upper molars, the hypocone is either more labially situated than the protocone or situated at the level of this cusp (i.e., distal). As for P4s, upper molars are labiolingually asymmetrical with a higher lingual crown (Hling = 0.71–1.07; Hlab = 0.42–0.65). In general, the cusps/styles are crestiform, and not well defined. In most cases (MUSM 4487, 4493, 4502, 4508, 4511, 4525, and 4527; Fig 10C', 10G', 10I' and 10L'), the protocone shows a long and curved anterior arm, which is slightly connected to a shorter and straight anteroloph (+ ?parastyle). These two crests can be also strongly connected (MUSM 4489 and 4520; Fig 10B') or separated by a narrow (MUSM 4491; Fig 10H') or wide (MUSM 4505; Fig 10W) furrow. The anteroloph is connected to the mesial aspect of the paracone on some M1s or M2s (MUSM 4483, 4487, 4489, 4495, and 4502; Fig 10B' and 10G') and all M3?s (MUSM 4520, 4526, and 4527), whereas it remains separate from the paracone by a narrow furrow such as on MUSM 4482, 4486, 4491, 4494, 4499, 4505, 4513, and 4514 (Fig 10W, 10X, 10H' and 10M'). On the MUSM 4525 M3?, the paracone is fully fused with the anteroloph (Fig 10I'). On MUSM 4511, the anteroloph bears a short but well-marked crestule lingually free on the paraflexus (Fig 10C'). The lingual protoloph is absent on all upper molars, thereby the paraflexus and hypoflexus are confluent (i.e., taeniodont pattern). The protocone displays a long and strong posterior outgrowth that can end close to the hypocone (MUSM 4495, 4498, 4517, 4519, 4520, 4524, and 4527–4530; Fig 10D' and 10N'), or can connect to this cusp with wear (MUSM 4521; Fig 10Y). On all upper molars, the hypocone is usually smaller than the other main cusps. It displays a strong and oblique anterior arm, which connects to the mure and tends to be aligned with it. A labial protoloph, strong, short and slightly (e.g., MUSM 4486 and 4529) or strongly (e.g., MUSM 4483, 4489, and 4520; Fig 10B') oblique, links the paracone to the mure. The anterior arm of the hypocone and the mure are in continuity with the labial protoloph, forming a long and diagonal central crest with a strong obliquity on some upper teeth (e.g., MUSM 4481, 4485, 4489, 4493, 4520; Fig 10B'). The labial protoloph can bear a short crestule mesially free on the paraflexus (MUSM 4498 and 4511; Fig 10C'). Labially, a strong mesostyle is present. It is clearly distinct on some upper molars (MUSM 4482, 4505, 4514, 4516, 4527, and 4530; Fig 10W, 10X and 10D'). It is separated from the paracone by a thin and shallow notch (e.g., MUSM 4483, 4487, 4505, and 4529; Fig 10W) or connected to this cusp (or to the posterior arm of paracone; MUSM 4489, 4516, and 4520; Fig 10B'), thereby forming a labial wall that closes the mesial mesoflexus. Distolabially, a metacone is either well separate from the posterior arm of the

mesostyle by a large but shallow notch (MUSM 4505 and 4514; Fig 10W and 10X), faintly connected to the posterior arm of the mesostyle (MUSM 4482, 4487, 4508, and 4522; Fig 10L') or more strongly linked to it (MUSM 4516 and 4521; Fig 10Y). The upper molars show an important variation in the development, configuration and orientation of the third crest and metaloph. From the mesostyle, a strong but short mesoloph runs either (i) lingually: it is free (MUSM 4510, 4512, 4514, 4516, 4522, and 4525; Fig 10X and 10I') or connects to a short mesolophule (MUSM 4471, 4487, 4494, 4513, 4515, and 4520; Fig 10M'); or (ii) backwardly: it is free (MUSM 4483 and 4499), or connects to the metaloph (MUSM 4508 and 4521; Fig 10Y and 10L'), to the posteroloph (via a spur or not; MUSM 4481, 4489, 4491, 4501, 4504, 4511, and 4528; Fig 10B', 10C' and 10H'), or seemingly to both crests (MUSM 4482). On MUSM 4521, the mesolophule is not connected to the mesoloph (Fig 10Y). On MUSM 4505, the mesoloph is slightly joined to a tiny mesolophule as well as to the metaloph (Fig 10W). Some upper molars show a distinct metacone, but without metaloph (or almost absent; MUSM 4491; Fig 10H'). If the metaloph is present and not linked to the mesoloph and/or to the posteroloph (via a spur or not; MUSM 4482, 4505, 4508, and 4521; Fig 10W, 10Y and 10L'), it is lingually free (MUSM 4487, 4514, 4516, and 4522; Fig 10X). On MUSM 4502, 4529 and 4530 (Fig 10D' and 10G'), there is neither the third crest nor the metaloph (i.e., trilophodont pattern). Most of molars at more advanced stages of wear are trilophodont (MUSM 4470, 4476, 4481, 4484, 4492, 4495, 4496, 4498, 4518, 4519, and 4528), but have a enlarged posteroloph, which may include a mesoloph and/or a metaloph and their associated cusps/styles (entirely subsumed within the posteroloph). The flexi are furrow-shaped, especially the anterior flexus (i.e., paraflexus + hypoflexus).

## Remark

The specimens attributed to *Microscleromys cribriphilus* show a great variation of size. For example, the size range of the lower molars extends from 1.08 mm to 1.78 mm for the length, and from 1.01 mm to 1.57 mm for the width. Likewise, the size range of upper molars is 1.17–1.71 mm for the length and 1.07–1.66 mm for the width. Due to the sample size that is distributed quite uniformly for the minimum range and without shape variation, we consider that all the material reported here documents a single taxon. Accordingly, we consider the smaller specimens to document smaller individuals. Moreover, it is worth noting that the smallest specimens are often teeth determined as m3?s and M3?s. We attribute the observed size variation between specimens partly resulting of size variation between loci.

## Common comparisons

Basically, among caviomorph rodents, (i) the presence of high-crowned teeth with a taeniodont pattern and oblique loph(-id)s, (ii) the absence or reduction of second transverse cristid on lower molars, and (iii) the absence of metaloph and third transverse crest, or the reduction of the latter (mesoloph) on upper molars, are dental features characterising Chinchilloidea. The specimens described here do not show any heterogeneous thickness of the enamel layer, usually found in most advanced chinchilloid forms (including *Incamys*) [114, 127, 128]. There is also no heterogeneous thickness of the enamel layer in the alleged chinchilloid *Branisamys* (Salla, Bolivia, late early Oligocene–late Oligocene; [129–131], but see [4]), *Saremmys* [132], and in taxa recognised as stem-Chinchilloidea (*Eoincamys* and *Chambiramys*) [4]. Furthermore, this character is only insinuated in *Scleromys*/"*Scleromys*" [127]. By having high-crowned teeth with a tendency to show oblique loph(-id)s, not forming lamina, and with flexi (-ds) not filled with cement; p4s with a complete tetralophodont pattern or with a more trilophodont pattern; lower molars with a groove on the metalophulid I and without second

transverse cristid (or the latter if present, being strongly reduced); upper teeth with a metaloph reduced or absent, a third transverse crest absent or limited usually to a short mesoloph lingually free or backwardly connected to the posteroloph, the concerned dental specimens from TAR-31 have a morphology matching that of *Microscleromys nom. nud.* from La Venta (Colombia; late middle Miocene; Fig 24.2G-24.2O, p. 395 [18]; Figs 11–13 [37]). In particular, some specimens from TAR-31 are larger and attribuable to *M. paradoxalis*, whereas other, smaller, are determinable as *M. cribriphilus* (S1 File) [18, 37]. In addition, *M. cribriphilus* differs from *M. paradoxalis* in having some dP4s showing a non-taeniodont pattern characterised by a complete and high lingual protoloph in continuity to the labial protoloph (this study). Several other chinchilloid genera (*Eoincamys*, *Chambiramys*, *Incamys*, *Saremmys*, and *Scleromys*/"*Scleromys*") [28, 70, 71, 73, 81, 114, 118, 130–134] have a similar pattern (i.e., high-crowned teeth, homogeneous thickness of the enamel layer; taeniodonty, obliquity of loph(-id)s, no lamina and no cement). Like *Microscleromys*, *Eoincamys*, *Chambiramys*, *Incamys* and *Scleromys*/"*Scleromys*" also show a great variation of the morphology of the second transverse cristid on lower molars, and third transverse crest and metaloph on upper molars in terms of presence/absence, length, orientation and connections. No lower molar from TAR-31 and La Venta has a neomesolophid as long as or longer than on the MUSM 2875 m2 of *Chambiramys sylvaticus* (Fig 5G, p. 80 [71]). *Microscleromys* differs from *Chambiramys* in being slightly higher-crowned, and with more quadrangular upper molars; from *Eoincamys*, *Incamys* and *Scleromys*/"*Scleromys*" in having upper molars more square-shaped and a reduction more important of the third transverse crest on upper molars (i.e., more frequent trilophodont pattern than in these genera); from *Eoincamys ameghinoi*, *Eoincamys pascuali* and *Incamys* in having a second transverse cristid usually more reduced on lower molars; from *Eoincamys pascuali* in having a thinner furrow (when it is present) on the metalophulid I, which can be situated in the middle of this lophid; from *Saremmys* in having lower-crowned teeth, thinner loph(-id)s, a less oblique hypolophid on p4s and lower molars, and some upper molars with a distinct metacone, and long metaloph and mesoloph; from *Saremmys* and *Scleromys*/"*Scleromys*" in having an anterior arm of the hypoconid, which can be present on p4s and lower molars, and with a more curved posterolophid on lower molars; and from *Incamys*, *Saremmys*, and *Scleromys*/"*Scleromys*" in having a smaller size. *Saremmys* is documented by two dp4s at early stages of wear (MPEF-PV 11348b and 11349a in Fig 6A and 6B [132]). However, in our opinion, MPEF-PV 11348b would be a dP4 by the proportions between its mesial and distal parts, shape of its outline margin and a pentalophodont pattern closer to an occlusal morphology of a dP4 than dp4. Comparatively to MPEF-PV 11349a, dp4s from La Venta and TAR-31 show a metaconid postionned more distally and strongly connected to the protoconid by a thick second cristid, which corresponds to a metaconid cristid and/or a posterior arm of the protoconid. Unlike dP4s of *Microscleromys*, MPEF-PV 11348b has a hypocone compressed distolabially-mesiolingually. Moreover, MPEF-PV 11348b displays a more crestiform protocone. *Microscleromys* differs from *Garridomys* and *Drytomomys* in having a smaller size, an anterior arm of the hypoconid that can be present on p4s and lower molars, and a second transverse cristid usually more reduced, and a more curved posterolophid on lower molars; from *Garridomys* in having a homogeneous thickness of the enamel layer, thinner loph(-id)s, and strong and high protoconid and ectolophid on p4s; from "*Scleromys*" *colombianus* and *Drytomomys aequatorialis* in having a separation between the hypocone and the posteroloph, which is narrower and less persistent at early wear stages on upper molars [28, 29, 127, 135]; and from *Drytomomys* in having a less oblique hypolophid on lower molars. To sum up, the material from TAR-31 is closer to *Microscleromys*. After revison, we validate *Microscleromys paradoxalis* and *M. cribriphilus* described by Walton [37] from La Venta, and identify both species at TAR-31.

Microscleromys sp. 1

Fig 8D and 8E and S1 Table

*Nom. nud. Microscleromys* ?*paradoxalis* Walton, 1990, Fig 13E, 13D, p. 49.

## Referred material

UNC unnumbered specimen, left P4 (Fig 8D); IGM-DU 85–411, left P4 (Fig 8E).

## Localities

Locality unknown (for UNC unnumbered specimen) and CVP locality of the Fish Bed (for IGM-DU 85–411), La Venta Badlands, upper Magdalena Valley, Huila Department, Colombia.

## Formation and age

Baraya Member of the Villavieja Fm.; late middle Miocene (i.e., Laventan SALMA).

## Description and comparison

UNC unnumbered specimen (Fig 8D) and IGM-DU 85–411 (Fig 8E) are two P4s at advanced states of wear. They have a morphology very similar to those of *Microscleromys paradoxalis* and *M. cribriphilus* characterised by a circular/oval outline, a taeniodont pattern, the reduction/absence of third crest and metaloph, and a hypoflexus lingually close due to a connection between the posterior outgrowth of the protocone and the hypocone. The latter structures, as well as the paracone, anteroloph, protocone, posteroloph and mesostyle (and metacone if present) are fused, and form a circular peripheral wall. The paracone is distally separate from this circular wall by a very thin furrow on UNC unnumbered specimen (Fig 8D), whereas both structures are slightly linked together on IGM-DU 85–411 (Fig 8E). The labial protoloph and the mure are mesiolabially oriented on IGM-DU 85–411 (Fig 8E). On UNC unnumbered specimen, the labial protoloph is transverse and the mure is longitudinal (i.e., parallel to the mesiodistal plan of the tooth; Fig 8D). On that tooth, the anterior arm of the hypocone is tiny, oblique, and links the mure to the hypocone. From the mesostyle area runs a short crest, corresponding to a mesoloph, which remains labially free on UNC unnumbered specimen (Fig 8D). On the two P4s, there is no apparent metaloph. In addition, IGM-DU 85–411 does not have any distinct mesoloph (Fig 8E). However, this tooth shows a more advanced stage of wear and a thick labial part of the posteroloph, which could encorporate a mesostyle, the metacone, and/or the rest of a mesoloph and/or metaloph. These two teeth differ from their counterparts in *M. paradoxalis* and *M. cribriphilus* in being clearly larger (S1 File). By their taeniodonty, complete and elevated mure, and the connection between the posterior outgrowth of the procone and hypocone, the pattern of these P4s recalls the morphology of P4s of certain stem-caviomorphs (*sensu* Boivin et al. [4]; such as *Cachiyacuy contamanensis*, *Cachiyacuy kummeli*, *Pozomys*, *Eoespina*, and *Tarapotomys*), the chinchilloid *Eoincamys parvus*, and some cavioids (*Australopracta*, *Luantus*, and *Neoreomys*) [6, 11, 16, 70, 81, 86, 101, 136, 137]. The material from La Venta differs from *Cachiyacuy*, *Pozomys*, *E. parvus* and *Tarapotomys* in having a larger size; from *C. contamanensis* and some P4s of *Tarapotomys subandinus* in having a more reduced third crest; from *C. kummeli* in having an anterior arm of the hypocone and third crest (when present) reduced to a mesoloph instead of a mesolophule; from *Pozomys* in having a complete anteroloph; from *Australoprocta*, *Luantus initialis* and *L. propheticus* in having a slightly larger size; from *Australoprocta* in having no (or more reduced) metaloph; from *L.*

*initialis* and *L. propheticus* in having a curved lingual edge of the hypocone; from *L. propheticus* in having a lower crown and a less quick closure of the lingual mesoflexus; from *Neoreomys* in having a lower crown, thinner crests, more expanded flexi, and a mesoloph not connected to the posteroloph. In summary, these two P4s from La Venta are clearly closer to those of *Microscleromys*. Their large size is not compatible with that of two species of the genus. Thus, we attribute them to *Microscleromys* sp. 1.

Microscleromys sp. 2

Fig 11 and Table 1

## Referred material

MUSM 4658, left astragalus.

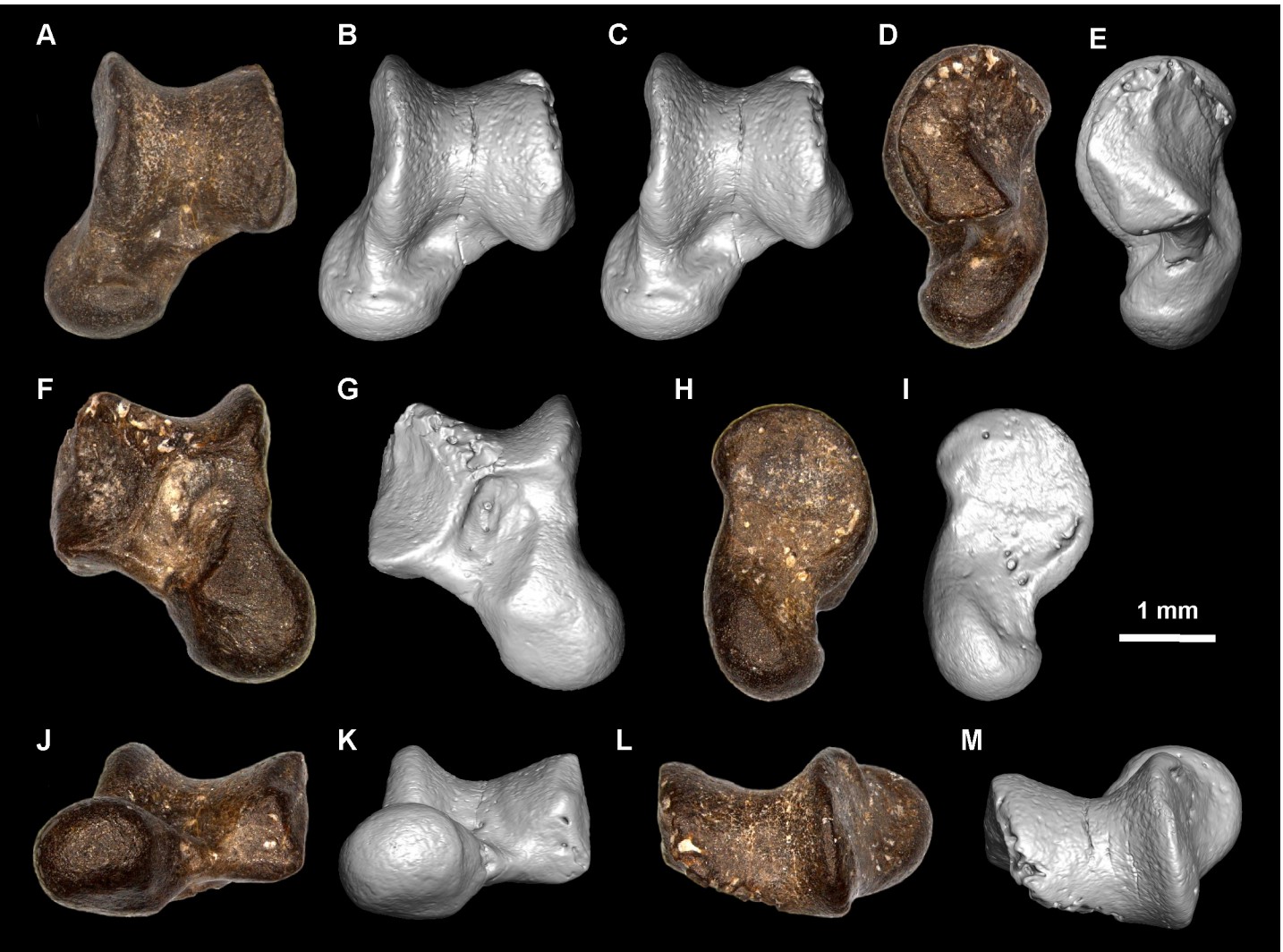

**Fig 11. *Microscleromys* sp. 2 from TAR-31, Peru, MUSM 4658, left astragalus.** (A-C) Dorsal, (D, E) lateral, (F, G) plantar, (H, I) medial, (J, K) distal, and (L, M) proximal views. (B, C) Stereomicroscopic views. The illustrations of the fossil specimen are the result of the fusion of numerical multi-focus images obtained with an optical stereomicroscope (A, D, F, H, J, and L) and 3D digital models obtained by X-ray μCT surface reconstruction (B-C, E, G, I, K, and M).

## Locality

TAR-31, Tarapoto/Juan Guerra, Mayo River, Shapaja road, San Martín Department, Western Amazonia, Peru.

## Formation and age

Lower member, Ipururo Fm., late middle Miocene (i.e., Laventan SALMA).

## Description

MUSM 4658 from TAR-31 is a well preserved left astragalus (Fig 11). It is slightly damaged at the level of the lateral ridge of the trochlea (= lateral tibial and lateral fibular facets) and the ectal facet. In dorsal view, the astragalar body is slightly longer (proximodistally) than wide (lateromedially; Fig 11A–11C). The combined length of the neck and head is shorter than that of the trochlea. The latter is characterised by steep-sided lateral and medial margins (fibular facet and medial tibial facet, respectively). It is moderately grooved with slightly asymmetric internal facets. The internal lateral facet of the trochlea (= lateral tibial facet) is extensive and slopes gently, whereas the medial one is more inclined, taller, narrower, and barely longer (Fig 11A–11C and 11J–11M). No astragalar foramen (= superior astragalar foramen *sensu* Cifelli [138]) is evident. The neck is thick, short and very deflected medially (36.2˚; Fig 11A–11C). The head is rounded and bears a navicular facet wider than long. In medial view (Fig 11H and 11I), the navicular and medial tarsal facets are slightly proximally distinct, but distally continuous. The medial tarsal facet is particularly extensive proximally on the medial aspect of the neck (Fig 11H and 11I). In lateral and medial views, the rims of the trochlea are circular, with a moderate radius of curvature (Fig 11D, 11E, 11H and 11I). In medial view, seven large foramina are recognisable (Fig 11H and 11I). Three foramina are present distally to the area along which the tendons of *m. peroneus longus* and *brevis* slide and ventroproximally situated to the most distalmost rim of the medial ridge of the trochlea. They are ventroproximally aligned, the smallest foramen being the most proximal, whereas the largest is in the middle. There are three other foramina on the dorsodistal part of the area along which the tendons of *m. peroneus longus* and *brevis* slide, which are dorsoproximally aligned. The last foramen is located proximally to the area along which the tendons of *m. peroneus longus* and *brevis* slide. In plantar view (Fig 11F and 11G), the ectal (= calcaneo-astragalar) facet is oval and faintly curved (very slightly concave; i.e., high radius of curvature). Its distal part is not markedly projected laterally (i.e., absence of a protruding lateral process). It is clearly separated from the sustentacular facet by a deep and wide sulcus calcaneus, which is oblique (proximomedially-distolaterally) and divergent distally. As the proximolateral side of the sustentacular facet is gently sloping, the sulcus appears particularly wide. Three main foramina are situated on the sulcus, the largest being the most proximal, close to the medial plantar tuberosity. It might correspond to the plantar astragalar foramen (= inferior astragalar foramen *sensu* Cifelli [138]). Another foramen is present at the base of the proximolateral slope of the sustentacular facet. This tuberosity is well marked. The sustentacular facet is triangular-shaped and with a major axis proximodistally oriented. It is well extended distally toward the head, and reaches the navicular facet. The distal extremity of the sustentacular facet is wide and concave, whereas its proximal one is narrower and convex. The sustentacular and ectal facets have similar dimensions (i.e., length and width). The navicular and medial tarsal facets are poorly distinct over their entire contact. In distal view (Fig 11J and 11K), two large foramina are present on the distal extremity of the lateral ridge of the trochlea.

## Comparisons

The trochlea of MUSM 4658 is shorter than in *Euryzygomatomys*, narrower than in *Steiromys*, *Neoepiblema* and *Coendou*, and less grooved than in *Cavia*, *Dolichotis* and *Proechimys*. It has a lateral ridge narrower and less distally positioned than *Chinchilla* and *Dactylomys*, and wider (but somewhat sloping) than the medial one, a feature observed on astragali of several other taxa (*Platypittamys*, *Drytomomys*, *Phoberomys*, *Abalosia*, *Actenomys*, *Dolicavia*, *Eucelophorus*, *Eumysops*, *Pithanotomys*, *Praectenomys*, *Dactylomys*, *Euryzygomatomys*, *Lagostomus*, *Proechimys*, and *Octodon*) and particularly accentuated on astragali of *Steiromys*, *Neoepiblema* and *Coendou*. The neck of MUSM 4658 is well differentiated from the body, and is strongly deflected medially as on astragali of *Steiromys*, *Drytomomys*, *Neoepiblema*, *Phoberomys*, *Abalosia*, *Eumysops*, *Coendou*, *Dactylomys*, *Euryzygomatomys*, *Myocastor* and *Proechimys*. Particularly, the deflection angle of MUSM 4658 is similar to that of astragali of *Abalosia*, *Eumysops*, *Myocastor* and *Proechimys* (35–37˚). Conversely, astragali of *Platypittamys*, *Neoreomys*, "*Scleromys*", *Actenomys*, *Dolicavia*, *Eucelophorus*, *Pithanotomys*, *Praectenomys*, *Cavia*, *Chinchilla*, *Clidomys*, *Ctenomys*, *Dinomys*, *Dolichotis*, *Galea*, *Lagostomus* and *Octodon* have their neck being clearly less or not deflected medially. On MUSM 4658, the head is rounded with the medial tarsal facet strongly developed on the medial aspect of the neck in dorsal view as on astragali of *Steiromys*, *Drytomomys*, *Actenomys*, *Eucelophorus*, *Eumysops*, *Pithanotomys*, *Coendou*, *Dactylomys*, *Octodon*, and *Proechimys*. MUSM 4658 displays an ectal facet narrower than that of astragali of *Neoepiblema*, *Actenomys* and *Eumysops*; shorter than in *Pithanotomys*; narrower and longer than in *Coendou*. Contrary to the astragali of *Neoreomys*, "*Scleromys*" colombianus, *Drytomomys*, *Neoepiblema*, *Eumysops*, *Pithanotomys*, *Cavia*, *Chinchilla*, *Dinomys*, *Dolichotis*, *Galea*, *Myocastor*, *Octodon* and *Proechimys cuvieri*, there is no well marked lateral process of the ectal facet on MUSM 4658. The ectal facet is less concave than in *Steiromys*, *Neoepiblema*, *Pithanotomys*, *Cavia porcellus*, *Dolichotis* and *Galea*. Several taxa (*Steiromys*, "*Scleromys*", *Drytomomys*, *Neoepiblema*, *Phoberomys*, *Abalosia*, *Actenomys*, *Dolicavia*, *Eucelophorus*, *Eumysops*, *Pithanotomys*, *Cavia*, *Chinchilla*, *Dactylomys*, *Dinomys*, *Dolichotis*, *Euryzygomatomys*, *Galea*, *Lagostomus*, *Myocastor*, *Octodon*, and *Proechimys*) have a wide sulcus calcaneus distally divergent like on MUSM 4658, but none seem to have a similar configuration of this sulcus and the proximolateral side of the sustentacular facet, which is gently sloping. In addition, MUSM 4658 differs from astragali of all aforementioned taxa in having a particular triangular-shaped sustentacular facet; from *Steiromys*, *Neoepiblema*, *Coendou*, *Lagostomus*, and *Myocastor* in having a sustentacular facet oriented proximodistally; from *Dolicavia*, *Cavia*, *Chinchilla*, *Dolichotis*, *Euryzygomatomys*, *Galea* and *Myocastor* in having a sustentacular facet not strongly fused with the navicular facet and/or the medial tarsal facet of the neck; and from *Steiromys*, *Abalosia*, *Eumysops*, *Pithanotomys* and *Dactylomys* in having no clear separation between the sustentacular facet and the navicular and medial tarsal facets. At first order, the MUSM 4658 astragalus matches very small body-sized extant and fossil rodents. Regression of the body mass in function of trochlear width in living mammals [139] shows that MUSM 4658 belonged to a small rodent having a body mass of about 43 g (Table 2). The smallest body-sized species from TAR-31, *Ricardomys longidens*, *Microscleromys paradoxalis* and *M. cribriphilus*, could be the best candidates. Considering the least-squares regression equation of [140] based on the m1 area in mammals, the body mass of *R. longidens* is estimated to be ~40 g, that of *M. paradoxalis* to be ~51 g and that of *M. cribriphilus* to be ~21 g (Table 2). The values estimated with equation proposed by the same author for rodents are clearly higher (~132, ~161, and ~76 g; Table 2), which makes it difficult to conclude. Given that the dental material from TAR-31 referred to *M. cribriphilus* is more abundant than that of *R. longidens* and *M. paradoxalis*, referral of MUSM 4658 to *M. cribriphilus* could be a plausible taxonomic

**Table 2. Metric feature of the MUSM 4658 astragalus (mm) and m1s (mm²) from TAR-31 and La Venta and body mass estimates (g).**

| Material | Trochlear Width (mm) | Mean area m1 (mm²) | Dental locus used | N | Body mass estimates (g) | Regression equation for: | Reference |
|---|---|---|---|---|---|---|---|
| MUSM 4658 | 1.832 | | | 1 | 43.00 | mammals | Tsubamoto, 2014 |
| *Microsteiromys jacobsi* | | 7.51 | m1 | 1 | 243.26 | mammals | Legendre, 1986 |
| | | | | | 614.50 | rodents | |
| *Nuyuyomys chinqaska* | | 4.36 | m1 | 6 | 89.98 | mammals | |
| | | | | | 262.28 | rodents | |
| *Ricardomys longidens* | | 2.82 | m1? | 2 | 40.49 | mammals | |
| | | | | | 132.40 | rodents | |
| *Microscleromys cribriphilus* | | 1.97 | m1? | 34 | 21.15 | mammals | |
| | | | | | 75.94 | rodents | |
| *Microscleromys paradoxalis* | | 3.19 | m1* | 2 | 50.90 | mammals | |
| | | | | | 161.06 | rodents | |
| "*Scleromys*" sp. | | 14.67 | m | 2 | 826.65 | mammals | |
| | | | | | 1751.02 | rodents | |

* based on MUSM 4643 from TAR-31 and IGM unnumbered specimen from La Venta.

option. Nevertheless, as the size ranges of two co-occuring species of *Microscleromys* are overlapping, we prefer to keep it in open nomenclature as *Microscleromys* sp. 2.

"*Scleromys*" Ameghino, 1887

"*Scleromys*" sp.

Fig 12 and S1 Table

### Referred material

MUSM 4277, left dp4; MUSM 4278, 4279, right fragmentary dp4s; MUSM 4280, left p4; MUSM 4281, 4282, right lower molars; MUSM 4270–4272, left fragmentary upper molars; MUSM 4273, 4274, left upper molars; MUSM 4275, 4276 right upper molars.

### Locality

TAR-31, Tarapoto/Juan Guerra, Mayo River, Shapaja road, San Martín Department, Western Amazonia, Peru.

### Formation and age

Lower member, Ipururo Fm., late middle Miocene (i.e., Laventan SALMA).

### Description

MUSM 4278 is a distal fragment of dp4 preserving only the posterolophid, the hypoconid, its anterior outgrowth, and the distal margin of the ectolophid, hypolophid, and of the entoconid. MUSM 4279 is a mesial fragment of dp4, broken at the level of the ectolophids, the hypolophid and the entoconid (Fig 12C). This tooth also shows several damaged areas, especially at the level of the protoconid and the distal part of the metalophulid I. MUSM 4277 (Fig 12A) is a

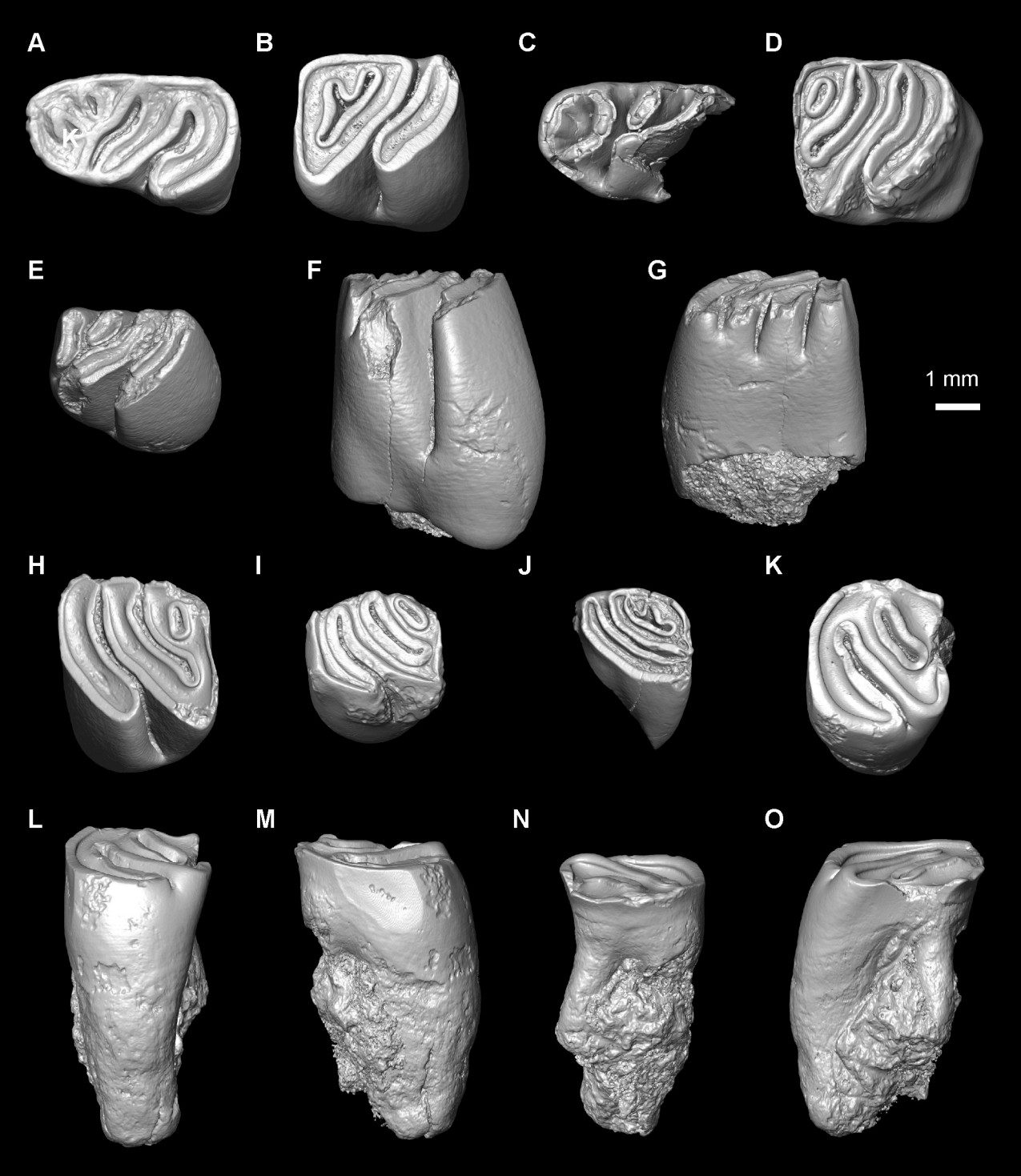

**Fig 12. "*Scleromys*" sp. indet. from TAR-31, Peru.** (A) MUSM 4277, left dp4. (B) MUSM 4281, right lower molar (reversed). (C) MUSM 4279, fragment of right dp4 (reversed). (D) MUSM 4282, right lower molar (reversed). (E–G) MUSM 4280, left p4. (H) MUSM 4273, left upper molar. (I) MUSM 4275, right upper molar (reversed). (J) MUSM 4272, left upper molar. (K–O) MUSM 4276, right upper molar (reversed). (A-E, H-K) Occlusal, (F, N) labial, (G, L) lingual, (M) anterior, and (O) posterior views. The illustrations of the fossil specimens are 3D digital models obtained by X-ray μCT surface reconstruction.

complete dp4 at a more advanced wear stage than MUSM 4278 and 4279. These three brachy-dont dp4s are characterised by thick lophids. MUSM 4277 is pentalophodont, probably like MUSM 4279. Despite its degree of wear, MUSM 4279 seemingly displays bulbous lingual cuspids. On this tooth, the metalophulid I is mesially convex and shows a mesiodistal thickening at its centre. The metaconid and protoconid are distally linked by a second cristid, mesially concave, corresponding to a posterior arm of the protoconid and/or a metaconid cristid. The metalophulid I, the second cristid, the metaconid and protoconid delimit a C-shaped antero-fossettid. The mesostylid is large and displays a short and strong neomesolophid mesiolabially directed and lingually free. The mesostylid is well separated from the metaconid and entoconid. This cuspid is higher than the mesostylid and metaconid. The hypolophid, distal ectolophid and mesial ectolophid are fused, and form a long cristid stemming from the entoconid and directed toward the protoconid, but without connection to it. This cristid is mesiolabially oriented, but its labial part is more longitudinal than its lingual part. The labial extremity of the neomesolophid is close to the mesial margin of the cristid combining the hypolophid and ectolophids. MUSM 4277 (Fig 12A) shows a similar pattern to that of MUSM 4279 (Fig 12C), but with features deriving from a more advanced stage of wear: loss of relief, lingual closure of the mesial mesoflexid and metaflexid, lingual and labial closure of the distal mesoflexid, and reduction of the anterofossettid and of the mesial mesofossettid to two small and circular fossettids. MUSM 4279 displays a heterogeneous thickness of the enamel layer, which is thicker on the trailing edges and thinner on the leading edges. On MUSM 4277 and 4278, the distal-most cristid (posterolophid + hypoconid + its anterior outgrowth) is curved and there is no anterior arm of the hypoconid, thereby involving the confluence of the metaflexid with the hypoflexid (i.e., taeniodont pattern).

The MUSM 4280 p4 (Fig 12E) is slightly damaged at the level of the protoconid, the hypoconid and its anterior outgrowth. MUSM 4280 is hypsodont (Hling = 1.13; Hlab = 1.52; Fig 12F and 12G). Compared to dp4s, MUSM 4280 is tetralophodont, more square-shaped, and with a trigonid clearly narrower than the talonid. The cuspids are crestiform and faintly differentiated from the lophids. On the lingual margin, the metaconid and mesostylid are equally sized and separated by the anterostriid. The metaconid is connected to a metalophulid I, which is very slightly concave and labially free. This crest corresponds to an anterior arm of the metaconid, the anterior arm of the protoconid being absent. Its labial extremity is thin and separated from the protoconid by a narrow and deep furrow, which forms a striid. The second transverse cristid is reduced to a short neomesolophid stemming from the mesostylid. The neomesolophid is mesiolabially directed and labially free. The protoconid is fused with the ectolophid, forming a cristid distolingually oriented. The hypolophid is parallel to the neomesolophid and in continuity of the cristid composed of the protoconid + ectolophid (but the latter being more longitudinal). The hypoconid has no anterior arm (i.e., taeniodont pattern) and it is not distinct from its anterior outgrowth and the posterolophid. The main direction of the posterolophid is the same that those of the neomesolophid and hypolophid. The flexids are furrow-shaped and form striids. On the labial margin, the hypostriid is deep, but ends far from the neck of the crow. On the lingual margin, the anterostriid and mesostriid have a similar depth, which represents more than the double that of the short metastriid.

As for p4, the lower molars, MUSM 4281 and 4282, are high crowned (but retaining roots), tetralophodont and taeniodont, with deep furrow-shaped flexids, forming striids, dental structures that fused with wear, forming thick and oblique cristids, and a second transverse cristid reduced to a neomesolophid (Fig 12B and 12D). Contrary to the MUSM 4280 p4, their trigonid and talonid are equally wide. They present a more advanced stage of wear with respect to MUSM 4280: their crown is lower (Hling = 0.82–0.83; Hlab = 0.82–1.19) and their lingual flexids are closed except the metaflexid on MUSM 4281. The metalophulid I is complete on both

molars. On MUSM 4281 (Fig 12B), the neomesolophid is very short and labially free. Consequently, the Y-shaped anterior fossettid corresponds to the confluence between the antero-flexid with the mesoflexid. Conversely, on MUSM 4282 (Fig 12D), the neomesolophid is longer and labially connected to the metalophulid I, thereby delimiting a small and circular anterofossettid. Like on MUSM 4280, the hypolophid, ectolophid and protoconid are in continuity, forming a central cristid. The latter is straight and oblique (i.e., mesiolabially directed) on MUSM 4281. On MUSM 4282, its labial part is mesiolabially directed, while the hypolophid is more transverse. Due to the closure of the anteroflexid and mesoflexus on MUSM 4281 and 4282, the trigonid is delimited by a triangular-shaped structure formed by the metalophulid I, metaconid, mesostylid, entoconid, hypolophid, ectolophid and protoconid. This triangular-shaped structure is separate from the distalmost cristid by a very thin furrow-shaped flexid (i.e., confluence of the metaflexid with the hypoflexid). The distalmost cristid, formed by the fusion of the posterolophid, hypoconid and its anterior outgrouwth, is mesially concave. The thickness of the enamel layer is homogenous on MUSM 4282 (Fig 12D). MUSM 4281 displays a homogenous thickness of the enamel layer at the level of the metaflexid but a heterogeneous one (thicker on the trailing edges and thinner on the leading edges) at the level of its mesial and distal margins and those of the hypoflexid (Fig 12B). The hypostriid has a height similar to that of the MUSM 4280 p4.

Upper molars have an occlusal pattern similar to that characterising p4 and lower molars (i.e., tetralophodonty, taeniodonty, deep furrow-shaped flexi, forming stria, and dental structures merging with wear, forming thick and oblique crests), which can contribute to confuse them. They show a unilateral hypsodonty characterised by a lingual crown higher that the labial one (Hling = 1.00–1.69; Hlab = 0.51–0.92), whereas the distolabial crown is the highest part of crown on p4 and lower molars. Upper molars are labiolingually wider than lower molars. Upper molars have three roots: a large lingual one and two small labial ones. The mesial margin of the crown is formed by a curved crest, which corresponds to the fusion of the anteroloph, protocone and its posterior outgrowth. On all complete upper molars, this crest is labially connected to the paracone. The labial protoloph is roughly transverse or slightly oblique and in continuity with the mure and anterior arm of the hypocone, which are more disto-lingually directed. The mesostyle is faintly isolated from the metacone on MUSM 4273 (Fig 12H). On the other upper molars, it is slightly (MUSM 4270 and 4272; Fig 12J) or strongly (MUSM 4274–4276; Fig 12I and 12K) connected to the this cusp and to the paracone. As for the second transverse cristid on p4 and lower molars, the third transverse crest is reduced to a mesoloph stemming from the mesostyle, either lingually free (MUSM 4270 and 4272) or connected to the posteroloph (MUSM 4273 and 4275; Fig 12H and 12I). On MUSM 4272 (Fig 12J), the posteroloph presents, on its mesial margin, a very short crestule (i.e., posteroloph spur), mesially free. On MUSM 4273 (Fig 12H) and 4275 (Fig 12I), the mesoloph and postero-loph (+ posteroloph spur?) delimit a small and circular fossette. On upper molars at a more advanced stage of wear (MUSM 4274 and 4276; Fig 12K), the small posterior fossette disappears due to the fusion between the mesoloph, mesostyle and posteroloph (+ posteroloph spur?). On MUSM 4270 and 4272 (Fig 12J), a tiny notch separtes the hypocone from the pos-teroloph. On the other upper molars, which show a more advanced stage of wear, the hypo-cone is strongly linked to the posteroloph. On the most worn upper molar, MUSM 4274, the two remaining flexi (the mesialmost) are fully closed and divided into two parts. Most upper molars (MUSM 4270, 4272 and 4274–4276; Fig 12J, 12I and 12K) display a homogenous thick-ness of the enamel layer, while MUSM 4273 (Fig 12H) has a slight heterogeneous thickness of the enamel layer (thicker on the trailing edges and thinner on the leading edges) at the level of its mesial and distal margins, and the mesial and distal margins of the anterior flexus.

## Comparisons

These large dental specimens from TAR-31 described here show typical chinchilloid affinities, notably characters documenting *Scleromys*/"*Scleromys*": (i) medium-sized and protohypsodont teeth, (ii) taeniodonty, (iii) dental structures merging with wear, forming thick and oblique crests/cristids, (iv) deep furrow-shaped flexi(-ds), forming striia/striids, (v) upper molars tetra-lophodont at early stages of wear, becoming trilophodont at more advanced stages of wear, (vi) a metaflexid as being the most persistent lingual flexid on lower teeth, and (vii) an enamel layer with a thickness homogeneous or slightly heterogeneous (i.e., insinuated heterogeneous thickness *sensu* Kramarz, Vucetich & Arnal [127]; see also [128]). The dental remains from TAR-31 are similar in size to those from La Venta (Colombia; late middle Miocene) attributed to either "*Scleromys*" *schurmanni* Stehlin, 1940 [18, 27, 28] or "*Scleromys*" cf. *schurmanni* [18]. By its short and labially-free metalophulid I and neomesolophid, the pattern of the MUSM 4280 p4 is particularly close to that of the p4 borne on the IGM 184233 specimen from La Venta attributed to "*Scleromys*" cf. *schurmanni* (Fig 24.3E, p. 397 [18]). However, it differs from the latter in having no mesiolingual protuberance of crown, and in showing separated metaconid and mesostylid. As for the material from TAR-31, the lower molars of "*Scleromys*" cf. *schurmanni* from La Venta can have a neomesolophid labially free or connected to the metalophulid I. The figured p4 of "*Scleromys*" *schurmanni*, also from La Venta (UCMP 38967 in Fig 24.3D, p. 397 [18]), has a complete second transverse cristid composed of a neomesolo-phid and a shorter posterior arm of the protoconid, contrary to MUSM 4280. The p4s of "*Scleromys*" *schurmanni* described by Field [28] show a complete second transverse cristid as well (UCMP 38987 in Figs 4A and 5D, p. 285 [28]) or a second cristid reduced to a neomesolo-phid linked to the metalophulid I (UCMP 37931 in Figs 4D and 5C, p. 285 [28]). The dp4s of "*S*". *schurmanni* (UCMP 39902 and 38974 in Fig 5A and 5B, p. 285 [28]) seem to be hexalo-phodont, having more anterior flexids than on dp4s from TAR-31. The TAR-31 teeth are smaller than those of *S. quadrangulatus* (Pinturas Fm., Argentina; late early Miocene) [133], *S. angustus* (Santa Cruz Fm., Argentina; late early Miocene) [73] and *S. osbornianus* (Pinturas and Santa Cruz Fm.; late early Miocene) [118, 133]. Comparatively to *S. quadrangulatus*, teeth from TAR-31 have a slender closure of the flexi(-ds), a longer metalophulid I on p4, and a paracone area smaller on upper molars at early stages of wear. The MUSM 4280 p4 has a lin-gual margin more parallel to the mesiodistal axis of the tooth than p4s of *S. quadrangulatus* and *S. angustus*. In addition, MUSM 4280 has a neomesolophid, contrary to those species of *Scleromys*. At early stages of wear, molars of *S. angustus* and *S. osbornianus* display a tilopho-dont pattern [73, 133], contrary to molars from TAR-31, which remain tetralophodont. The dp4s from TAR-31 share a similar pattern in terms of number, types and morphology of the main cristids with the two dp4s from Madre de Dios, Peru: dp4 referred to *S. quadrangulatus* (MD-61, early Miocene) [141] and dp4 referred to "*Scleromys*" sp., gr. *quadrangulatus-schur-manni-colombianus* (MD-67, early middle Miocene) [31]. Nevertheless, they have a metalo-phulid I connected to the metaconid and protoconid, contrary to dp4s from MD-61 and -67. Unlike dp4s from TAR-31, dp4s from MD-61 and -67 have supernumerary structures on the anteroflexid: a cuspid-shaped structure on the dp4 from MD-67, and a low cristulid connected to the lingual and labial extremity of the metalophulid I on the dp4 from MD-61. The dp4s from TAR-31 display more opened lingual flexids than the dp4 from MD-61. They show other distinctive characters with respect to the dp4 from MD-67: a metaconid isolated from the mesostylid, and a second cristid and a hypolophid-ectolophid cristid, which are more curved. "*Scleromys*" *colombianus* and *Drytomomys aequatorialis*, both from La Venta (and from Nabón region, Ecuador, for *D. aequatorialis*; late middle Miocene) [28, 29, 135] are recognisa-ble in having larger-sized teeth and a separation between the hypocone and the posteroloph,

wider and more persistent at early stages of wear on upper molars, than on the TAR-31 material. The material from TAR-31 differs from "*S.*" *colombianus* [18, 28] and "*Scleromys*" cf. *colombianus* [18] in having p4s with a transverse mesial margin and a shorter neomesolophid labially free, and upper molars with a more persistent posterior fossette. It differs from *Drytomomys* in having a quicker reduction of the anterofossettid on lower molars; from *D. aequatorialis* in having shorter neomesolophid on p4s; from *D. typicus* (Ituzaingo Fm., Argentina; late Miocene) [29, 106] in having different morphology of dp4s with a third cristid reduced to a short neomesolophid and without neolophid on the anterofossettid. Comparatively to MUSM 4280, "*Drytomomys*" sp. (small) and "*Drytomomys*" sp. (large) from La Venta ("*Olenopsis*" sp. *sensu* Walton [18], but see [29]) show a longer neomesolophid labially connected to the metalophulid I on p4s. Moreover, p4s of "*Drytomomys*" sp. (large) are pentalophodont with a short metaconid cristid, contrary to MUSM 4280. The moderate size, marked hypsodonty and thickness of the crests of these teeth from TAR-31 exclude affinities with *Eoincamys* (Brazil and Peru; ?late Eocene–early Oligocene) [8, 19, 70, 81], *Microscleromys* (La Venta, TAR-31; this study) [18] and with the octodontoid *Protadelphomys* (Sarmiento Fm., Argentina; early Miocene) [142–144]. They are also too high-crowned and slightly larger for *Maquiamys* (CTA-61, Peru; late Oligocene) [4, 71]. To sum up, the material from TAR-31 has the closest affinities with the specimens of "*Scleromys*" found in northern South America (La Venta, Colombia and MD-61 and MD-67, Peru). As it would be necessary to deeply revise *Scleromys* and notably the northern forms (recognized as "*Scleromys*" [18, 71]), we prefer keeping the material from TAR-31 in open nomenclature as "*Scleromys*" sp.

## Discussion

### The caviomorph fauna from La Venta: Revision and validation of Walton's *nomina nuda*

*Microsteiromys jacobsi*, *Ricardomys longidens*, *Microscleromys paradoxalis* and *Microscleromys cribriphilus*, originally described in La Venta in the framework of the Anne H. Walton's Ph.D [37], have remained *nomina nuda* given that they were not formally described so far. According to the International Code of Zoological Nomenclature (p. 246 [38]), a *nomen nudum* is not available but the same name can be made available subsequently for the same concept or for a different concept, in a way meeting the criteria of availability (chapter 4 [38]). Our revision of the original material of these species, combined with the study of the new specimens from TAR-31, allowed us to validate all the concerned taxa. It is worth noting that we have usually not followed the unpublished diagnoses proposed by Walton [37]. In the case of *Microscleromys*, a classical descriptive and comparative approach combined with statistical analyses on dental dimensions (length and width; S1 File) allowed us to test the taxonomic groups proposed by Walton [18, 37]. Our results (i.e., statistical differences) suggest an inappropriate taxonomic delimitation of these potential groups (S1 File). Thus, we propose a new delimitation of the two *Microscleromys* species implying changes in the assignation of some specimens (i.e., from *M. cribriphilus* to *M. paradoxalis*, and vice versa). In addition, we revised the suprageneric status of this taxon, which was previously assigned to the cavioid family Dasyproctidae [18, 37]. Like *Microscleromys*, other fossil taxa such as *Eoincamys*, *Incamys* and *Scleromys*, were previously attributed to the dasyproctids [81, 130, 131]. However, *Eoincamys*, *Incamys* and *Scleromys* have then been related to the superfamily Chinchilloidea [70, 114, 133]. As noted by Walton [18], *Microscleromys* shows an occlusal pattern similar to *Scleromys*, as well as to chinchilloids in general (i.e., high-crowned teeth with a taeniodont pattern and oblique loph [-id]s, the absence of the second transverse cristid or its reduction on lower molars, and the absence of the metaloph and third transverse crest, or the reduction of the latter to notably a

mesoloph on upper molars). According to the recent phylogenetic analyses of Boivin et al. [4], the phylogenetic position of *Microscleromys*, but also those of *Eoincamys*, *Incamys* and *Scleromys* were retrieved as stem representatives of Chinchilloidea (but see [127, 145, 146]). Finally, as noted by Walton [18, 37], *Ricardomys* shows an occlusal morphology similar to *Adelphomys* and other adelphomyine octodontoids. Here, we assign it to the family Adelphomyidae, following the results of Boivin et al. [4].

## The caviomorph assemblage from TAR-31

In being restricted to a single 5–20 cm-thick channelised lens in a very small area (~5 square metres in one piece), TAR-31 records a single attritional accumulation [36], which strongly contrasts with La Venta and Fitzcarrald local faunas (several levels from distinct outcrops spanning a long or short interval, respectively) [26, 33]. Nevertheless, TAR-31 yields a total of nine rodent taxa, with the co-occurrence of the four extant superfamilies: Erethizontoidea (three taxa), Cavioidea (one taxon), Octodontoidea (two taxa) and Chinchilloidea (three taxa). Compared with other Laventan caviomorph assemblages from low latitudes of South America, TAR-31 ranks second just in front to the fauna from Fitzcarrald Arch (eight taxa) [32, 33], but far from La Venta, which includes 28 taxa (21 excluding the indeterminate genera and specie; La Victoria and Villavieja formations, Colombia; see [18, 28] and this work). Like TAR-31, the faunas from La Venta and Fitzcarrald Arch, as well as that from MD-67 (Madre de Dios; similar outcrop conditions as in TAR-31) [31], show representatives of the four caviomorph superfamilies. Without considering the validation of the *nomina nuda* from La Venta, TAR-31 records a new erethizontoid (*Nuyuyomys chinqaska* gen. et sp. nov.) and perhaps a new stem-caviid, here described as Caviidae gen. et sp. indet. (see Systematic Palaeontology section). The single specimen assigned to this taxon does not allow for its formal description as a new species. TAR-31 shares two taxa or closely related taxa with MD-67 ("*Scleromys*" sp. and *Nuyuyomys chinqaska*) and one with the assemblages from Fitzcarrald local fauna and La Victoria Fm. of La Venta ("*Scleromys*"). TAR-31 shares with the Villavieja Fm. of La Venta a representative of "*Scleromys*", as well as three species only recorded in La Venta so far: *Microscleromys paradoxalis*, *M. cribriphilus* and *Ricardomys longidens*. In striking contrast, there is no taxon in common between TAR-31 and most middle Miocene assemblages at mid- and high latitudes in South America (Argentina, Bolivia; Chile) [10, 30, 51, 87, 105, 123, 126, 147–152]. The only assemblage showing similarities with TAR-31 is that from the Marino Fm. (Divisadero Largo area, Mendoza, Argentina) [153], in yielding *Scleromys* sp. Nevertheless, *Scleromys* (and "*Scleromys*") had a wide geographic (Argentina, Brazil, Chile, Colombia, and Peru) and a long stratigraphical range (late Oligocene–late middle Miocene) [15–19, 27, 28, 31, 33, 34, 73, 133, 141, 154]. Moreover, some doubts were expressed about the cogeneric afinities between *Scleromys* spp. from northern *versus* southern South America [18, 71].

## Astragalus of *Microscleromys* from TAR-31: Morpho-functional implications

The trochlea of MUSM 4658 does not have a mediolateral widening nor a high degree of asymmetry between its medial and lateral rims, thereby suggesting that parasagittal movements would be more favoured during plantar- or dorsi- flexion of the foot [57, 61, 155]. The trochlea is however moderately grooved with shallower and less sharp crests than those of astragali of cursorial-jumping taxa in which lateral movements are strongly restricted (cavioids: *Cavia*, *Dasyprocta*, *Dolichotis*, *Galea*) [61]. The ectal facet is long with a high radius of curvature and not projected laterally, which would give proximodistal and mediolateral mobility with the calcaneus (astragalo-calcaneal joint = sub-astragalar joint) [57]. The narrowing and the

proximodistal orientation of the sustentacular facet (here long reaching the navicular facet) would indicate parasagittal movements at the astragalo-calcaneal joint, limiting the inversion of the foot [57, 61]. In addition, the sulcus calcaneus is particularly wide, thereby suggesting the presence of a strong ligament linked to the dorsal side of the calcaneus [57]. This character is found in numerous caviomorphs, but associated with a proximodistal sustentacular facet, it would confer more resistance of the ankle by limiting the general mobility of the foot with respect to the leg [156, 157]. The neck is long and medially deflected with a medial tarsal facet on its dorsal and medial aspects, and a round head, characters typically found in arboreal taxa such as *Coendou* or *Sciurus*, but also in some terrestrial echimyids (*Proechimys*) [57, 61]. In addition to the increase of the articular surface of the sustentacular facet with the calcaneus [57], these features would therefore be facilitated lateral and rotator movements of the foot at the astragalo-navicular joint and at the contact between the astragalus and the medial tarsal bone (= first sesamoid) [61, 155]. Moreover, the extension of the medial tarsal facet on the neck would permit to stabilize the astragalus during inversion of the foot [57, 155, 158]. In summary, MUSM 4658 shows an intermediary morphology between morphotypes of the cursorial-jumping and climbing locomotory groups recognised by Ginot et al. [57]. Some of its characters (i.e., ectal facet slightly concave and not projected laterally, neck medially deflected, round head, medial tarsal facet extended on the dorsal and medial aspects of the neck) would enhance mediolateral movements particularly at the level of the distal joints with the navicular and medial tarsal bone, whereas others (i.e., moderately groeved and slightly asymmetric trochlea with no mediolateral widening, narrowing and the proximodistal orientation of the sustentacular facet, wide sulcus calcaneus) would tend to favor parasagittal movements especially of the astragalo-tibial joint (= cruro-astragalar joint). Such an astragalar morphology matches that of terrestrial generalists with a possible higher trend toward arboreal locomotion.

## Age of TAR-31 and biostratigraphical implications

The caviomorph fauna from TAR-31 shares four taxa with the assemblage characterising the lower part of the Villavieja Fm. at La Venta (see above). Similar affinities are observed on their non-rodent mammalian content (e.g., *Miocochilius anomopodus*, *Neosaimiri fieldsi* and *Megadolodus molariformis*; Fig 29.5, p. 508 [26]; see also [36] and the section Material and methods of this paper). An age ranging between 13.18–12.47 Ma can therefore be hypothesised for TAR-31, corresponding with the timespan of the magnetostratigraphically-constrained lower part of the Villavieja Fm. (C5AAn–C5AR.1R chrons) [26, 49].

The MUSM 4283 p4 of Caviidae gen. et sp. indet. from TAR-31 is particular in having more primitive characters (i.e., protohypsodonty, presence of flexids/fossettids, and a hypoflexid not reaching the lingual margin of the tooth) than the caviids found in other caviomorph faunas at low latitudes of South America from Laventan deposits such as *Prodolichotis pridiana* (La Venta and Fitzcarrald Arch), Dolichotinae gen. 2 large and Dolichotinae gen. 2 small (La Venta), and *Guiomys* sp. (MD-67) [18, 31, 33] or from sligthly younger deposits (Mayoan SALMA; CTA-44 Top, CTA-43) [34]. Similarly, it differs from middle Miocene caviids from southern South America (*Cardiomys*? *andinus*, *Cardiomys*? *buemulensis*, *Eocardia robusta*, *Eocardia robertoi*, *Guiomys unica*, *Microcardiodon huemulensis* and *Microcardiodon williensis*) [51, 87, 105]; in having of flexids/fossettids, and a hypoflexid, which does not reach the lingual margin of the tooth, characters otherwise found only in early Miocene stem-caviids such as *Chubutomys* or *Luantus* [52, 101, 103]. Thus, the discovery of a tooth with such characters suggests that this kind of morphotype may have subsisted until the late middle Miocene, at least in some representatives of the superfamily at low latitudes of the continent. Occurring in both MD-67 and TAR-31 localities, *Nuyuyomys chinqaska* likely spans the entire middle Miocene interval.

## Palaeobiogeographical implications

The Miocene epoch is marked by important environmental changes, particularly in low-latitude regions of South America [159]. During the latest Oligocene–early Miocene interval, Andean deformations are generalised, in particular by the Eastern Cordillera of the Northern and Central Andes (beginning of the "Quechua phase" as defined by Steinmann [160]; [161–165]). However, the Eastern Cordillera of Northern Andes had not reached its full elevation at that time, with a major uplift episode during late Miocene–Pliocene times [166–168]. In relation with the uplift of this region, the Solimões Basin in Brazil is occupied by swamps and shallow lakes [169–173]. These bodies of water belonged to a vast wet system (mega-wetland), which also covered the Andean foreland retroarc basins in Peru and Colombia, and had a north connection with the Caribbean Sea [174]. This system reaches its maximal extension from ~16.0 to 11.3 Ma and forms the Pebas System, which corresponds to a dynamic mosaic of swampy, lake, fluvial and fluvio-marine environments [174, 175].

With our results, the geographical ranges of *Ricardomys longidens*, *Microscleromys paradoxalis* and *M. cribriphilus* are now extended southerly to Peruvian Amazonia for the Laventan interval. More generally, strong taxonomic affinities between the vertebrate assemblages of La Venta (Colombia) and Fitzcarrald, Madre de Dios (MD-67), Iquitos, Contamana, and TAR-31 in Peruvian Amazonia support the existence of a single "Pebasian" biome throughout the middle–earliest late Miocene, and more specifically during the Laventan SALMA, as shown by their terrestrial and aquatic components (crocodylomorphs, chelonians, marsupials, rodents, primates, notoungulates, astrapotheres, litopterns, sirenians, and/or cetaceans; [21, 26, 31–34, 36, 176–178] and see this work). These moderate or strong faunal similarities suggest that the Eastern Cordillera of the North Andes and Pebas System would have not constituted barriers for mammalian dispersals during this interval at least for a region close to the Andes (Western Corridor) [36, 44].

## Conclusions

The TAR-31 locality in Tarapoto area, Peru, documents nine caviomorph taxa encompassing all four living superfamilies. This assemblage encompasses: *Microsteiromys jacobsi* gen. et sp. nov., *Nuyuyomys chinqaska* gen. et sp. nov., Erethizontoidea gen. et sp. indet. 1 and 2 (Erethizontoidea), Caviidae gen. et sp. indet. (Cavioidea), *Ricardomys longidens* gen. et sp. nov. and Octodontoidea gen. et sp. indet. (Octodontoidea), plus *Microscleromys paradoxalis* gen. et sp. nov., *M. cribriphilus* gen. et sp. nov., and "*Scleromys*" sp. (Chinchilloidea). The study of an astragalus suggests that at least a representative of *Microscleromys* was a terrestrial generalist rodent.

Although very restricted in terms of area (5 square meters) and volume investigated (~550 kg wet-screened) with respect to other assemblages, this locality yields the most diverse middle Miocene caviomorph fauna recorded in Western Amazonia. In addition, it is the second most diverse for this time window in low-latitude regions of South America, well after La Venta. The geographical range of *Ricardomys longidens*, *Microscleromys paradoxalis* and *M. cribriphilus* is now expanded 1,100 km to the south, during the Laventan SALMA, as a further testimony of the extent of the Pebasian biome at those times.

## Supporting information

**S1 File. Exploring the potential species number of *Microscleromys* from TAR-31 and La Venta.**
(DOCX)

**S2 File. Data and corresponding script(s) for each dental locus used for statistical analyses.**
(ZIP)

**S1 Table. Dental measurements (in millimeters) of rodent material from TAR-31 (Tarapoto, Peru; late middle Miocene) and La Venta (Colombia; late middle Miocene).** The previous assignation of the material from La Venta are specified (Walton, 1990, 1997). * indicates the specimens selected by Walton (1990) as holotypes, *** indicates the specimens selected in this study as holotypes.
(XLSX)

**S2 Table. Extinct and extant caviomorph taxa used for dental comparisons in this study.** For the references, see the manuscript. Fm., Formation; INGEMMET, Instituto Geológico Minero y Metalúrgico, Lima; MACN, Museo Argentino de Ciencias Naturales, Buenos Aires; MLP, Museo de Ciencias Naturales de La Plata; MMP, Museo Municipal De Ciencias Naturales Lorenzo Scaglia, Mar del Plata; MNHN-Bol, Museo Nacional de Historia Natural, La Paz, Bolivia; MNHN, Musée National d'Histoire Naturelle, Paris; MUSM, Museo de Historia Natural de la Universidad Nacional Mayor San Marcos, Lima; UM, Université de Montpellier.
(XLSX)

## Acknowledgments

Many thanks to M. Roddaz (GET, Toulouse, France), M.A. Custódio and R. Ventura (GEO-CRON/IG, Brasília, Brazil) and whoever helped us in the field and in the lab. We are particularly grateful to our drivers Giancarlo and Manuel, for their long standing help during the yearly field seasons. We warmly thank M.R. Ruiz-Monachesi (IBIGEO, Salta, Argentina) for discussing on statistics, his relevant suggestions and aid with respect to the Bayesian models and the osteological preparation of the *Galea*. We are much indebted to M.C. Madozzo-Jaén (Lillo, San Miguel de Tucumán, Argentina) and to M.E. Pérez (MEF, Trelew, Argentina) for discussing on phylogeny and homology of dental structures in cavioids. Many thanks to A. Álvarez and M.D. Ercoli (INECOA, San Salvador de Jujuy, Argentina) for providing us photos of collection material for dental comparisons. Christine Argot (MNHN, Paris, France), M. Reguero (MLP, La Plata, Argentina), L. Chornogubsky and A.G. Kramarz (MACN, Buenos Aires, Argentina), and M. Taglioretti (MMP, Mar del Plata, Argentina) kindly granted access to the collections under their care. We are much indebted to M. Gómez Pérez, the team of Museo Geológico Nacional—Servicio Geológico Colombiano and J.D. Carrillo (MNHN) for the correspondance between the field numbers and the collection numbers for the material from La Venta. We warmly thank B. Rondeau and L. Lena (LPG, Nantes, France) for access to a Keyence Digital Microscope facility; C. Cazeveille (Institut des Neurosciences de Montpellier [INM], France) for access to a electron microscope scanning facility; and A.-L. Charruault and R. Lebrun (ISEM, Montpellier, France) for µCT scan acquisitions and treatments. We thank the MRI and the LabEx CeMEB for access to the electron microscope and µCT scanning facilities (ISEM, Montpellier). We are indebted to A.-L. Charruault (ISEM, Montpellier, France) for the cast preparation of the rodent material from TAR-31. We gratefully thank the E. Beucler and O. Bourgeois (LPG, Nantes, France) in order to have allowed MB realizing a journey in Montpellier. This is ISEM publication 2021-225-Sud.

## Author Contributions

**Conceptualization:** Myriam Boivin, Laurent Marivaux, Pierre-Olivier Antoine.

**Formal analysis:** Myriam Boivin.

**Funding acquisition:** Myriam Boivin, Laurent Marivaux, François Pujos, Pierre-Olivier Antoine.

**Investigation:** Myriam Boivin, Laurent Marivaux, Walter Aguirre-Diaz, Aldo Benites-Palomino, Guillaume Billet, François Pujos, Rodolfo Salas-Gismondi, Narla S. Stutz, Julia V. Tejada-Lara, Rafael M. Varas-Malca, Anne H. Walton, Pierre-Olivier Antoine.

**Methodology:** Myriam Boivin.

**Resources:** Laurent Marivaux, Pierre-Olivier Antoine.

**Visualization:** Myriam Boivin, Laurent Marivaux, Narla S. Stutz.

**Writing – original draft:** Myriam Boivin.

**Writing – review & editing:** Myriam Boivin, Laurent Marivaux, Guillaume Billet, Narla S. Stutz, Julia V. Tejada-Lara, Anne H. Walton, Pierre-Olivier Antoine.

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
