## [Decision Letter · Decision Letter 0]

8 Sep 2021

PONE-D-21-23419Late middle Miocene caviomorph rodents from Tarapoto, Peruvian AmazoniaPLOS ONE

Dear Dr. Boivin,

Thank you for submitting your manuscript to PLOS ONE. After careful consideration, we feel that it has merit but does not fully meet PLOS ONE’s publication criteria as it currently stands. Therefore, we invite you to submit a revised version of the manuscript that addresses the points raised during the review process.

We look forward to receiving your revised manuscript.

Kind regards,

Jürgen Kriwet

Academic Editor

PLOS ONE

Journal Requirements:

"We are indebted to A.-L. Charruault (ISEM, Montpellier, France) for the cast preparation of the rodent material from TAR-31. We gratefully thank the E. Beucler, O. Bourgeois (LPG, Nantes, France) and the ‘Thème Terre’ of LPG in order to have covered the costs of a journey in Montpellier. This is ISEM publication 202X-0XX-Sud."

"LM and POA received funding from The Leakey Foundation and the LabEx CEBA (ANR-10-LABX-0025-01). POA received funding from the National Geographic Society and from the French ‘Agence Nationale de la Recherche’ (ANR) in the framework of the GAARAnti program (ANR-17-CE31-0009). LM received funding from the CoopIntEER CNRS-CONICET (n° 252540). POA and FP received funding from the ECOS-SUD/FONCyT (n° A-14U01) international collaboration programs. The funders had no role in study design, data collection and analysis, decision to publish, or preparation of the manuscript."

3. We note that Figure 1 in your submission contain map images which may be copyrighted. All PLOS content is published under the Creative Commons Attribution License (CC BY 4.0), which means that the manuscript, images, and Supporting Information files will be freely available online, and any third party is permitted to access, download, copy, distribute, and use these materials in any way, even commercially, with proper attribution. For these reasons, we cannot publish previously copyrighted maps or satellite images created using proprietary data, such as Google software (Google Maps, Street View, and Earth). For more information, see our copyright guidelines: http://journals.plos.org/plosone/s/licenses-and-copyright.

Additional Editor Comments (if provided):

Dear Authors:

Thank you very much again for submitting this manuscript to Plos One. Both reviewers commented on the manuscript and agree with your conclusions. They both find it a very valuable work that solves severl long-lasting taxonomic issues. There are however some minor issues that I'd like you to take care of before the manuscript can be accepted for publication.

Kind regards,

Jürgen Kriwet

Reviewers' comments:

Reviewer's Responses to Questions

**Comments to the Author**

1. Is the manuscript technically sound, and do the data support the conclusions?

Reviewer #1: Yes

Reviewer #2: Yes

2. Has the statistical analysis been performed appropriately and rigorously? 

Reviewer #1: Yes

Reviewer #2: I Don't Know

3. Have the authors made all data underlying the findings in their manuscript fully available?

Reviewer #1: Yes

Reviewer #2: Yes

4. Is the manuscript presented in an intelligible fashion and written in standard English?

Reviewer #1: Yes

Reviewer #2: Yes

5. Review Comments to the Author

Reviewer #1: In this work, the anatomy of dozens of newly collected specimens from Perú was exhaustively described (and illustrated), using up-to-date caviomorph dental terminology. The information provided in this study is pretty valuable as it resolves long-term uncertainties about several caviomorph taxa informally described 30 years ago in northern South America, a region much less studied than middle and high latitudes. I find this manuscript very interesting, competent, and methodologically sound and recommend it be published. Please, see the annotated manuscript file attached here for some specific minor comments and suggestions. Although my native language is not English, I cannot advise or detect severe mistakes in the manuscript. In addition, the figures, cites, tables, and analyses are adequate, as far as I can tell.

Reviewer #2: The manuscript is an impressive piece of work in describing not only the new late middle Miocene caviomorphs from Tarapoto but also offering a detailed formal description of the contemporaneous caviomorph rodents from La Venta so that the scientific names for these taxa finally become available for science. The comparisons are extensive and the paper shows the author’s deep knowledge of this systematic group in the South American Miocene.

The paper is well organized; the descriptions clear and the figures and tables all are well executed and necessary. However, I stumbled over one statement in the materials and Methods section where the authors say that “The description and comparison of this material [from La Venta] in the present work is based on published photos and drawings” (lines 195/196). Why did the authors not analyze the actual materials described by Walton? Were they not possible to loan for some reason? Maybe adding an explaining sentence would be advisable.

There are some comments/corrections in the annotated PDF, which all should be easy to accommodate by the authors.

I recommend publication of this manuscript after very minor revisions.

6. PLOS authors have the option to publish the peer review history of their article (what does this mean?). If published, this will include your full peer review and any attached files.

Reviewer #1: No

Reviewer #2: No

---

## [Author Response · Author response to Decision Letter 0]

21 Sep 2021

JOURNAL REQUIREMENTS:

EMAIL: 08/09/2021

1. We revised all elements so that our manuscript meets PLOS ONE's style requirements. Compared to PLOS ONE style templates, we noted and changed some minor style mistakes about the Headings 1 and 2, the Figure Citations and the place of the legend of some figures in the text. Moreover, we modified the file names following the indications of the Editor’s mail.

2. Yes, we modified in the Acknowledgments:

‘We gratefully thank the E. Beucler, O. Bourgeois (LPG, Nantes, France) and the ‘Thème Terre’ of LPG in order to have covered the costs of a journey in Montpellier.’

by

‘We gratefully thank the E. Beucler and O. Bourgeois (LPG, Nantes, France) in order to have allowed MB realizing a journey in Montpellier.’

Moreover, we added the Laboratoire de Planétologie et de Géodynamique de Nantes to the list of our funds.

3. We request the permission to Elsevier for using of the Figure 1 in Antoine et al. (2021) and the Figures 1 and 2 in Marivaux et al. (2020). These articles are not licensed under CC BY 4.0. They are Rightslink enabled. So, we replaced the maps (A, B) for new ones and deleted the stratigraphic log (C), as it is was recently published (Marivaux et al. 2020). We cited instead the fig. 2 in Marivaux et al. (2020).

REVIEWER 1 COMMENTS: pdf file

Please see Solórzano et al (2020). Late early Miocene caviomorph rodents from Laguna del Laja (~ 37° S), Cura-Mallín Formation, south-central Chile. Journal of South American Earth Sciences, 102, 102658.

Yes, exactly. Thank you. We changed ‘(Laguna del Laja and Pampa Castillo)’ by ‘(e.g., Laguna del Laja and Pampa Castillo)’. (line 79)

Despite its much lower diversity the Urumaco rodent fauna (Venezuela) is also interesting...

Yes, clearly the fauna from Urumaco Formation is very interesting and important in paleontological view. Here we had considered the most diverse caviomorph faunas at low latitudes of South America. As far we know, this fauna includes six caviomorph taxa (Sánchez-Villagra, 2012; Carrillo & Sánchez-Villagra, 2015), while some Peruvian faunas include equal number of taxa or more (e.g., 8 from Fitzcarrald Arch; Tejada-Lara et al. 2015). It is the reason why we preferred to not add it to our list, but we changed the sentence for more clarity: ‘At low latitudes of South America, two remarkable and species-rich local faunas are known, La Venta in Colombia (Honda Group; late middle Miocene) [18] and Acre in Brazil (Solimões Fm. [Formation]; late Miocene; see [19]).’ by ‘At low latitudes of South America, two remarkable and very species-rich caviomorph faunas are known, La Venta in Colombia (Honda Group; late middle Miocene) [18] and Acre in Brazil (Solimões Fm. [Formation]; late Miocene; see [19]).’ (lines 79-82)

Sorry, but compare with what?

We delete “and compare”. (line 106)

I believe this is part of your novel and interesting results. If that is the case, you should be a little more clear in this sentence.

 We changed the sentence by ‘This locality shares taxa or closely related taxa with La Venta fauna described by one of us (AHW) in the framework of her PhD [37]: Microsteiromys, Ricardomys, and Microscleromys.’. (lines 107-109)

I am not sure why you stated "Colloncuran". In above paragrahs you considered the studied rodent assemblage from Peru to be Laventan.

 We tentatively assigned to Nuyuyomys chinqaska the erethizontoid material from MD-67. According to Antoine et al. (2013, 2016), MD-67 is early middle Miocene (i.e., Colloncuran-Laventan SALMA). To clarify it, we added a sentence in the material and methods: ‘Three isolated teeth from MD-67 locality in Peru (MUSM 1974, 1975, and 4298; early middle Miocene) were originally attributed to cf. Microsteiromys sp [31]. They are reassigned here in light of the new material from Tarapoto.’. (lines 238-240)

but also some doubts abouth the cogeneric afinities betweenn Scleromys spp. in northern and southern South America

Yes, clearly. We added a sentence with respect to this point: ‘Moreover, some doubts were expressed about the cogeneric afinities between Scleromys spp. from northern versus southern South America [18,71].’. (lines 2784-2786). Moreover, we decided to follow the notation “Scleromys” proposed by Walton (1997) for our taxon and the other northern species of Scleromys and modified consequently the manuscript, Table 2, S1 Table and S2 Table.

I think this paragraph fits better in the "paleobiogeographical implcations" section

Yes clearly. We put it in the “paleobiogeographical implcations" section. In this section, we changed the paragraph:

‘The moderate or strong faunal similarities between MD-67, Fitzcarrald fauna, CTA-44 Top, TAR-31, and La Venta ([33,34,36] and see this work) suggest that the Eastern Cordillera of the North Andes and Pebas System would have not constituted barriers for mammals dispersals during the middle and earliest late Miocene at least for a region close to the Andes (Western Corridor) [36,44].’

by

‘With our results, the geographical ranges of Ricardomys longidens, Microscleromys paradoxalis and M. cribriphilus are now extended southerly to Peruvian Amazonia for the Laventan interval. More generally, strong taxonomic affinities between the vertebrate assemblages of La Venta (Colombia) and Fitzcarrald, Madre de Dios (MD-67), Iquitos, Contamana, and TAR-31 in Peruvian Amazonia support the existence of a single “Pebasian” biome throughout the middle–earliest late Miocene, and more specifically during the Laventan SALMA, as shown by their terrestrial and aquatic components (crocodylomorphs, chelonians, marsupials, rodents, primates, notoungulates, astrapotheres, litopterns, sirenians, and/or cetaceans; [21,26,31-34,36,158-160] and see this work). These moderate or strong faunal similarities suggest that the Eastern Cordillera of the North Andes and Pebas System would have not constituted barriers for mammalian dispersals during this interval at least for a region close to the Andes (Western Corridor) [36,44].’ (lines 2869-2880)

We changed the order of the concerned citations in the bibliography section and in the manuscript (ex-159 and 160 after ex-177).

This section appears to contradict with those stated above (also highlighted). With your available data I thing there is not much support for this ideas. Alternatively, you can stated that the idea it is just a plausible scenario that deserve further scrutiny. 

Yes, we agree, we deleted this part and the citations associated in the bibliography section. 

We thank very much the reviewer 1 for his comments.

REVIEWER 2 COMMENTS: 

The description and comparison of this material [from La Venta] in the present work is based on published photos and drawings” (lines 195/196). Why did the authors not analyze the actual materials described by Walton? Were they not possible to loan for some reason? Maybe adding an explaining sentence would be advisable.

Yes, it is very important point. Our original project was that two of our co-authors, Dr. Susan Walton (Springfield Technical Community College, USA) and Myriam Boivin (INECOA, Argentina), meet in USA or France (where the material from Tarapoto was studied before being returned to Peru) with original and/or casts of the material from La Venta and Tarapoto, respectively, in order to study all the material together. However, the covid pandemic and the confinement did not allow to manage it, to make and send casts to each other, or to loan request to the concerned Museums. Therefore, we decided to work from the pictures and drawings from Walton (1990, 1997).

We added a sentence in the Material and Methods section (La Venta paragraph) to clarify this point: “For reasons related to the covid pandemic and confinement, the original material from La Venta could not be directly observed.”. (lines 194-195)

Main comments in the pdf in chronological order:

I think you mixed up the descriptions for (B) and (C)?

Please double check the respective cross references in the text.

 Yes, thank you. We had mixed up in the legend. Moreover, we had mistake in the citation of the holotype of Microsteiromys jacobsi using the field number (FMNH PM 54677) instead of the collection number (FMNH PM 54672) in the manuscript. We checked and corrected each citation.

I suggest to mark the holotype specimen, e.g., with an asterisk symbol

I suggest to use the same symbol in the figures to mark the holotype specimens.

 We changed the mention ‘holotype’ by three asterisks and we added the sentence ‘The asterisks appoint the holotype material.’ in all legends of figures with a holotype. 

We thank very much the reviewer 2 for his comments.

MAIN PERSONNAL MODIFICATIONS: 

We registered our manuscript and the new taxon names on ZooBank and put the associated links on the manuscript body text.

We deeply apologize because we had made three errors in the statistical part in the original version of our manuscript (S1 and S2 Files). Noteworthily, these changes do not have implications for the taxonomic assignment of specimens of Microscleromys or for broader systematic conclusions.

1. We had calculated the correlation coefficient of Spearman between len and wid for all loci in the first version of our manuscript. However, these variables are normal or not depending on the locus. The correlation coefficient of Spearman is non-parametric and is generally used when two continuous variables have a distribution not normal or if one of them is not normal (Schober et al., 2018). Conversely, the correlation coefficient of Pearson is used in case of two normal continuous variables. Thus, we applied the Pearson coefficient when we had two normal variables. We modified the Table S1A (S1 File) and the scripts (S2 File). 

2. For the tests of average comparison, we deleted the results of non-parametric tests keeping only those of the parametric tests in case of normal variables. We modified the Table S1B,C (S1 File) and the scripts (S2 File).

3. We had made a mistake in the figure names in the S1 File. We modified it. For example, for Table S3A, we put Table S1A.

---

## [Editor Report · Decision Letter 1]

28 Sep 2021

Late middle Miocene caviomorph rodents from Tarapoto, Peruvian Amazonia

PONE-D-21-23419R1

Dear Dr. Boivin,

We’re pleased to inform you that your manuscript has been judged scientifically suitable for publication and will be formally accepted for publication once it meets all outstanding technical requirements.

Kind regards,

Jürgen Kriwet

Academic Editor

PLOS ONE

Additional Editor Comments (optional):

Dear authors:

Thank you very much for carefully considering all comments and suggestions posed by the reviewers. And alsomthank you for checking your statistical analyses and to correct the errors you found.

Kind regards,

Jürgen
---

## [Editor Report · Acceptance letter]

7 Oct 2021

PONE-D-21-23419R1 

Late middle Miocene caviomorph rodents from Tarapoto, Peruvian Amazonia 

Dear Dr. Boivin:

I'm pleased to inform you that your manuscript has been deemed suitable for publication in PLOS ONE. Congratulations! Your manuscript is now with our production department. 

Kind regards, 

on behalf of

Dr. Jürgen Kriwet 

Academic Editor

PLOS ONE